# Disentangled Representation Learning with the Gromov-Monge Gap

**Theo Uscidda**[1,2*]   **Luca Eyring**[2,3,4,*]   **Karsten Roth**[2,4,5]
**Fabian Theis**[2,3,4]   **Zeynep Akata**[2,3,4,†]   **Marco Cuturi**[1,6,†]

[1]CREST-ENSAE   [2]Helmholtz Munich   [3]TU Munich
[4]Munich Center of Machine Learning   [5]Tubingen AI Center   [6]Apple
theo.uscidda@ensae.fr   luca.eyring@tum.de

## Abstract

Learning disentangled representations from unlabelled data is a fundamental challenge in machine learning. Solving it may unlock other problems, such as generalization, interpretability, or fairness. Although remarkably challenging to solve in theory, disentanglement is often achieved in practice through prior matching. Furthermore, recent works have shown that prior matching approaches can be enhanced by leveraging geometrical considerations, e.g., by learning representations that preserve geometric features of the data, such as distances or angles between points. However, matching the prior while preserving geometric features is challenging, as a mapping that *fully* preserves these features while aligning the data distribution with the prior does not exist in general. To address these challenges, we introduce a novel approach to disentangled representation learning based on quadratic optimal transport. We formulate the problem using Gromov-Monge maps that transport one distribution onto another with minimal distortion of predefined geometric features, preserving them *as much as can be achieved*. To compute such maps, we propose the Gromov-Monge-Gap (GMG), a regularizer quantifying whether a map moves a reference distribution with minimal geometry distortion. We demonstrate the effectiveness of our approach for disentanglement across four standard benchmarks, outperforming other methods leveraging geometric considerations.

## 1 Introduction

Learning low-dimensional representations of high-dimensional data is a fundamental challenge in unsupervised deep learning (Bengio et al., 2014). Emphasis is put on learning representations that allow for efficient and robust adaptation across a wide range of tasks (Higgins et al., 2018; Locatello et al., 2019a). The fundamental property of *disentanglement* has shown significant promise to improve generalization (Locatello et al., 2020; Roth et al., 2023; Hsu et al., 2023; Barin-Pacela et al., 2024), interpretability and fairness (Locatello et al., 2019b; Träuble et al., 2021). Most works regard disentanglement as a one-to-one map between learned representations and ground-truth latent factors, effectively seeking to recover these factors from data alone in an unsupervised fashion. While unsupervised disentanglement is theoretically impossible (Locatello et al., 2019a), the inductive biases of autoencoder architectures ensure effective disentanglement in practice (Rolinek et al., 2019; Zietlow et al., 2021). Most approaches operate using variational autoencoder (VAE) frameworks (Kingma and Welling, 2014), using objectives that match latent VAE posteriors to factorized priors (Higgins et al., 2017; Kim and Mnih, 2018; Kumar et al., 2018; Burgess et al., 2018; Chen et al., 2018).

More recently, studies such as Gropp et al. (2020); Chen et al. (2020a); Lee et al. (2022); Horan et al. (2021); Nakagawa et al. (2023); Huh et al. (2023); Hahm et al. (2024) have provided a new perspective, showing that geometric constraints on representation spaces may also enable disentanglement. Typically, latent representations are encouraged to preserve key geometric features of the data distribution, such as (scaled) distances or angles between samples. Horan et al. (2021) even demonstrate that unsupervised disentanglement is *always* possible provided that the latent space

---

*Equal contribution
†Equal advising

is locally isometric to the data, further supporting the geometric desiderata. However, combining prior matching with these geometric aspects is challenging. In general, a mapping that perfectly aligns the data distribution with the prior while *fully* preserving the geometric features of interest may not exist. This leads to an *inherent trade-off*: Practitioners must carefully fine-tune regularization terms, either by altering prior matching to prioritize geometry preservation, or vice-versa.

In this work, we demonstrate how to *effectively combine geometric desiderata with prior matching* within the VAE framework, using optimal transport (OT) theory (Santambrogio, 2015; Peyré and Cuturi, 2019). By treating mappings from the data manifold to the latent space (encoders) or vice versa (decoders) as transport maps $T : \mathcal{X} \rightarrow \mathcal{Y}$, we can leverage the Gromov-{Monge, Wasserstein} paradigm (Sturm, 2023; Mémoli, 2011), which aligns two distributions by finding a mapping that minimizes the distortion between intra-domain cost functions defined on their supports. Specifically, we consider cost functions $c_{\mathcal{X}}(\mathbf{x}, \mathbf{x}')$ on $\mathcal{X}$ and $c_{\mathcal{Y}}(\mathbf{y}, \mathbf{y}')$ on $\mathcal{Y}$ that encode geometric features such as scaled distances or angles. Consequently, the resulting mapping transforms one distribution onto the other while preserving these geometric features *as much as possible*.

**Our Contribution**: A novel OT-based approach to disentanglement through geometric considerations.

(i) We address the challenge of learning disentangled representations using geometric constraints by leveraging Gromov-Monge mappings between the data and prior distributions. Since *fully* preserving geometric features—such as (scaled) distances or angles between points—during the alignment of these two distributions is generally impossible, we aim to find an alignment that, instead, minimizes the distortion of these features, thereby preserving them *as much as possible*.

(ii) Inspired by (Uscidda and Cuturi, 2023), we introduce the *Gromov-Monge Gap* (GMG), a regularizer that measures how closely a map $T$ approximates a Gromov-Monge map for costs $c_{\mathcal{X}}, c_{\mathcal{Y}}$. GMG measures whether $T$ transports distributions with minimal distortion w.r.t. $c_{\mathcal{X}}, c_{\mathcal{Y}}$. We propose an efficient procedure to compute GMG and describe how to integrate it within the VAE framework.

(iii) We show that when $c_{\mathcal{X}}$ and $c_{\mathcal{Y}}$ encode scaled distances or angles, the GMG and its finite-sample counterpart are weakly convex functions. In both cases, we precisely characterize the weak convexity constants and analyze their practical implications for practitioners.

(iv) Across four standard disentangled representation learning benchmarks, we show that incorporating geometry-preserving desiderata via the GMG significantly enhances disentanglement across various methods, from the standard $\beta$-VAE to the combination of $\beta$-TCVAE with HFS (Roth et al., 2023).

## 2 BACKGROUND: ON DISENTANGLEMENT, QUADRATIC-OT AND DISTORTION

### 2.1 DISENTANGLED REPRESENTATION LEARNING

**The Disentanglement Formalism.** Disentanglement has varying operational definitions. In this work, we follow the common understanding (Locatello et al., 2019a; 2020; Roth et al., 2023; Träuble et al., 2021; Higgins et al., 2017) where data $\mathbf{x}$ is generated by a process $p(\mathbf{x}|\mathbf{z})$ operating on ground-truth latent factors $\mathbf{z} \sim p(\mathbf{z})$, modeling underlying source of variations (s.a. object shape, color, background...). Given a dataset $\mathcal{D} = \{\mathbf{x}_i\}_{i=1}^N$, $\mathbf{x}_i \sim p_{\text{data}}$, unsupervised disentangled representation learning aims to find a mapping $e_\phi$ s.t. $e_\phi(\mathbf{x}_i) \approx \mathbb{E}[\mathbf{z}|\mathbf{x}_i]$, up to element-wise transformations. Notably, this is to be achieved without prior information on $p(\mathbf{z})$ and $p(\mathbf{x}|\mathbf{z})$.

**Unsupervised Disentanglement through Prior Matching.** Most unsupervised disentanglement methods operate on variational autoencoders (VAEs, Kingma and Welling (2014)), which define a generative model of the form $p_\theta(\mathbf{x}, \mathbf{z}) = p(\mathbf{z})p_\theta(\mathbf{x}|\mathbf{z})$. Here, $p_\theta(\mathbf{x}|\mathbf{z})$ is a product of exponential family distributions with parameters computed by a decoder $d_\theta(\mathbf{z})$. The latent prior $p(\mathbf{z})$ is usually chosen as a standard Gaussian $\mathcal{N}(\mathbf{z}|\mathbf{0}_d, \mathbf{I}_d)$, and the probabilistic encoder $q_\phi(\mathbf{z}|\mathbf{x})$ is implemented through neural networks $e_\phi(\mathbf{x}), \sigma_\phi(\mathbf{x})$ that predicts the latent parameters so that $q_\phi(\mathbf{z}|\mathbf{x}) = \mathcal{N}(\mathbf{z}|e_\phi(\mathbf{x}), \sigma_\phi^2(\mathbf{x}))$. The $\beta$-VAE (Higgins et al., 2017) achieves disentanglement by minimizing

$$\min_{\theta, \phi} \mathbb{E}_{\mathbf{x} \sim p_{\text{data}}, \mathbf{z} \sim q_\phi(\mathbf{z}|\mathbf{x})} [\underbrace{-\log p_\theta(\mathbf{x}|\mathbf{z})}_{\text{(i) reconstruction}} + \underbrace{\beta D_{\text{KL}}(q_\phi(\mathbf{z}|\mathbf{x})||p(\mathbf{z}))}_{\text{(ii) prior matching}}], \quad (1)$$

which enforces $\beta$-weighted prior matching on top of the reconstruction loss, assuming statistical factor independence (Roth et al., 2023). Several follow-ups refine latent prior matching through

different losses or prior choices (Rolinek et al., 2019; Kim and Mnih, 2018; Kumar et al., 2018; Burgess et al., 2018; Chen et al., 2018; Moor et al., 2021; Balabin et al., 2024).

**Disentanglement through a Geometric Lens.** Recent studies have revealed a fundamental connection between geometric structure preservation and disentanglement in learned representations (Gropp et al., 2020; Chen et al., 2020a; Lee et al., 2022; Nakagawa et al., 2023; Huh et al., 2023). This connection was theoretically established by Horan et al. (2021) proving that unsupervised disentanglement is always feasible when the generative factors are sufficiently non-Gaussian and maintain local *isometry* to the data. Our work builds directly on this insight by developing a learning framework that promotes representations that are as close as possible to being *isometric* to the data. To quantify geometric preservation between spaces of different dimensions, we leverage quadratic OT theory, which originated in Koopmans and Beckmann (1957) and was formalized by Mémoli (2011) as a framework for measuring isometric correspondence between metric spaces. We detail these tools in Section 2.2 before showing how they can be used to learn representations in Section 2.3. Additionally, in a concurrent work, Sotiropoulou and Alvarez-Melis (2024) also introduced the Gromov-Monge gap. However, they do not apply it for disentangled representational learning.

## 2.2 QUADRATIC OPTIMAL TRANSPORT

OT (Peyré and Cuturi, 2019) theory studies efficient ways to map a probability distribution onto another. *Linear* OT formulations, such as the Monge (1781) problem, require domains $\mathcal{X}, \mathcal{Y}$ that can be directly compared through a cost function $c(x, y)$ defined between their elements. When these distributions lie on incomparable domains, one must instead rely on *quadratic* formulations of OT (Q-OT), which instead compare geometric structure through *intra-domain* costs, also known as the Gromov-Monge (GM) and GW problems. In the context of representation learning, representation and data spaces are *incomparable* by design, which necessitates the use of Q-OT in this work.

**Gromov-{Monge, Wasserstein} Formulations.** Consider two compact $\mathcal{X} \subset \mathbb{R}^{d_{\mathcal{X}}}, \mathcal{Y} \subset \mathbb{R}^{d_{\mathcal{Y}}}$, equipped with *intra-domain* cost $c_{\mathcal{X}} : \mathcal{X} \times \mathcal{X} \to \mathbb{R}$ and $c_{\mathcal{Y}} : \mathcal{Y} \times \mathcal{Y} \to \mathbb{R}$. We assume that $c_{\mathcal{X}}$ and $c_{\mathcal{Y}}$ (or $-c_{\mathcal{X}}$ and $-c_{\mathcal{Y}}$) are CPD kernels (Def. (A.1)). For $p \in \mathcal{P}(\mathcal{X})$ and $q \in \mathcal{P}(\mathcal{Y})$, two distributions supported on each domain, the GM problem (Mmoli and Needham, 2022) seeks a map $T : \mathcal{X} \to \mathcal{Y}$ that push-forwards $p$ onto $q$, while minimizing the distortion of the costs:

$$\inf_{T:T\sharp p=q} \int_{\mathcal{X} \times \mathcal{X}} \tfrac{1}{2} |c_{\mathcal{X}}(\mathbf{x}, \mathbf{x}') - c_{\mathcal{Y}}(T(\mathbf{x}), T(\mathbf{x}'))|^2 \, \mathrm{d}p(\mathbf{x}) \, \mathrm{d}p(\mathbf{x}') \,. \tag{GMP}$$

When it exists, we call a solution $T^\star$ to (GMP) a *Gromov-Monge map* for costs $c_{\mathcal{X}}, c_{\mathcal{Y}}$. However, this formulation is ill-suited for discrete distributions $p, q$, as the constraint set might be empty in that case. Replacing maps by coupling $\pi \in \Pi(p, q)$, i.e. distributions on $\mathcal{X} \times \mathcal{Y}$ with marginals $p$ and $q$, we obtain the GW problem (Sturm, 2023; Mémoli, 2011)

$$\mathrm{GW}(p, q) := \min_{\pi \in \Pi(p,q)} \int_{(\mathcal{X} \times \mathcal{Y})^2} \tfrac{1}{2} |c_{\mathcal{X}}(\mathbf{x}, \mathbf{x}') - c_{\mathcal{Y}}(\mathbf{y}, \mathbf{y}')|^2 \, \mathrm{d}\pi(\mathbf{x}, \mathbf{y}) \, \mathrm{d}\pi(\mathbf{x}', \mathbf{y}') \,. \tag{GWP}$$

A solution $\pi^\star$ to (GWP) always exists, making $\mathrm{GW}(p, q)$ a well-defined quantity. It quantifies the minimal distortion of the geometries induced by $c_{\mathcal{X}}$ and $c_{\mathcal{Y}}$ achievable when coupling $p$ and $q$.

**Discrete Solvers.** When both $p$ and $q$ are instantiated as samples, GW Prob. (GWP) translates to a quadratic assignment problem, whose objective can be regularized using entropy (Cuturi, 2013; Peyré et al., 2016). For empirical measures $p_n = \frac{1}{n} \sum_{i=1}^{n} \delta_{\mathbf{x}_i}, q_n = \frac{1}{n} \sum_{j=1}^{n} \delta_{\mathbf{y}_j}$ and $\varepsilon \geq 0$, we set:

$$\mathrm{GW}_\varepsilon(p_n, q_n) := \min_{\mathbf{P} \in U_n} \sum_{i,j,i',j'=1}^{n} (\mathbf{C}_{\mathcal{X}_{i,i'}} - \mathbf{C}_{\mathcal{Y}_{j,j'}})^2 \mathbf{P}_{i,j} \mathbf{P}_{i',j'} - \varepsilon H(\mathbf{P}) \,, \tag{EGWP}$$

with $\mathbf{C}_{\mathcal{X}} = [c_{\mathcal{X}}(\mathbf{x}_i, \mathbf{x}_{i'})]_{i,i'}, \mathbf{C}_{\mathcal{Y}} = [c_{\mathcal{Y}}(\mathbf{y}_j, \mathbf{y}_{j'})]_{j,j'} \in \mathbb{R}^{n \times n}, U_n = \{\mathbf{P} \in \mathbb{R}_+^{n \times n}, \mathbf{P}\mathbf{1}_n = \mathbf{P}^T \mathbf{1}_n = \frac{1}{n}\mathbf{1}_n\}$ and $H(\mathbf{P}) = -\sum_{i,j=1}^{n} \mathbf{P}_{i,j} \log(\mathbf{P}_{i,j})$. As $\varepsilon \to 0$, we recover $\mathrm{GW}_0^{c_{\mathcal{X}},c_{\mathcal{Y}}} = \mathrm{GW}^{c_{\mathcal{X}},c_{\mathcal{Y}}}$. Entropic regularization improves computational performance, as we can solve (EGWP) using a scheme that iterates the Sinkhorn algorithm (see Appendix D.3 for full details). This solver has $\mathcal{O}(n^2)$ memory complexity. Its time complexity depends on $c_{\mathcal{X}}, c_{\mathcal{Y}}$. For general $c_{\mathcal{X}}, c_{\mathcal{Y}}$, it runs in $\mathcal{O}(n^3)$. However, for the most common practical choices of $c_{\mathcal{X}} = c_{\mathcal{Y}} = \langle \cdot, \cdot \rangle$ or $c_{\mathcal{X}} = c_{\mathcal{Y}} = \| \cdot - \cdot \|_2^2$, it can also be reduced to $\mathcal{O}(n^2(d_{\mathcal{X}} + d_{\mathcal{Y}}))$, as detailed by Scetbon et al. (2022, §3 & Alg. 2).

## 2.3 Distortion in Representation Learning

Given an arbitrary map $T : \mathcal{X} \to \mathcal{Y}$, we consider how it can be learned to preserve predefined geometric features. In a VAE, $T$ may represent either the encoder $e_\phi$, which generates latent codes from the data, or the decoder $d_\theta$, which reconstructs the data from these codes. In the case of the encoder, $\mathcal{X}$ corresponds to the data, and $\mathcal{Y}$ is the latent space, while for the decoder, these roles are swapped. Assuming that $d_\theta$ perfectly reconstructs the data from the latents produced by $e_\phi$, i.e., $e_\phi \circ d_\theta = \mathrm{Id}$, the preservation of geometric features by either $e_\phi$ or $d_\theta$ becomes equivalent. Therefore, in the following sections, we refer to $T$ as either the encoder or the decoder without loss of generality.

We encode geometric features using a cost function for each domain: $c_\mathcal{X} : \mathcal{X} \times \mathcal{X} \to \mathbb{R}$ and $c_\mathcal{Y} : \mathcal{Y} \times \mathcal{Y} \to \mathbb{R}$. Ideally, $T$ should preserve geometry, which means that $T$ preserves costs, that is, $c_\mathcal{X}(\mathbf{x}, \mathbf{x}') \approx c_\mathcal{Y}(T(\mathbf{x}), T(\mathbf{x}'))$ for $\mathbf{x}, \mathbf{x}' \in \mathcal{X}$. In practice, two types of costs are often used:

**[i] (Scaled) squared L2 distance**: $c_\mathcal{X}(\mathbf{x}, \mathbf{x}') = \|\mathbf{x} - \mathbf{x}'\|_2^2$ and $c_\mathcal{Y}(\mathbf{y}, \mathbf{y}') = \alpha^2 \|\mathbf{y} - \mathbf{y}'\|_2^2$, with $\alpha > 0$. A map $T$ preserving $c_\mathcal{X}, c_\mathcal{Y}$ preserves the scaled distances between the points, i.e. it is a *scaled isometry*. When $\alpha = 1$, we recover the standard definition of an *isometry*.

**[ii] Cosine-Similarity**: $c_\mathcal{X}(\mathbf{x}, \mathbf{x}') = \mathrm{cos\text{-}sim}(\mathbf{x}, \mathbf{x}') := \langle \frac{\mathbf{x}}{\|\mathbf{x}\|_2}, \frac{\mathbf{x}'}{\|\mathbf{x}'\|_2} \rangle$ and $c_\mathcal{Y}(\mathbf{y}, \mathbf{y}') = \mathrm{cos\text{-}sim}(\mathbf{y}, \mathbf{y}')$ similarly. On has $\mathrm{cos\text{-}sim}(\mathbf{x}, \mathbf{x}') = \cos(\theta_{\mathbf{x}, \mathbf{x}'})$ where $\theta_{\mathbf{x}, \mathbf{x}'}$ is the angle between $\mathbf{x}$ and $\mathbf{x}'$. A map $T$ preserving $c_\mathcal{X}, c_\mathcal{Y}$ then preserves the angles between the points, i.e. it is a *conformal map*. Note that if $T$ is (scaled) isometry (see above), it is a conformal map.

We refer to these costs via $\mathbf{L2^2}$ for **[i]** with $\alpha = 1$, $\mathbf{ScL2^2}$ for **[i]** with $\alpha \neq 1$ and **Cos** for **[ii]**. Introducing a reference distribution $r \in \mathcal{P}(\mathcal{X})$, weighting the areas of $\mathcal{X}$ where we penalize deviations of $c_\mathcal{X}(\mathbf{x}, \mathbf{x}')$ from $c_\mathcal{Y}(T(\mathbf{x}), T(\mathbf{x}'))$, we can quantify this property using the following criterion:

**Definition 2.1** (Distortion). The distortion (DST) of a map $T$, for cost functions $c_\mathcal{X}, c_\mathcal{Y}$ and reference distribution $r$, is defined as:

$$\mathrm{DST}_r(T) := \int_{\mathcal{X} \times \mathcal{X}} \tfrac{1}{2}(c_\mathcal{X}(\mathbf{x}, \mathbf{x}') - c_\mathcal{Y}(T(\mathbf{x}), T(\mathbf{x}')))^2 \, dr(\mathbf{x}) \, dr(\mathbf{x}') . \qquad \text{(DST)}$$

$\mathrm{DST}_r(T)$ quantifies how much $T$ distorts geometric features encoded by $c_\mathcal{X}, c_\mathcal{Y}$ on the support of $r$, that is, when $\mathrm{DST}_r(T) = 0$, one has $c_\mathcal{X}(\mathbf{x}, \mathbf{x}') = c_\mathcal{Y}(T(\mathbf{x}), T(\mathbf{x}'))$ for $\mathbf{x}, \mathbf{x}' \in \mathrm{Spt}(r)$.

**Distortion as a Loss for Representation Learning.** Nakagawa et al. (2023) suggest promoting geometry preservation by regularizing the encoder $d_\theta$ using the DST, with $\mathbf{ScL2^2}$ as costs, and the latent representation as reference distribution, namely $r = e_\phi \sharp p_{\mathrm{data}}$. While they use it within a WAE, we translate their objective to VAE setting adopted later in the paper. This results in:

$$\min_{\theta, \phi} \mathbb{E}_{\mathbf{x} \sim p_{\mathrm{data}}, \mathbf{z} \sim q_\phi(\mathbf{z}|\mathbf{x})} \underbrace{[- \log p_\theta(\mathbf{x}|\mathbf{z})}_{\text{(i) reconstruction}} + \underbrace{\beta D_{\mathrm{KL}}(q_\phi(\mathbf{z}|\mathbf{x}) \| p(\mathbf{z}))]}_{\text{(ii) prior matching}} + \underbrace{\lambda \, \mathrm{DST}_r(d_\theta)}_{\text{(iii) geom. preservation}} , \quad \lambda > 0 \qquad (2)$$

Given the choice of the costs, $d_\theta$ is distortion-free (i.e., $\mathrm{DST}_r(d_\theta) = 0$), if it is a scaled isometry.

**Challenges Arising from a Mixed Loss.** Since a scaled isometry that maps the prior onto the data distribution may not exist, there is an inherent trade-off between minimizing terms (ii) and (iii), which are responsible for achieving practical disentanglement. As these terms cannot be simultaneously minimized to 0, the DST loss will move away from accurately matching the prior, which will negatively impact the quality of the learned latent representations. This naturally raises the question of how to avoid this over-penalization. Instead of seeking a distortion-free decoder, we should seek a decoder that transports the prior to the data distribution with *minimal distortion* of the costs $c_\mathcal{X}, c_\mathcal{Y}$. In other words, the decoder should be a Gromov-Monge map between the prior and the data distributions for costs $c_\mathcal{X}, c_\mathcal{Y}$ (see 2.2). Conversely, if we choose to regularize the encoder, the same reasoning applies by swapping the roles of the prior and the data distribution. In the next section, we introduce a regularizer to fit Gromov-Monge maps, which we will use as a replacement to the DST.

## 3 Disentanglement with the Gromov-Monge gap

Building on these geometric preservation principles, we introduce in §3.1 the Gromov-Monge Gap (GMG), a regularizer that measures whether a map moves distributions while preserving geometric

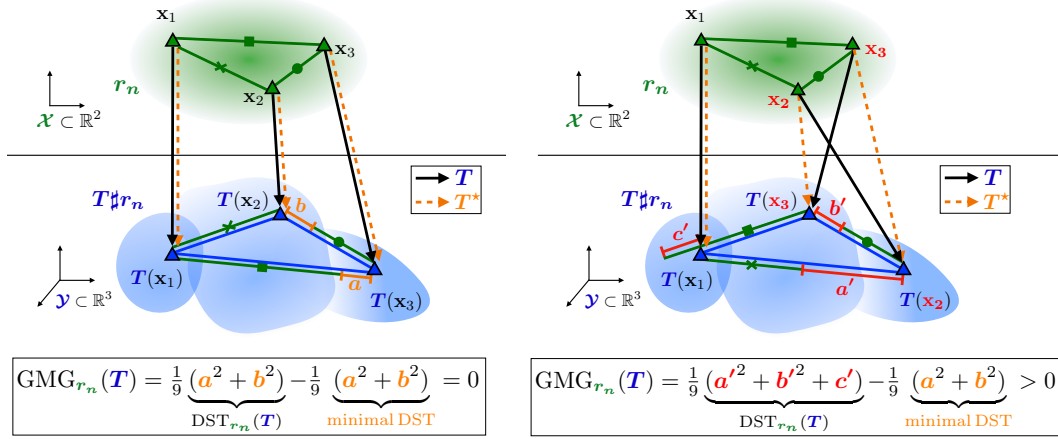

Figure 1: Sketch of $\text{GMG}_{r_n}(T)$ for two different maps $T$. We use a discrete reference distribution $r_n$ on 3 points, and $c_{\mathcal{X}} = c_{\mathcal{Y}} = \| \cdot - \cdot \|_2$, hence measuring if $T$ minimally distorts the distances. On the left, $T$ is the optimal map, $T^\star$, and maps the three points with minimal (yet non-zero) distortion, which is measured as the sum of the squared lengths of the orange segments. This results in $\text{GMG}_{r_n}(T) = 0$. On the right, $T$ swaps two points compared to $T^\star$ specifically, $\mathbf{x_2}$ and $\mathbf{x_3}$ causing a higher distortion than the minimal one, and measured as the sum of the squared lengths of the red segments. This results in $\text{GMG}_{r_n}(T) > 0$, equal to the gap between these distortions.

features as much as possible, i.e., minimizing distortion while fitting the marginal constraints. §3.2 then shows how the GMG can be efficiently computed from samples to be practically applicable in the VAE framework. This transitions into §3.3 studying (weak) convexity properties of the GMG, as an operator. Finally, in §3.4, we describe how to integrate the GMG with disentangled representation learning objectives, effectively combining prior matching with geometric constraints.

## 3.1 An Efficient Gap Formulation for Distortion

Recently, Uscidda and Cuturi (2023) introduced the Monge gap, a regularizer that measures whether a map $T$ transports a reference distribution at the minimal displacement cost. Building on this concept, we replace "displacement" with "distortion" to introduce the Gromov-Monge gap, a regularizer that assesses whether a map $T$ transports a reference distribution at the minimal distortion cost.

**Definition 3.1** (Gromov-Monge gap)**.** The Gromov-Monge gap (GMG) of a map $T$, for cost functions $c_{\mathcal{X}}, c_{\mathcal{Y}}$ and reference distribution $r$, is defined as:

$$\text{GMG}_r(T) := \text{DST}_r(T) - \text{GW}(r, T\sharp r) \tag{GMG}$$

We recall from Eq. (GWP) that $\text{GW}(r, T\sharp r)$ represents the minimal distortion of $c_{\mathcal{X}}, c_{\mathcal{Y}}$ achievable when transporting $r$ to $T\sharp r$. Thus, the GMG quantifies the difference between the distortion incurred when transporting $r$ to $T\sharp r$ via $T$, and this minimal distortion. Formally, when Prob. (GMP) and Prob. (GWP) between $r$ and $T\sharp r$ are equivalent, the the GMG is the suboptimality gap of $T$ in Prob. (GMP). This is the case, for example, when $r$ is a density and $c_{\mathcal{X}} = c_{\mathcal{Y}} = \langle \cdot, \cdot \rangle$ (Dumont et al., 2022). Otherwise, the GMG is the suboptimality gap of $\pi = (\text{I}_d, T)\sharp r$ in Prob. (GWP) between $r$ and $T\sharp r$. In light of this, it is a well-defined quantity and:

- **The GMG measures how close $T$ is to be a Gromov-Monge map for costs $c_{\mathcal{X}}, c_{\mathcal{Y}}$.** Indeed, $\text{GMG}_r(T) \geq 0$ with equality if $T$ is a Gromov-Monge map solution of Prob. (GMP) between $r$ and $T\sharp r$, i.e., $T$ moves $r$ with minimal (but eventually non zero) distortion; see App. B.1.
- **When transport without distortion is possible, the GMG coincides with the distortion.** When there exists another map $U : \mathcal{X} \to \mathcal{Y}$ transporting $r$ to $T\sharp r$ with zero distortion, i.e., $U\sharp r = T\sharp r$ and $\text{DST}_r(U) = 0$, then $\text{GMG}_r(T) = \text{DST}_r(T)$. Indeed, $\text{GW}(r, T\sharp r) = 0$ in that case, as the coupling $\pi = (\text{Id}, U)\sharp r$ sets the GW objective to zero, thereby minimizing it.

The last point illustrates how the GMG functions as a *debiased distortion*. It compares the distortion induced by $T$ to a baseline distortion, defined as the minimal achievable distortion when transforming $r$ into $T\sharp r$. Thus, when transformation without distortion is achievable, this baseline becomes zero, and the GMG equals the distortion. Consequently, the GMG offers the optimal compromise: it avoids

the over-penalization induced by the distortion when fully preserving $c_\mathcal{X}, c_\mathcal{Y}$ is not feasible, yet it coincides with it when such full preservation is feasible. See Fig. 1 for a simple illustration.

**The Influence of the Reference Distribution.** A crucial property of $\mathrm{DST}_r$ is that if $T$ transforms $r$ without distortion, it will also apply distortion-free to any distribution $s$ whose support is contained within that of $r$. Formally, if $\mathrm{DST}_r(T) = 0$ and $s \in \mathcal{P}(\mathcal{X})$ with $\mathrm{Spt}(s) \subseteq \mathrm{Spt}(r)$, then $\mathrm{DST}_s(T) = 0$. This raises a question for the GMG: If $T$ maps $r$ with minimal distortion, does it similarly map $s$ with minimal distortion? We answer this question positively with Prop. (3.2).

**Proposition 3.2.** *If* $\mathrm{GMG}_r(T) = 0$, $\forall s \in \mathcal{P}(\mathcal{X})$ *s.t.* $\mathrm{Spt}(s) \subseteq \mathrm{Spt}(r)$, *one has* $\mathrm{GMG}_s(T) = 0$.

## 3.2 Estimation and Computation from Samples

**Plug-In Estimation.** In practice, we estimate Eq. (DST) and Eq. (GMG) from samples $\mathbf{x}_1, ..., \mathbf{x}_n \sim r$. We consider the empirical version $r_n := \frac{1}{n} \sum_{i=1}^{n} \delta_{\mathbf{x}_i}$ of $r$ and use plug-in estimators, i.e.

$$\mathrm{DST}_{r_n}(T) = \frac{1}{n^2} \sum_{i,j=1}^{n} \left( c_\mathcal{X}(\mathbf{x}_i, \mathbf{x}_j) - c_\mathcal{Y}(T(\mathbf{x}_i), T(\mathbf{x}_j)) \right)^2, \tag{3}$$

and $\mathrm{GMG}_{r_n}(T) = \mathrm{DST}_{r_n}(T) - \mathrm{GW}(r_n, T\sharp r_n)$, where $T\sharp r_n = \frac{1}{n} \sum_{i=1}^{n} \delta_{T(\mathbf{x}_i)}$.

**Efficient and Stable Computation.**
Computing the GMG requires solving a discrete GW problem between $r_n$ and $T\sharp r_n$ to get $\mathrm{GW}(r_n, T\sharp r_n)$. We compute this term using an entropic regularization $\varepsilon \geq 0$, as in Eq. (EGWP):

$$\begin{aligned} \mathrm{GMG}_{r_n,\varepsilon}(T) &:= \mathrm{DST}_{r_n}(T) \\ &- \mathrm{GW}_\varepsilon(r_n, T\sharp r_n). \end{aligned} \tag{4}$$

Choosing $\varepsilon = 0$, we recover $\mathrm{GMG}_{r_n,0}(T) = \mathrm{GMG}_{r_n}(T)$. More-over, the entropic estimator preserves

---

**Algorithm 1** $\mathrm{GMG}(\mathbf{x}_1, \ldots \mathbf{x}_n, T, \varepsilon)$.

1: **Require:** $\mathbf{x}_1, \ldots, \mathbf{x}_n \sim r$; map $T$; entropic solver `GW`, entropic regularization scale $\varepsilon_0$ (default $= 0.1$), statistic operator on cost matrix `stat` (default $=$ `mean`).
2: $\mathbf{t}_1, \ldots \mathbf{t}_n \leftarrow T(\mathbf{x}_1), \ldots, T(\mathbf{x}_n)$.
3: $\mathbf{C}_\mathcal{X} \leftarrow [c_\mathcal{X}(\mathbf{x}_i, \mathbf{x}_{i'})]_{1 \leq i,i' \leq n}$ ▷ usually $\mathcal{O}(n^2 d_\mathcal{X})$
4: $\mathbf{C}_\mathcal{Y} \leftarrow [c_\mathcal{Y}(\mathbf{t}_j, \mathbf{t}_{j'})]_{1 \leq j,j' \leq n}$ ▷ usually $\mathcal{O}(n^2 d_\mathcal{Y})$
5: $\mathrm{DST} \leftarrow \mathtt{mean}((\mathbf{C}_\mathcal{X} - \mathbf{C}_\mathcal{Y})^2)$ ▷ $n^2$
6: $\varepsilon \leftarrow \varepsilon_0 \cdot \mathtt{stat}(\mathbf{C}_\mathcal{X}) \cdot \mathtt{stat}(\mathbf{C}_\mathcal{Y})$ ▷ usually $\mathcal{O}(n^2)$
7: $\mathrm{GW} \leftarrow \mathtt{GW}(\mathbf{C}_\mathcal{X}, \mathbf{C}_\mathcal{Y}, \varepsilon)$ ▷ $\mathcal{O}(n^3)$ or $\mathcal{O}(n^2(d_\mathcal{X} + d_\mathcal{Y}))$
8: **return** DST - GW

---

positivity, as for $\varepsilon \geq 0$, $\mathrm{GMG}_{r_n,\varepsilon}(T) \geq 0$ (see B.3). We compute $\mathrm{GW}_\varepsilon(r_n, T\sharp r_n)$ using Peyré et al. (2016)'s solver introduced in 2.2. We use the implementation provided by `ott-jax` (Cuturi et al., 2022). In practice, we select $\varepsilon$ based on (positive) statistics from the cost matrices $\mathbf{C}_\mathcal{X}, \mathbf{C}_\mathcal{Y}$. We define a scale $\varepsilon_0$ and set $\varepsilon = \varepsilon_0 \cdot \mathtt{stat}(\mathbf{C}_\mathcal{X}) \cdot \mathtt{stat}(\mathbf{C}_\mathcal{Y})$. Standard options for the statistic include $\mathtt{stat} \in \{\mathtt{mean}, \mathtt{max}, \mathtt{std}\}$. This procedure is equivalent to running the entropic GW solver on the re-scaled cost matrices $\mathbf{C}_\mathcal{X}/\mathtt{stat}(\mathbf{C}_\mathcal{X})$ and $\mathbf{C}_\mathcal{Y}/\mathtt{stat}(\mathbf{C}_\mathcal{Y})$ with $\varepsilon = \varepsilon_0$; see App. B.2. It is a common practical trick, initially suggested for stabilizing the Sinkhorn algorithm Cuturi (2013).

**Computational Complexity.** For usual costs, including inner products, $\ell_p^q$ distances, and standard CPD kernels, the computation of $c_\mathcal{X}(\mathbf{x}, \mathbf{x}')$ (resp. $c_\mathcal{Y}(\mathbf{y}, \mathbf{y}')$) can be done in $\mathcal{O}(d_\mathcal{X})$ time (resp. $\mathcal{O}(d_\mathcal{Y})$). Consequently, the DST can be computed in $\mathcal{O}(n^2(d_\mathcal{X} + d_\mathcal{Y}))$ time. Furthermore, as discussed in § 2.2, the time complexity of the entropic GW solver is $\mathcal{O}(n^3)$ in general, but can be reduced to $\mathcal{O}(n^2(d_\mathcal{X} + d_\mathcal{Y}))$ when $c_\mathcal{X} = c_\mathcal{Y} = \langle \cdot, \cdot \rangle$, or $c_\mathcal{X} = \|\cdot - \cdot\|_2^2, c_\mathcal{Y} = \alpha \|\cdot - \cdot\|_2^2$. Therefore, since the cosine similarity is equivalent to the inner product, up to pre-normalization of $\mathbf{x}_i$ and $T(\mathbf{x}_i)$, this solver runs in $\mathcal{O}(n^2(d_\mathcal{X} + d_\mathcal{Y}))$ for the costs of interest (**Sc**)**L2²** and **Cos**. The complete algorithm, along with a time complexity analysis of each step, is described in Alg. 1. We stress that, in all cases and for any cost function, the time complexity depends linearly on the dimensions of the source and target spaces, $d_\mathcal{X}$ and $d_\mathcal{Y}$, making the GMG scalable to high-dimensional distributions.

## 3.3 (Weak) Convexity of the Gromov-Monge gap

As laid out, the GMG can be used as a regularization loss to push any model $T$ to move distributions with minimal distortion. A natural question arises: is this regularizer convex? In the following, we study the convexity of $T \mapsto \mathrm{GMG}_r(T)$ and its finite-sample counterpart $T \mapsto \mathrm{GMG}_{r_n}(T)$. We focus on the costs **L2** and **Cos**. For simplicity, we replace **Cos** with $\langle \cdot, \cdot \rangle$, as these costs are equivalent, up

to normalization of $r$ and $T$. We respectively denote by $\mathrm{GMG}_r^2$ and $\mathrm{GMG}_r^{\langle\cdot,\cdot\rangle}$ the GMG for these costs. We start by introducing a weaker notion of convexity, previously defined on $\mathbb{R}^d$ (Davis et al., 2018), which we extend here to $L_2(r) := \{T \mid \|T\|_{L_2(r)}^2 := \int_{\mathcal{X}} \|T(\mathbf{x})\|_2^2 \, \mathrm{d}r(\mathbf{x}) < +\infty\}$.

**Definition 3.3** (Weak convexity.). With $\gamma > 0$, a functional $\mathcal{F} : L_2(r) \to \mathbb{R}$ is $\gamma$-weakly convex if $\mathcal{F}_\gamma : T \mapsto \mathcal{F}(T) + \frac{\gamma}{2}\|T\|_{L_2(r)}^2$ is convex.

A weakly convex functional is convex up to an additive quadratic perturbation. The weak convexity constant $\gamma$ quantifies the magnitude of this perturbation and indicates a degree of non-convexity of $\mathcal{F}$. A lower $\gamma$ suggests that $\mathcal{F}$ is closer to being convex, while a higher $\gamma$ indicates greater non-convexity.

**Theorem 3.4.** *Both $\mathrm{GMG}_r^2$ and $\mathrm{GMG}_r^{\langle\cdot,\cdot\rangle}$, as well as their finite sample versions, are weakly convex.*

- ***Finite sample.*** *We note $\mathbf{X} \in \mathbb{R}^{n \times d}$ the matrix that stores the $\mathbf{x}_i$, i.e. the support of $r_n$, as rows. Then, (i) $\mathrm{GMG}_{r_n}^2$ and (ii) $\mathrm{GMG}_{r_n}^{\langle\cdot,\cdot\rangle}$ are respectively (i) $\gamma_{2,n}$ and (ii) $\gamma_{inner,n}$-weakly convex, where: $\gamma_{inner,n} = \lambda_{\max}(\frac{1}{n}\mathbf{X}\mathbf{X}^\top) - \lambda_{\min}(\frac{1}{n}\mathbf{X}\mathbf{X}^\top)$ and $\gamma_{2,n} = \gamma_{inner,n} + \max_{i=1\ldots n}\|\mathbf{x}_i\|_2^2$.*

- ***Asymptotic.*** *(i) $\mathrm{GMG}_r^2$ and (ii) $\mathrm{GMG}_r^{\langle\cdot,\cdot\rangle}$ are respectively (i) $\gamma_2$ and (ii) $\gamma_{inner}$-weakly convex, where: $\gamma_{inner} = \lambda_{\max}(\mathbb{E}_{\mathbf{x}\sim r}[\mathbf{x}\mathbf{x}^\top])$ and $\gamma_{2,n} = \gamma_{inner} + \max_{\mathbf{x}\in\mathrm{Spt}(r)}\|\mathbf{x}\|_2^2$.*

From a practitioner's perspective, we analyze the insights provided by Thm. (3.4) in three parts.

- First, we have $\gamma_2 \geq \gamma_{\mathrm{inner}}$. Therefore, $\mathrm{GMG}_r^2$ is less convex than $\mathrm{GMG}_r^{\langle\cdot,\cdot\rangle}$, making it harder to optimize, and the same argument holds for their estimator. In other words, we provably recover that, in practice, preserving the (scaled) distances is harder than simply preserving the angles.

- Second, as $\gamma_{\mathrm{inner}} = \lambda_{\max}(\mathbb{E}_{\mathbf{x}\sim r}[\mathbf{x}\mathbf{x}^\top]) \geq \lambda_{\max}(\mathrm{Cov}_{\mathbf{x}\sim r}[\mathbf{x}])$, this exhibits a tradeoff w.r.t. Prop. (3.2): by choosing a bigger reference distribution $r$, we trade the convexity of the GMG. For $\gamma_2$, the dependency in $r$ is even worse. In practice, we then choose $r$ with support as small as possible, precisely where we want $T$ to move points with minimal distortion.

- Third, and most surprising, the finite sample GMG is more convex in high dimension. Indeed, $\gamma_{\mathrm{inner},n}$ is the spectral width of $\frac{1}{n}\mathbf{X}\mathbf{X}^\top$, which contains the (rescaled) inner products between $\mathbf{x}_i \sim r$. When $n > d$, $\lambda_{\min}(\mathbf{X}\mathbf{X}^\top) = 0$ as $\mathrm{rank}(\mathbf{X}\mathbf{X}^\top) = d$. Then, $\gamma_{\mathrm{inner},n}$ increases, which in turn decreases the GMG's convexity. However, when $d > n$, $\lambda_{\min}(\mathbf{X}\mathbf{X}^\top) > 0$ if $\mathbf{X}$ is full rank. Intuitively, $\mathrm{GMG}_{r_n}^{\langle\cdot,\cdot\rangle}$ is nearly convex when $\mathbf{X}\mathbf{X}^\top$ is well conditioned. Assuming that the $\mathbf{x}_i$ are normalized, this might happen in high dimension, as they will be orthogonal with high probability. This suggests that, contrary to the insights provided by the statistical OT literature (Weed and Bach, 2017; Zhang et al., 2023), the GMG might not benefit a large sample size.

### 3.4 LEARNING WITH THE GROMOV-MONGE GAP

**General Learning Procedure.** Given a source $p$ and a target distribution $q$, we can use the GMG to guide a parameterized map $T_\theta$ towards approximating a Gromov-Monge map between $p$ and $q$. We handle the marginal constraint $T_\theta \sharp p = q$ separately through a fitting loss $\Delta(T_\theta, p, q)$. Provided any reference $r$ s.t. $\mathrm{Spt}(p) \subset \mathrm{Spt}(r)$, we minimize

$$\min_\theta \Delta(T_\theta, p, q) + \lambda \mathrm{GMG}_r(T_\theta) \tag{5}$$

$\Delta$ can operate on paired (e.g., in VAE, the reconstruction loss), or unpaired (e.g., in VAE, the KL loss in VAE), samples of $p$ and $q$. Note that, in theory and as stated in Prop. (3.2), we can select any reference $r$ such that $\mathrm{Spt}(p) \subset \mathrm{Spt}(r)$. However, based on the insights from Thm. (3.3), we typically choose $r$ with minimal support size and, in practice, set $r = p$. This learning procedure is illustrated in Fig. 2, where we also explore the effect of replacing $\mathrm{GMG}_r(T_\theta)$ by $\mathrm{DST}_r(T_\theta)$ in Eq. (5).

**VAE Learning Procedure.** In the VAE setting, we can use the GMG promote the (i) encoder $e_\phi$ or the (ii) decoder $d_\theta$ to mimic a Gromov-Monge map. In (i) we use $r_e = p_{\mathrm{data}}$ the data distribution as reference $r$, while in (ii) we use the latent distribution $r_d = e_\phi \sharp p_{\mathrm{data}}$. Introducing weightings $\lambda_e, \lambda_d \geq 0$, determining which mapping we regularize, this remains to minimize

$$\min_{\theta,\phi} \mathbb{E}_{\mathbf{x}\sim p_{\mathrm{data}}, \mathbf{z}\sim q_\phi(\mathbf{z}|\mathbf{x})} \underbrace{[-\log p_\theta(\mathbf{x}|\mathbf{z})}_{\text{(i) reconstruction}} + \underbrace{\beta D_{\mathrm{KL}}(q_\phi(\mathbf{z}|\mathbf{x})\|p(\mathbf{z}))]}_{\text{(ii) prior matching}} + \underbrace{\lambda_e \mathrm{GMG}_{r_e}(e_\phi) + \lambda_d \mathrm{GMG}_{r_d}(d_\theta)}_{\text{(iii) geom. preservation}},$$

$$\tag{6}$$

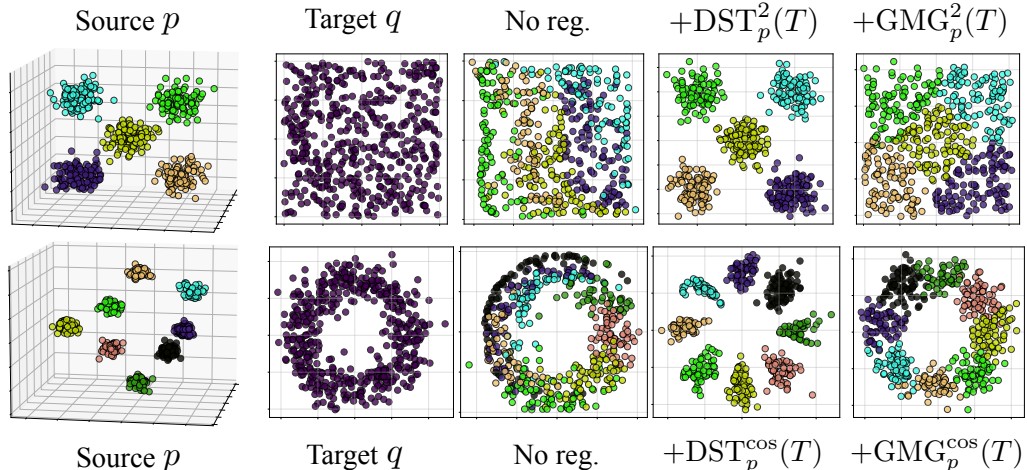

Figure 2: Learning of geometry-preserving maps with the DST and the GMG. Provided a mixture of Gaussian source distribution $p$ and a uniform or circular target distribution $q$, we minimize a fitting loss together with a geometry-preserving regularization. Each line correspond to a different data setup and cost. As costs on the top line, we use $\mathbf{L2^2} = \| \cdot - \cdot \|_2^2$, to preserve distances between points while on the bottom line, we use $\mathbf{Cos} = \text{cos-sim}(\cdot, \cdot)$, to preserve angles.

With this loss, prior matching and geometric desiderata can be efficiently combined, as terms (ii) and (iii) can simultaneously be 0. Note that this loss can be easily extended to more advanced prior matching objectives, such as the $\beta$-TCVAE loss (Chen et al., 2018), and can be combined with other regularizers, such as the HFS (Roth et al., 2023). We explore this strategy in experiments § 4.2.

# 4 EXPERIMENTS

## 4.1 GMG VS. DST: ILLUSTRATIVE EXAMPLE

In Fig. 2, we start by illustrating the difference between the DST and GMG. We train $T_\theta$ to map from a 3D source $p$ to a 2D target $q$ (first and second column) by minimizing $\mathcal{L}(\theta) := S_\varepsilon(T_\theta \sharp p, q) + R(T_\theta)$, where $S_\varepsilon$ is the Sinkhorn divergence (Feydy et al., 2019). We compare three settings, [i] no regularization $R = 0$, [ii] $R = \text{DST}_p$ as regularizer , and [iii] $R = \text{GMG}_p$. For all cases, we plot the transported distribution $T_\theta \sharp p$ after training. Without regularization (third column), we fit the marginal constraint $T_\theta \sharp p = q$ but do not preserve the geometric features. With the DST (fourth column), we preserve geometric features but do not fit the marginal constraint, i.e., $T_\theta \sharp p \neq q$. On the other hand, with the GMG (fifth column), we get the best compromise by approximating a Gromov-Monge map: we fit $T_\theta \sharp p = q$, while preserving the geometric features as fully as possible.

## 4.2 LEVERAGING THE GROMOV-MONGE GAP FOR DISENTANGLEMENT

**Experimental Setup.** Having demonstrated how the GMG enables (i) fitting a marginal constraint while (ii) preserving geometric features as much as possible, we now apply it to disentangled representation learning. Our primary goal is to investigate whether the GMG results in enhanced disentanglement compared to the DST by efficiently combining (i) prior matching with (ii) geometric constraints on the representation space. Moreover, we aim to determine which $c_\mathcal{X}, c_\mathcal{Y}$ to choose and what part of the pipeline should be regularized, the encoder $e_\phi$, or the decoder $d_\theta$.

- **Baselines.** We use the standard $\beta$-VAE and $\beta$-TCVAE as our starting models, with the option to apply the recent HFS regularization (Roth et al., 2023) to each, resulting in a total of four base configurations. Note that the latter does not leverage geometric constraints; it is only used to enhance prior matching. We then investigate the effect of various geometry-preserving regularizations on disentanglement on top of these four base configurations. We consider GMG, DST and the Jacobian-based (Jac) regularization (Lee et al., 2022) discussed in 2.1. Given the inclusion of the DST, we naturally consider Nakagawa et al. (2023) as a baseline.
- **Metrics.** We evaluate the learned representations using the **DCI-D** (Eastwood and Williams, 2018) as it was found that it is the metric most suitable for measuring the disentanglement (Locatello et al., 2020; Dittadi et al., 2021). We report mean and standard deviation over 5 seeds.

Table 1: Effect of different regularization on disentanglement (DCI-D on Shapes3D). We highlight the best method per regularization type ($\mathbf{L2^2}$, $\mathbf{ScL2^2}$, or $\mathbf{Cos}$), and the **best**/second best per column.

| | $\beta$-VAE | $\beta$-TCVAE | $\beta$-VAE + HFS | $\beta$-TCVAE + HFS |
|---|---|---|---|---|
| Base | 65.8 $\pm$15.6 | 75.0 $\pm$3.4 | 88.1 $\pm$7.4 | 90.2 $\pm$7.5 |
| $\mathbf{L2^2}$ : $c_{\mathcal{X}} = c_{\mathcal{Y}} = \|\cdot - \cdot\|_2^2$ | | | | |
| + Enc-DST | 59.6 $\pm$6.9 | 75.7 $\pm$3.0 | 88.7 $\pm$7.1 | 90.3 $\pm$7.9 |
| + Enc-GMG | 62.3 $\pm$8.4 | 75.4 $\pm$5.3 | 88.4 $\pm$7.7 | 90.1 $\pm$4.3 |
| + Dec-DST | 71.5 $\pm$3.6 | 75.8 $\pm$6.6 | 92.1 $\pm$9.7 | 90.9 $\pm$7.6 |
| + Dec-GMG | 72.0 $\pm$8.5 | 78.9 $\pm$5.0 | 92.5 $\pm$4.4 | 91.7 $\pm$6.0 |
| $\mathbf{ScL2^2}$ : $c_{\mathcal{X}} = \|\cdot - \cdot\|_2^2, c_{\mathcal{Y}} = \alpha\|\cdot - \cdot\|_2^2, \alpha > 0$ learnable | | | | |
| + Jac | 61.4 $\pm$12.8 | 76.7 $\pm$4.5 | 90.5 $\pm$3.8 | 91.5 $\pm$5.6 |
| + Enc-DST | 65.8 $\pm$11.9 | 73.0 $\pm$7.9 | 92.4 $\pm$3.7 | 89.2 $\pm$3.8 |
| + Enc-GMG | 65.1 $\pm$5.5 | 76.1 $\pm$7.7 | 90.8 $\pm$9.2 | 92.0 $\pm$5.3 |
| + Dec-DST | 67.4 $\pm$7.1 | 77.9 $\pm$4.5 | 93.2 $\pm$9.7 | 94.5 $\pm$6.9 |
| + Dec-GMG | 70.0 $\pm$5.9 | 81.0 $\pm$3.2 | 93.3 $\pm$8.6 | 96.1 $\pm$3.8 |
| $\mathbf{Cos}$ : $c_{\mathcal{X}} = c_{\mathcal{Y}} = \text{cos-sim}(\cdot, \cdot)$ | | | | |
| + Enc-DST | 69.2 $\pm$9.1 | 77.2 $\pm$7.5 | 87.7 $\pm$7.7 | 90.5 $\pm$5.9 |
| + Enc-GMG | 70.9 $\pm$9.5 | 79.6 $\pm$6.6 | 92.5 $\pm$5.9 | 93.5 $\pm$6.9 |
| + Dec-DST | 76.8 $\pm$4.1 | 81.3 $\pm$4.7 | 87.5 $\pm$3.3 | 91.9 $\pm$9.4 |
| + Dec-GMG | **82.1** $\pm$4.5 | **83.7** $\pm$8.8 | **95.7** $\pm$5.8 | **96.9** $\pm$4.9 |

- **Datasets.** We benchmark over four $64 \times 64$ image datasets: Shapes3D (Kim and Mnih, 2018), DSprites (Higgins et al., 2017), SmallNORB (LeCun et al., 2004), and Cars3D (Reed et al., 2015).

- **Hyperparameters.** To ensure a fair experimental comparison, we follow recent works (Locatello et al., 2019a; 2020; Roth et al., 2023) by using the same architecture and hyperparameters. We perform a similar small grid search for the weighting terms: $\beta$ for the KL loss and $\gamma$ for HFS. Additionally, we include the weighting $\lambda$ for the geometry-preserving regularizer in the grid search. Note that we search over the same loss weightings $\lambda$ for DST, GMG, and Jac. For all experiments with the GMG, we compute it with Alg. 1, and systematically use $\varepsilon_0 = 0.1$ and $\texttt{stat} = \texttt{mean}$. As a result, $\varepsilon_0$ is *not* included in the grid search. We conduct an ablation study on $\varepsilon_0$ in App. D.3. We use a batch size of $n = 64$. At this scale, the computational cost of compute the GMG loss for a batch is negligible, about 3 milliseconds. See App. C for full details on hyperparameters.

**Which costs $c_{\mathcal{X}}, c_{\mathcal{Y}}$ should we choose?** The first question that naturally arises when using a geometry-preserving regularizer is: Which geometric features should be preserved? Previous works (Lee et al., 2022; Nakagawa et al., 2023; Huh et al., 2023) focused on preserving scaled distances between points, with the scale being learnable. We follow and extend this approach by also investigating plain distances and angles. This leads to three choices for $c_{\mathcal{X}}, c_{\mathcal{Y}}$ ($\mathbf{L2^2}, \mathbf{ScL2^2}, \mathbf{Cos}$), as introduced in §2.3, following the hierarchy of geometry-preserving mappings proposed in Lee et al. (2022). We benchmark these on Shapes3D across various settings including DST, GMG, and Jac. Table 1 shows that angle preservation ($\mathbf{Cos}$), previously unconsidered for disentangled representation learning, consistently outperforms (scaled) distance preservation. This result is intuitive, as preserving angles imposes a weaker constraint, allowing for greater latent space expressiveness. In practice, preserving scaled distances seems to overly restrict the expressiveness of the latent space.

**Should we regularize the encoder $e_\phi$, or the decoder $d_\theta$?** The next question we aim to answer is whether the decoder or encoder should be regularized. Therefore, we follow the previous setup on Shapes3D and benchmark all geometry-preserving regularizers on $e_\phi$ and $d_\theta$ as reported in Table 1. We find that regularizing the decoder is beneficial over regularizing the encoder. We hypothesize this is due to the regularization of $d_\theta$ offering a stronger signal as its gradients impact both $\phi$ and $\theta$, as in this case, the reference $r = e_\phi \sharp p_{\text{data}}$ is the distribution of encoded images. Our findings align with prior works (Lee et al., 2022; Nakagawa et al., 2023), which focus on regularizing $d_\theta$ yet do not offer this type of analysis. Additionally, we find that the GMG consistently outperforms the DST over all. Overall, the GMG on $d_\theta$ with $\mathbf{Cos}$ achieves best DCI-D results over all baselines. Consequently, moving forward we regularize the decoder for angle preservation.

Table 2: Effect of GMG and DST with **Cos** as costs on disentanglement, as measured by **DCI-D**, over three datasets. We highlight the **best**, and second best result for each dataset and method.

| With **Cos** costs | $\beta$-VAE | $\beta$-TCVAE | $\beta$-VAE + HFS | $\beta$-TCVAE + HFS |
|---|---|---|---|---|
| **DSprites** (Higgins et al., 2017) | | | | |
| Base | 26.2 $\pm$18.5 | 32.3 $\pm$19.3 | 33.6 $\pm$17.9 | 48.7 $\pm$10.2 |
| + Dec-DST | 28.6 $\pm$19.3 | 32.4 $\pm$8.5 | 39.3 $\pm$18.1 | 49.0 $\pm$11.2 |
| + Dec-GMG | **39.5** $\pm$15.2 | **42.2** $\pm$3.6 | **46.7** $\pm$2.0 | **50.1** $\pm$8.5 |
| **SmallNORB** (LeCun et al., 2004) | | | | |
| Base | 26.8 $\pm$0.2 | 29.8 $\pm$0.4 | 26.8 $\pm$0.2 | 29.8 $\pm$0.4 |
| + Dec-DST | 28.2 $\pm$0.3 | **29.9** $\pm$0.4 | 28.2 $\pm$0.3 | **29.9** $\pm$0.4 |
| + Dec-GMG | **28.3** $\pm$0.6 | **29.9** $\pm$0.5 | **28.3** $\pm$0.6 | **29.9** $\pm$0.5 |
| **Cars3D** (Reed et al., 2015) | | | | |
| Base | 29.6 $\pm$5.7 | 32.3 $\pm$4.6 | 29.6 $\pm$5.7 | 32.3 $\pm$4.6 |
| + Dec-DST | 26.8 $\pm$3.6 | 33.7 $\pm$4.2 | 26.8 $\pm$3.6 | 33.7 $\pm$4.2 |
| + Dec-GMG | **30.1** $\pm$5.6 | **36.4** $\pm$5.7 | **30.1** $\pm$5.6 | **36.4** $\pm$5.7 |

**GMG Consistently Enhances Disentanglement.** To further validate our findings, we benchmark the GMG for decoder regularization with angle preservation (**Cos**) against its distortion counterpart across three more datasets. We report full results in Table 2. Again we observe that the GMG outperforms or performs equally well to its distortion equivalent. Note that for SmallNORB and Cars3D, we found no benefits with respect to DCI-D in adding HFS and obtained the best results without it. We emphasize that using the GMG with **Cos** regularization significantly improves results for all datasets. This establishes the GMG as an effective tool for enhanced disentanglement.

**Stability of the GMG.** Furthermore, we investigate the stability of the GMG compared to the DST. Computing the GMG using Algorithm 1 involves solving an optimization problem through a GW solver. This experiment aims to demonstrate that our proposed method for solving the GW problem leads to a stable loss in the GMG . We assess the stability by measuring the alignment of GMG gradients $\nabla_\theta \text{GMG}_{r_n}(T_\theta)$ across 5 randomly sampled batches from each dataset $r_n \sim \mathcal{D}$, for a fixed neural map $T_\theta$. We repeat this procedure for each of the four datasets $\mathcal{D}$ considered and apply the same methodology to the DST. Figure 4 presents results. We observe that the GMG's gradients exhibit significantly higher alignment compared to those of the DST, demonstrating greater stability of our proposed regularizer.

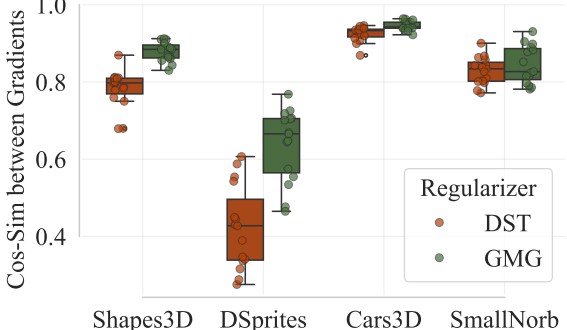

Figure 3: Stability analysis for the DST and the GMG applied to the decoder $d_\theta$ with cost **Cos**. For each regularizer, we assess stability by measuring the alignment of its gradients across 5 batches. We compute the cosine similarity between all pairs of gradients. Values closer to 1 indicates better alignment, hence better stability.

This suggests that the GMG, which accounts for the the minimal distortion, effectively mitigates the inherent variability inherent of the distortion, leading to more stable gradient computations.

## 5 CONCLUSION

In this work, we introduce an OT perspective on unsupervised disentangled representation learning to incorporate general latent geometrical constraints. We derive the GMG, a provably weakly convex regularizer that measures whether a map $T$ transports a fixed reference distribution with minimal distortion of some predefined geometric features. By formulating disentangled representation learning as a transport problem, we integrate the GMG into standard training objectives, allowing for incorporating and studying various geometric constraints on the learned representation spaces. We show significant performance benefits of our approach on four standard disentanglement benchmarks.

## 6 REPRODUCIBLITY

In this work, we introduce the GMG, computed as detailed in Algorithm 1. To facilitate reproducibility, we provide the implementation code for computing the GMG on source and target batches in Appendix E. Comprehensive proofs, including all underlying assumptions, are presented in Appendix B. For our experiments on disentanglement benchmarks, we adhere to standard practices, employing streamlined preprocessing across all datasets. Detailed descriptions of these procedures are available in Appendix C. All experiments described in this paper can be conducted using a single RTX 2080TI GPU, ensuring accessibility and replicability of our results.

## ACKNOWLEDGEMENTS

Co-funded by the European Union (ERC, DeepCell - 101054957). Views and opinions expressed are, however, those of the author(s) only and do not necessarily reflect those of the European Union or the European Research Council. Neither the European Union nor the granting authority can be held responsible for them. Luca Eyring and Karsten Roth thank the European Laboratory for Learning and Intelligent Systems (ELLIS) PhD program for support. Karsten Roth also thanks the International Max Planck Research School for Intelligent Systems (IMPRS-IS) for support. Zeynep Akata was supported by BMBF FKZ: 01IS18039A, by the ERC (853489 - DEXIM), by EXC number 2064/1 project number 390727645. Fabian J. Theis consults for Immunai Inc., Singularity Bio B.V., CytoReason Ltd, Cellarity, and has ownership interest in Dermagnostix GmbH and Cellarity

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

APPENDIX

The Appendix is organized as follows:

- Section A provides additional background information to supplement the main text.
- Section B presents all theoretical proofs, including detailed assumptions.
- Section C outlines comprehensive experimental details, ensuring reproducibility.
- Section D offers supplementary empirical results that further support our findings.
- Section E includes the implementation code for computing the GMG.

## A  ADDITIONAL BACKGROUND

### A.1  REMINDERS ON MONGE AND KANTOROVICH OT

In this section, we recall the Monge and Kantorovich formulations of OT, which we will use to prove various results. These are the classical formulations of OT. Although we introduce them here after discussing the Gromov-Monge and Gromov-Wasserstein formulations, it should be noted that they are generally introduced beforehand. Indeed, the Gromov-Monge and Gromov-Wasserstein formulations were historically developed to derive OT formulations for comparing measures supported on incomparable spaces.

**Monge Formulation.**  Instead of intra-domain cost functions, we consider here an *inter-domain* continuous cost function $c : \mathcal{X} \times \mathcal{Y} \to \mathbb{R}$. This assumes that we have a meaningful way to compare elements $\mathbf{x}, \mathbf{y}$ from the source and target domains. The Monge (1781) problem (MP) between $p \in \mathcal{P}(\mathcal{X})$ and $p \in \mathcal{P}(\mathcal{Y})$ consists of finding a map $T : \mathcal{X} \to \mathcal{Y}$ that push-forwards $p$ onto $p$, while minimizing the average displacement cost quantified by $c$

$$\inf_{T:T\sharp p=p} \int_{\mathcal{X}} c(\mathbf{x}, T(\mathbf{x})) \, \mathrm{d}p(\mathbf{x}) \,. \tag{MP}$$

We call any solution $T^\star$ to this problem a Monge map between $p$ and $q$ for cost $c$. Similarly to the Gromov-Monge Problem (GMP), solving the Monge Problem (MP) is difficult, as the constraint set is not convex and might be empty, especially when $p, q$ are discrete.

**Kantorovich Formulation.**  Instead of transport maps, the Kantorovich problem (KP) seeks a couplings $\pi \in \Pi(p, q)$:

$$\mathrm{W}(p, q) := \min_{\pi \in \Pi(p,q)} \int_{\mathcal{X} \times \mathcal{Y}} c(\mathbf{x}, \mathbf{y}) \, \mathrm{d}\pi(\mathbf{x}, \mathbf{y}) \,. \tag{KP}$$

An optimal coupling $\pi^\star$ solution of (KP), always exists. Studying the equivalence between (MP) and (KP) is easier than in the Gromov-Monge and Gromov-Wasserstein cases. Indeed, when (MP) is feasible, the Monge and Kantorovich formulations coincide and $\pi^\star = (\mathrm{Id}, T^\star)\sharp p$.

### A.2  CONDITIONALLY POSITIVE KERNELS

In this section, we recall the definition of a conditionally positive kernel, which is involved in multiple proofs relying on the linearization of the Gromov-Wasserstein problem as a Kantorovich problem.

**Definition A.1.**  A kernel $k : \mathbb{R}^d \times \mathbb{R}^d \to \mathbb{R}$ is CPD, i.e., conditionally positive, if it is symmetric and for any $\mathbf{x}_1, ..., \mathbf{x}_n \in \mathbb{R}^d$ and $\mathbf{a} \in \mathbb{R}^n$ s.t. $\mathbf{a}^\top \mathbf{1}_n = 0$, one has

$$\sum_{i,j=1}^{n} \mathbf{a}_i \mathbf{a}_j \, k(\mathbf{x}_i, \mathbf{x}_j) \geq 0$$

CPD include all positive kernels, such as the inner-product $k(\mathbf{x}, \mathbf{x}') = \langle \mathbf{x}, \mathbf{x}' \rangle$, or the cosine similarity $k(\mathbf{x}, \mathbf{x}') = \text{cos-sim}(\mathbf{x}, \mathbf{x}') = \langle \frac{\mathbf{x}}{\|\mathbf{x}\|_2}, \frac{\mathbf{x}'}{\|\mathbf{x}'\|_2} \rangle$, but also the (scaled) negative squared Euclidean distance $k(\mathbf{x}, \mathbf{x}') = -\alpha\|\mathbf{x} - \mathbf{x}'\|_2^2, \alpha > 0$. Therefore, each of the costs of interest is either a conditionally positive kernel - for the inner product and the cosine distance - or its opposite is - for the squared Euclidean distance. Additionally, CPD kernels also include more exotic cost functions, such as:

- The RBF kernel $k(\mathbf{x}, \mathbf{x}') = \exp(-\|\mathbf{x} - \mathbf{y}\|_2^2/\gamma), \gamma > 0$.
- The power kernel $k(\mathbf{x}, \mathbf{x}') = -\|\mathbf{x} - \mathbf{x}'\|_2^p, 0 < p < 2$.
- The thin plate spline kernel $k(\mathbf{x}, \mathbf{x}') = \|\mathbf{x} - \mathbf{x}'\|_2^2 \log(\|\mathbf{x} - \mathbf{x}'\|)$
- The inverse multi-quadratic kernel $k(\mathbf{x}, \mathbf{x}') = 1/\sqrt{\|\mathbf{x} - \mathbf{x}'\|_2^2 + c^2}, c \in \mathbb{R}$.

As a result, the family of CPD kernels includes a large variety of cost functions, which can be used to define various GMGs.

## B  PROOFS

### B.1  THE GMG CHARACTERIZES GROMOV-MONGE OPTIMALITY

We show here that if $\mathrm{GMG}_r(T) = 0$, then $T$ is a Gromov-Monge map between $r$ and $T\sharp r$ for costs $c_{\mathcal{X}}, c_{\mathcal{Y}}$. As the set of deterministic couplings $\{\pi_F := (\mathrm{I}_d, F)\sharp r | F : \mathcal{X} \to \mathcal{Y}, F\sharp r = T\sharp r\} \subset \Pi(r, T\sharp r)$, we immediately get that

$$\inf_{F\sharp r = T\sharp r} \int_{\mathcal{X} \times \mathcal{X}} (c_{\mathcal{X}}(\mathbf{x}, \mathbf{x}') - c_{\mathcal{Y}}(F(\mathbf{x}), F(\mathbf{x}')))^2 \, dr(\mathbf{x}) \, dr(\mathbf{x}') \geq \mathrm{GW}(r, T\sharp r) \tag{7}$$

On the other hand, if $\mathrm{GMG}_r(T) = 0$, one has

$$\mathrm{GW}(r, T\sharp r) = \int_{\mathcal{X} \times \mathcal{X}} (c_{\mathcal{X}}(\mathbf{x}, \mathbf{x}') - c_{\mathcal{Y}}(T(\mathbf{x}), T(\mathbf{x}')))^2 \, dr(\mathbf{x}) \, dr(\mathbf{x}') \tag{8}$$

Therefore, combining Eq. (7) and Eq. (8), we get

$$\inf_{F\sharp r = T\sharp r} \int_{\mathcal{X} \times \mathcal{X}} (c_{\mathcal{X}}(\mathbf{x}, \mathbf{x}') - c_{\mathcal{Y}}(F(\mathbf{x}), F(\mathbf{x}')))^2 \, dr(\mathbf{x}) \, dr(\mathbf{x}')$$
$$= \int_{\mathcal{X} \times \mathcal{X}} (c_{\mathcal{X}}(\mathbf{x}, \mathbf{x}') - c_{\mathcal{Y}}(T(\mathbf{x}), T(\mathbf{x}')))^2 \, dr(\mathbf{x}) \, dr(\mathbf{x}') \tag{9}$$

Finally, as $T$ naturally satisfies the marginal constraint, we conclude that $T$ is a Gromov-Monge map between $r$ and $T\sharp r$ for costs $c_{\mathcal{X}}, c_{\mathcal{Y}}$.

### B.2  ON RESCALING THE COSTS MATRICES IN THE ENTROPIC GW SOLVER

We remind, from Eq. (EGWP), that

$$\mathrm{GW}_\varepsilon(p_n, q_n) = \min_{\mathbf{P} \in U_n} \sum_{i,j,i',j'=1}^{n} (\mathbf{C}_{\mathcal{X}_{i,i'}} - \mathbf{C}_{\mathcal{Y}_{j,j'}})^2 \mathbf{P}_{i,j} \mathbf{P}_{i',j'} - \varepsilon H(\mathbf{P}). \tag{10}$$

By developing each terms, and using the fact that $\mathbf{P} \in U_n$, we get

$$\mathrm{GW}_\varepsilon(p_n, q_n) = \min_{\mathbf{P} \in U_n} \frac{1}{n^2}\langle \mathbf{C}_{\mathcal{X}}^{\odot 2} \mathbf{1}_n, \mathbf{1}_n \rangle + \frac{1}{n^2}\langle \mathbf{C}_{\mathcal{Y}}^{\odot 2} \mathbf{1}_n, \mathbf{1}_n \rangle - 2\langle \mathbf{C}_{\mathcal{X}} \mathbf{P} \mathbf{C}_{\mathcal{Y}}, \mathbf{P} \rangle - \varepsilon H(\mathbf{P})$$
$$= \frac{1}{n^2}\langle \mathbf{C}_{\mathcal{X}}^{\odot 2} \mathbf{1}_n, \mathbf{1}_n \rangle + \frac{1}{n^2}\langle \mathbf{C}_{\mathcal{Y}}^{\odot 2} \mathbf{1}_n, \mathbf{1}_n \rangle + \min_{\mathbf{P} \in U_n} -2\langle \mathbf{C}_{\mathcal{X}} \mathbf{P} \mathbf{C}_{\mathcal{Y}}, \mathbf{P} \rangle - \varepsilon H(\mathbf{P}) \tag{11}$$

where $\mathbf{C}_{\mathcal{X}}^{\odot 2} = \mathbf{C}_{\mathcal{X}} \odot \mathbf{C}_{\mathcal{X}}$, with $\odot$ the Hadamard (i.e., elementwise) product, and similarly for $\mathbf{C}_{\mathcal{Y}}^{\odot 2}$. As we can see that the two terms on the left do not depend on $\mathbf{P}$, they do not impact the minimization, an OT coupling $\mathbf{P}^\star$ solving the problem satisfies:

$$\mathbf{P}^\star \in \arg\min_{\mathbf{P} \in U_n} -2\langle \mathbf{C}_{\mathcal{X}} \mathbf{P} \mathbf{C}_{\mathcal{Y}}, \mathbf{P} \rangle - \varepsilon H(\mathbf{P}). \tag{12}$$

As a result, if we now replace $\mathbf{C}_{\mathcal{X}}$ and $\mathbf{C}_{\mathcal{Y}}$ by $\mathbf{C}_{\mathcal{X}}/\mathtt{stat}(\mathbf{C}_{\mathcal{X}})$ and $\mathbf{C}_{\mathcal{Y}}/\mathtt{stat}(\mathbf{C}_{\mathcal{Y}})$, respectively, the new OT coupling $\tilde{\mathbf{P}}^\star$ solving the problem satisfies

$$\tilde{\mathbf{P}}^\star \in \arg\min_{\mathbf{P} \in U_n} -2\langle \frac{\mathbf{C}_{\mathcal{X}}}{\mathtt{stat}(\mathbf{C}_{\mathcal{X}})} \mathbf{P} \frac{\mathbf{C}_{\mathcal{Y}}}{\mathtt{stat}(\mathbf{C}_{\mathcal{Y}})}, \mathbf{P} \rangle - \varepsilon H(\mathbf{P})$$
$$\Leftrightarrow \quad \tilde{\mathbf{P}}^\star \in \arg\min_{\mathbf{P} \in U_n} -\frac{2}{\mathtt{stat}(\mathbf{C}_{\mathcal{X}}) \cdot \mathtt{stat}(\mathbf{C}_{\mathcal{Y}})} \langle \mathbf{C}_{\mathcal{X}} \mathbf{P} \mathbf{C}_{\mathcal{Y}}, \mathbf{P} \rangle - \varepsilon H(\mathbf{P}) \tag{13}$$
$$\Leftrightarrow \quad \tilde{\mathbf{P}}^\star \in \arg\min_{\mathbf{P} \in U_n} -2\langle \mathbf{C}_{\mathcal{X}} \mathbf{P} \mathbf{C}_{\mathcal{Y}}, \mathbf{P} \rangle - \mathtt{stat}(\mathbf{C}_{\mathcal{X}}) \cdot \mathtt{stat}(\mathbf{C}_{\mathcal{Y}}) \cdot \varepsilon H(\mathbf{P}),$$

where in the last line, we use the fact that $\mathrm{stat}(\mathbf{C}_{\mathcal{X}}), \mathrm{stat}(\mathbf{C}_{\mathcal{Y}}) > 0$. This yields the desired equivalence on scaling the cost matrix and adapting the entropic regularization strength.

### B.3    Positivity of the Entropic GMG estimator

Recall that

$$\mathrm{GMG}_{r_n, \varepsilon}(T) = \mathrm{DST}_{r_n}(T) - \mathrm{GW}_{\varepsilon}(r_n, T \sharp r_n)$$

$$= \mathrm{DST}_{r_n}(T) - \min_{\mathbf{P} \in U_n} \sum_{i,j,i',j'=1}^{n} (c_{\mathcal{X}}(\mathbf{x}_i, \mathbf{x}_j) - c_{\mathcal{Y}}(\mathbf{y}_i, \mathbf{y}_j))^2 \mathbf{P}_{ij} \mathbf{P}_{i'j'} - \varepsilon H(\mathbf{P}),$$

For any coupling $\mathbf{P} \in U_n$, since $-\varepsilon H(\mathbf{P}) = -\varepsilon \sum_{i,j=1}^{n} \mathbf{P}_{ij} \log(\mathbf{P}_{ij}) < 0$, one has:

$$\sum_{i,j,i',j'=1}^{n} (c_{\mathcal{X}}(\mathbf{x}_i, \mathbf{x}_j) - c_{\mathcal{Y}}(\mathbf{y}_i, \mathbf{y}_j))^2 \mathbf{P}_{ij} \mathbf{P}_{i'j'} - \varepsilon H(\mathbf{P}) < \sum_{i,j,i',j'=1}^{n} (c_{\mathcal{X}}(\mathbf{x}_i, \mathbf{x}_j) - c_{\mathcal{Y}}(\mathbf{y}_i, \mathbf{y}_j))^2 \mathbf{P}_{ij} \mathbf{P}_{i'j'}$$

As a result, applying minimization on both sides yields that $\mathrm{GW}_{\varepsilon}(r_n, T \sharp r_n) < \mathrm{GW}_0(r_n, T \sharp r_n) = \mathrm{GW}(r_n, T \sharp r_n)$, and therefore:

$$\mathrm{GMG}_{r_n, \varepsilon}(T) > \mathrm{GMG}_{r_n, 0}(T) = \mathrm{GMG}_{r_n}(T) \geq 0.$$

### B.4    Proofs of Prop. 3.2

**Proposition 3.2.** *If* $\mathrm{GMG}_r(T) = 0$, $\forall s \in \mathcal{P}(\mathcal{X})$ *s.t.* $\mathrm{Spt}(s) \subseteq \mathrm{Spt}(r)$, *one has* $\mathrm{GMG}_s(T) = 0$.

*Proof.* Let $T, r, s$ as described and suppose that $\mathcal{GM}_r^c(T) = 0$. Then, $\pi^r := (\mathrm{Id}, T) \sharp r$ is an optimal Gromov-Wasserstein coupling, solution of Problem (GWP) between $r$ and $T \sharp r$ for costs $c_{\mathcal{X}}$ and $c_{\mathcal{Y}}$. Therefore, from (Sjourn et al., 2023, Theorem. 3), $\pi^r$ is an optimal Kantorvich coupling, solution of Problem (KP) between $r$ and $T \sharp r$ for the linearized cost:

$$\tilde{c} : (\mathbf{x}, \mathbf{y}) \in \mathcal{X} \times \mathcal{Y} \mapsto \int_{\mathcal{X} \times \mathcal{Y}} \tfrac{1}{2} |c_{\mathcal{X}}(\mathbf{x}, \mathbf{x}') - c_{\mathcal{Y}}(\mathbf{y}, \mathbf{y}')|^2 \, \mathrm{d}\pi^r(\mathbf{x}', \mathbf{y}') \tag{14}$$

Additionally, $\mathcal{X} \times \mathcal{Y}$ is a compact set as a product of compact sets, so since $(\mathbf{x}, \mathbf{y}) \mapsto |c_{\mathcal{X}}(\mathbf{x}, \mathbf{x}') - c_{\mathcal{Y}}(\mathbf{y}, \mathbf{y}')|^2$ is continuous as $c_{\mathcal{X}}$ and $c_{\mathcal{Y}}$ are continuous, it is bounded on $\mathcal{X} \times \mathcal{Y}$. Afterward, since $\pi^r$ has finite mass, by Lebesgue's dominated convergence Theorem, it follows that $\tilde{c}$ is continuous, and hence uniformly continuous, again since $\mathcal{X} \times \mathcal{Y}$ is compact.

Afterwards, by virtue of (Santambrogio, 2015, Theorem 1.38), $\mathrm{Spt}(\pi^r)$ is a $\tilde{c}$-cyclically monotone (CM) set (see (Santambrogio, 2015, Definition. 1.36)). From the definition of cyclical monotonicity, this property translates to subsets. Then, by defining $\pi^s = (\mathrm{Id}, T) \sharp s$, as $\mathrm{Spt}(p) \subset \mathrm{Spt}(r)$, one has $\mathrm{Spt}(\pi^s) = \mathrm{Spt}((\mathrm{Id}, T) \sharp s) \subset \mathrm{Spt}((\mathrm{Id}, T) \sharp r) = \mathrm{Spt}(\pi^r)$, so $\mathrm{Spt}(\pi^s)$ is $\tilde{c}$-CM. Finally, since $\mathcal{X}$ and $\mathcal{Y}$ are compact, and $\tilde{c}$ is uniformly continuous, the $\tilde{c}$-cyclical monotonicity of its support implies that the coupling $\pi^p$ is a Kantorovich optimal coupling between its marginals for cost $\tilde{c}$, thanks to (Santambrogio, 2015, Theorem 1.49). By re-applying (Sjourn et al., 2023, Theorem. 3), we get that $\pi^s$ solves the Gromov-Wasserstein problem between its marginals for costs $c_{\mathcal{X}}$ and $c_{\mathcal{Y}}$. In other words, $\pi^s = (\mathrm{Id}, T) \sharp s$ is Gromov-Wasserstein optimal coupling between $s$ and $T \sharp s$ so $T$ is a Gromov-Monge map between $s$ and $T \sharp s$ and $\mathrm{GMG}_s(T) = 0$. $\qquad\square$

### B.5    Proofs of Thm. 3.4

**Theorem 3.4.** *Both* $\mathrm{GMG}_r^2$ *and* $\mathrm{GMG}_r^{\langle \cdot, \cdot \rangle}$, *as well as their finite sample versions, are weakly convex.*

- ***Finite sample.*** *We note* $\mathbf{X} \in \mathbb{R}^{n \times d}$ *the matrix that stores the* $\mathbf{x}_i$, *i.e. the support of* $r_n$, *as rows. Then, (i)* $\mathrm{GMG}_{r_n}^2$ *and (ii)* $\mathrm{GMG}_{r_n}^{\langle \cdot, \cdot \rangle}$ *are respectively (i)* $\gamma_{2,n}$ *and (ii)* $\gamma_{inner,n}$-*weakly convex, where:* $\gamma_{inner,n} = \lambda_{\max}(\tfrac{1}{n} \mathbf{X}\mathbf{X}^{\top}) - \lambda_{\min}(\tfrac{1}{n} \mathbf{X}\mathbf{X}^{\top})$ *and* $\gamma_{2,n} = \gamma_{inner,n} + \max_{i=1...n} \|\mathbf{x}_i\|_2^2$.

- ***Asymptotic.*** *(i)* $\mathrm{GMG}_r^2$ *and (ii)* $\mathrm{GMG}_r^{\langle \cdot, \cdot \rangle}$ *are respectively (i)* $\gamma_2$ *and (ii)* $\gamma_{inner}$-*weakly convex, where:* $\gamma_{inner} = \lambda_{\max}(\mathbb{E}_{\mathbf{x} \sim r}[\mathbf{x}\mathbf{x}^{\top}])$ *and* $\gamma_{2,n} = \gamma_{inner} + \max_{\mathbf{x} \in \mathrm{Spt}(r)} \|\mathbf{x}\|_2^2$.

Before proving Thm. 3.4, we first demonstrate some **technical results** that will be useful later.

**Reformulation of the empirical GMG using permutations.** We start by showing that $\mathrm{GMG}_{r_n}(T)$ is always the sub-optimality gap of $T$ in Prob. (GMP) between $r_n$ and $T\sharp r_n$. This occurs because Prob. (GMP) and Prob. (GWP) coincide when applied between empirical measures on the same number of points. In other words, we can reformulate Prob. (GWP) between $r_n$ and $T\sharp r_n$ using permutation matrices, instead of (plain) couplings.

**Proposition B.1.** *The empirical GMG reads*

$$\mathrm{GMG}_{r_n}(T) = \mathrm{DST}_{r_n}(T) - \min_{\sigma \in \mathcal{S}_n} \frac{1}{n^2} \sum_{i,j=1}^{n} \big(c_{\mathcal{X}}(\mathbf{x}_i, \mathbf{x}_j) - c_{\mathcal{Y}}(T(\mathbf{x}_{\sigma(i)}), T(\mathbf{x}_{\sigma(j)}))\big)^2 \qquad (15)$$

*Proof.* We first show a more general results, stating that when $c_{\mathcal{X}}, c_{\mathcal{Y}}$ are conditionally positive kernels (see A.1), the discrete GW couplings between uniform, empirical distributions supported on the same number of points, ae permutation matrices.

**Proposition B.2** (Equivalence between Gromov-Monge and Gromov-Wasserstein problems in the discrete case.). *Let $p_n = \frac{1}{n}\sum_{i=1}^n \delta_{\mathbf{x}_i}$ and $q_n = \frac{1}{n}\sum_{i=1}^n \delta_{\mathbf{y}_i}$ two uniform, empirical measures, supported on the same number of points. We denote by $P_n = \{\mathbf{P} \in \mathbb{R}^{n \times n}, \exists \sigma \in \mathcal{S}_n, \mathbf{P}_{ij} := \delta_{j,\sigma(i)}\}$ the set set of permutation matrices. Assume that $c_{\mathcal{X}}$ and $c_{\mathcal{Y}}$ (or $-c_{\mathcal{X}}$ and $-c_{\mathcal{Y}}$) are conditionally positive kernels (see A.1). Then, the GM and GW formulations coincide, in the sense that we can restrict the GW problem to permutations, namely*

$$\mathrm{GW}(p_n, p_n) = \min_{\mathbf{P} \in U_n} \sum_{i,j,i',j'=1}^{n} (c_{\mathcal{X}}(\mathbf{x}_i, \mathbf{x}_{i'}) - c_{\mathcal{Y}}(\mathbf{y}_j, \mathbf{y}_{j'}))^2 \mathbf{P}_{ij}\mathbf{P}_{i'j'}$$

$$= \frac{1}{n^2} \min_{\mathbf{P} \in P_n} \sum_{i,j,i',j'=1}^{n} (c_{\mathcal{X}}(\mathbf{x}_i, \mathbf{x}_{i'}) - c_{\mathcal{Y}}(\mathbf{y}_j, \mathbf{y}_{j'}))^2 \mathbf{P}_{ij}\mathbf{P}_{i'j'} \qquad (16)$$

$$= \frac{1}{n^2} \min_{\sigma \in \mathcal{S}_n} \sum_{i,j=1}^{n} (c_{\mathcal{X}}(\mathbf{x}_i, \mathbf{x}_j) - c_{\mathcal{Y}}(\mathbf{y}_{\sigma(i)}, \mathbf{y}_{\sigma(j)}))^2$$

*Proof.* Let $\mathbf{P}^\star \in U_n$ solution of the Gromov-Wasserstein between $p_n$ and $p_n$, i.e.

$$\mathbf{P}^\star \in \arg\min_{\mathbf{P} \in U_n} \sum_{i,j,i',j'=1}^{n} (c_{\mathcal{X}}(\mathbf{x}_i, \mathbf{x}_{i'}) - c_{\mathcal{Y}}(\mathbf{y}_j, \mathbf{y}_{j'}))^2 \mathbf{P}_{ij}\mathbf{P}_{i'j'}$$

that always exists by continuity of the GW objective function on the compact $U_n$. We show that $\mathbf{P}^\star$ can be chosen as a (rescaled) permutation matrix without loss of generality.

As we assume that $c_{\mathcal{X}}$ and $c_{\mathcal{Y}}$ (or $-c_{\mathcal{X}}$ and $-c_{\mathcal{Y}}$) are conditionally positive kernels, from (Sjourn et al., 2023, Theorem. 3), $\mathbf{P}^\star$ also solves:

$$\mathbf{P}^\star \in \arg\min_{\mathbf{Q} \in U_n} \sum_{i,j,i',j'=1}^{n} (c_{\mathcal{X}}(\mathbf{x}_i, \mathbf{x}_{i'}) - c_{\mathcal{Y}}(\mathbf{y}_j, \mathbf{y}_{j'}))^2 \mathbf{P}^\star_{ij}\mathbf{Q}_{i'j'} \qquad (17)$$

We then define the linearized cost matrix $\tilde{C} \in \mathbb{R}^{n \times n}$, s.t.

$$\tilde{\mathbf{C}}_{ij} = \sum_{i',j'=1}^{n} (c_{\mathcal{X}}(\mathbf{x}_i, \mathbf{x}_{i'}) - c_{\mathcal{Y}}(\mathbf{y}_j, \mathbf{y}_{j'}))^2 \mathbf{P}^\star_{ij}$$

which allows us to reformulate Eq. (17) as

$$\mathbf{P}^\star \in \arg\min_{\mathbf{Q} \in U_n} \langle \tilde{\mathbf{C}}, \mathbf{Q} \rangle \qquad (18)$$

Birkhoff's theorem states that the extremal points of $U_n$ are the permutation matrices $P_n$. Moreover, a seminal theorem of linear programming (Bertsimas and Tsitsiklis, 1997, Theorem 2.7) states that the minimum of a linear objective on a bounded polytope, if finite, is reached at an extremal point of the polyhedron. Therefore, as $\mathbf{P}^\star$ solves Eq. (18), it is an extremal point of $U_n$, so it can always be chosen as a permutation matrix. Therefore, the equivalence between GW and GM follows.

$\square$

To conclude the proof of Prop. B.1, we simply remark that $r_n = \frac{1}{n}\sum_{i=1}^{n}\delta_{\mathbf{x}_i}$ and $T\sharp r_n = \frac{1}{n}\sum_{i=1}^{n}\delta_{T(\mathbf{x}_i)}$ are uniform, empirical distribution, and supported on the same number of points.

$\square$

**Consistency of the empirical GMG.** We continue by proving a consistency result for the empirical GMG, which we will later use to deduce the asymptotic weak convexity constant from the finite-sample case.

**Proposition B.3.** *For both $c_{\mathcal{X}} = c_{\mathcal{Y}} = \|\cdot - \cdot\|_2^2$ and $c_{\mathcal{X}} = c_{\mathcal{Y}} = \langle\cdot,\cdot\rangle$, one has $\mathrm{GMG}_{r_n}(T) \to \mathrm{GMG}_r(T)$ almost surely.*

*Proof.* We first note that the empirical estimator of the distortion is consistent, as both costs are continuous, and $\mathcal{X}$ is compact. We then need to study, in both cases, the convergence of $\mathrm{GW}(r_n, T\sharp r_n)$ to $\mathrm{GW}(r_n, T\sharp r)$.

To that end, we first remark that as, almost surely, $r_n \to r$ in distribution, one also has that, almost surely, $T\sharp r_n \to T\sharp r$ in distribution. Indeed, since $\mathcal{Y}$ is compact, $T$ is bounded so for any bounded and continuous $f : \mathcal{Y} \to \mathbb{R}$ and $X \sim r$, $f \circ T(X)$ is well defined and bounded so integrable. Afterwards, one can simply adapt the proof of the almost sure weak convergence of empirical measure based on the strong law of large numbers to show that, almost surely, $T\sharp r_n \to T\sharp r$ in distribution. See for instance (Le Gall, Theorem 10.4.1).

We start with the squared Euclidean distance. As, almost surely, both $r_n \to r$ and $T\sharp r_n \to T\sharp r$ in distribution, the results follows from (Mémoli, 2011, Thm 5.1, (e)).

We continue with the inner product. As noticed by Rioux et al. (2023, Lemma 2)in the first version of the paper the GW for inner product costs can be reformulated as:

$$\mathrm{GW}^{\langle\cdot,\cdot\rangle}(p,q) = \int_{\mathcal{X}\times\mathcal{X}} \langle\mathbf{x},\mathbf{x}'\rangle\,\mathrm{d}p(\mathbf{x})\,\mathrm{d}p(\mathbf{x}') + \int_{\mathcal{Y}\times\mathcal{Y}} \langle\mathbf{y},\mathbf{y}'\rangle\,\mathrm{d}q(\mathbf{y})\,\mathrm{d}q(\mathbf{y}')$$
$$+ \min_{\mathbf{M}\in\mathcal{M}}\min_{\pi\in\Pi(p,q)}\int_{\mathcal{X}\times\mathcal{Y}} -4\langle\mathbf{M}\mathbf{x},\mathbf{y}\rangle\,\mathrm{d}\pi(\mathbf{x},\mathbf{y}) + 4\|\mathbf{M}\|_2^2, \tag{19}$$

where we define $\mathcal{M} = [-M/2, M/2]^{d_{\mathcal{X}}\times d_{\mathcal{Y}}}$ with $M = \sqrt{\int_{\mathcal{X}}\|\mathbf{x}\|_2^2\,\mathrm{d}p(\mathbf{x})\int_{\mathcal{Y}}\|\mathbf{y}\|_2^2\,\mathrm{d}q(\mathbf{y})}$. In particular, they show this result for the entropic GW problem with $\varepsilon > 0$, but their proof is also valid for $\varepsilon = 0$. The above terms only involving the marginal, i.e., not involved in the minimization, are naturally stable under convergence in distribution, as $\mathcal{X}$ and $\mathcal{Y}$ are compact, so as $\mathcal{X}\times\mathcal{X}$ and $\mathcal{Y}\times\mathcal{Y}$. As a result, we only need to study the stability of this quantity under the convergence in distribution of the following functional:

$$\mathcal{F}(p,q) = \min_{\mathbf{M}\in\mathcal{M}}\min_{\pi\in\Pi(p,q)}\int_{\mathcal{X}\times\mathcal{Y}} -4\langle\mathbf{M}\mathbf{x},\mathbf{y}\rangle\,\mathrm{d}\pi(\mathbf{x},\mathbf{y}) + 4\|\mathbf{M}\|_2^2, \tag{20}$$

We first remark that:

$$|\mathcal{F}(p,q) - \mathcal{F}(p_n,q_n)|$$
$$\leq \sup_{M\in\mathcal{M}}|\min_{\pi\in\Pi(p,q)}\int_{\mathcal{X}\times\mathcal{Y}} -4\langle\mathbf{M}\mathbf{x},\mathbf{y}\rangle\,\mathrm{d}\pi(\mathbf{x},\mathbf{y}) - \min_{\pi\in\Pi(p,q)}\int_{\mathcal{X}\times\mathcal{Y}} -4\langle\mathbf{M}\mathbf{x},\mathbf{y}\rangle\,\mathrm{d}\pi(\mathbf{x},\mathbf{y})|$$
$$\leq \sup_{M\in\mathcal{M}}|\min_{\pi\in\Pi(p,q)}\int_{\mathcal{X}\times\mathcal{Y}} 2\|\mathbf{M}\mathbf{x} - \mathbf{y}\|_2^2\,\mathrm{d}\pi(\mathbf{x},\mathbf{y}) - \min_{\pi\in\Pi(p_n,q_n)}\int_{\mathcal{X}\times\mathcal{Y}} 2\|\mathbf{M}\mathbf{x} - \mathbf{y}\|_2^2\,\mathrm{d}\pi(\mathbf{x},\mathbf{y})2|$$
$$+2\cdot\sup_{M\in\mathcal{M}}|\int_{\mathcal{X}}\|\mathbf{M}\mathbf{x}\|_2^2\,\mathrm{d}p(\mathbf{x}) - \int_{\mathcal{X}}\|\mathbf{M}\mathbf{x}\|_2^2\,\mathrm{d}p_n(\mathbf{x})|$$
$$+2\cdot|\int_{\mathcal{Y}}\|\mathbf{y}\|_2^2\,\mathrm{d}q(\mathbf{y}) - \int_{\mathcal{Y}}\|\mathbf{y}\|_2^2\,\mathrm{d}q_n(\mathbf{y})|$$

$$\tag{21}$$

Then, we show the convergence of each term separately.

- For the first term, we remark that (up to a constant factor) it can be reformulated:

$$\sup_{M \in \mathcal{M}} |W_2^2(\mathbf{M}\sharp p, q) - W_2^2(\mathbf{M}\sharp p_n, q_n)|$$

where we remind that that $W_2^2$ is the (squared) Wasserstein distance, solution of Eq. (KP) induced by $c(\mathbf{x}, \mathbf{y}) = \|\mathbf{x} - \mathbf{y}\|_2^2$. By virtue of (Manole and Niles-Weed, 2024, Theorem 2), there exists a constant $C > 0$, s.t. we can uniformly bound

$$\sup_{M \in \mathcal{M}} |W_2^2(\mathbf{M}\sharp p, q) - W_2^2(\mathbf{M}\sharp p_n, q_n)| \leq Cn^{-1/d}$$

and the convergence follows.

- For the second one, this follows from from the convergence in distribution of $p_n$ to $p$ along with the Ascoli-Arzela theorem, since both $\mathcal{M}$ and $\mathcal{X}$ are compact sets, so the $\{f_\mathbf{M} \mid f_\mathbf{M} : \mathbf{x} \mapsto \|\mathbf{M}\mathbf{x}\|_2^2\}$ are uniformly bounded and equi-continuous.

- For the third one, this follows from the convergence in distribution of $q_n$ to $q$.

As a result, we finally get $\mathrm{GW}^{\langle \cdot, \cdot \rangle}(p_n, q_n) \to \mathrm{GW}^{\langle \cdot, \cdot \rangle}(p, q)$.

$\square$

**Weak convexity.**    Finally, we demonstrate some useful results on weakly convex functions on $\mathbb{R}^d$.

**Definition B.4.**  A function $f : \mathbb{R}^d \to \mathbb{R}$ is $\gamma$-weakly convex if $f + \gamma\| \cdot \|_2^2$ is convex.

From the definition, we see that if $f$ is $\gamma$-weakly convex, than $f$ is also $\gamma'$ weakly convex for any $\gamma' \geq \gamma$. This naturally extends to weakly convex functionals $\mathcal{F}$ on $L_2(r)$.

**Lemma B.5.**  *Let $\mathbf{A} \in S_d(\mathbb{R})$ a symmetric matrix and define the quadratic form $f_\mathbf{A} : \mathbf{x} \in \mathbb{R}^d \mapsto \mathbf{x}^\top \mathbf{A}\mathbf{x}$. Then, $f_\mathbf{A}$ is $\max(0, -\lambda_{\min}(\mathbf{A}))$-weakly convex.*

*Proof.*  We use the fact that a twice continuously differentiable function is convex i.f.f. its hessian is positive semi-definite (Boyd and Vandenberghe, 2004, §(3.1.4)). Therefore, $f_\mathbf{A}$ is convex i.f.f. $\nabla^2 f_\mathbf{A} = \mathbf{A} \geq 0$. If $\lambda_{\min}(\mathbf{A}) \geq 0$, then $\mathbf{A} \geq 0$ so $f_\mathbf{A}$ is convex, i.e. 0-weakly convex. Otherwise, $f_\mathbf{A} - \frac{1}{2}\lambda_{\min}(\mathbf{A})\| \cdot \|_2^2$ has hessian $A - \lambda_{\min}(\mathbf{A}) \geq 0$, so it is convex, which yields that $f_\mathbf{A}$ is $-\lambda_{\min}(\mathbf{A})$-weakly convex. $\square$

**Lemma B.6.**  *Let $(f_i)_{i \in I}$ a family of $\gamma$-weakly convex functions, with potentially infinite $I$. Then, $f : \mathbf{x} \in \mathbb{R}^d \mapsto \sup_{i \in I} f_i(\mathbf{x})$ is $\gamma$-weakly convex.*

*Proof.*  As the $f_i$ are $\gamma$-weakly convex, $f_i + \frac{1}{2}\gamma$ is convex, so $\mathbf{x} \mapsto \sup_{i \in I} f_i(\mathbf{x}) + \frac{1}{2}\gamma\|x\|_2^2 = (\sup_{i \in I} f_i(\mathbf{x})) + \frac{1}{2}\gamma\|x\|_2^2$ is convex (Boyd and Vandenberghe, 2004, Eq. (3.7)). Therefore, the $\gamma$-weak convexity of $f$ follows $\square$

Lets now proceed to prove the main Thm. (3.4).

*Proof of Thm. (3.4).*  **Finite sample**. We first study the weak convexity of $\mathcal{GM}_{r_n}^{\langle \cdot, \cdot \rangle}$, i.e. the Gromov-Monge gap for the inner product. For a map $T \in L_2(r)$, it reads

$$\mathrm{GMG}_{r_n}^{\langle \cdot, \cdot \rangle}(T) = \frac{1}{n^2} \sum_{i,j=1}^{n} \frac{1}{2}|\langle \mathbf{x}_i, \mathbf{x}_j \rangle - \langle T(\mathbf{x}_i), T(\mathbf{x}_j) \rangle|^2$$

$$- \min_{\mathbf{P} \in U_n} \sum_{i,j,i',j'=1}^{n} \frac{1}{2}|\langle \mathbf{x}_i, \mathbf{x}_{i'} \rangle - \langle T(\mathbf{x}_j), T(\mathbf{x}_{j'}) \rangle|^2 \mathbf{P}_{ij}\mathbf{P}_{i'j'}$$

As $r_n$ and $T\sharp r_n$ are uniform empirical supported on the same number of points, using Prop. B.2, we can reformulate the RHS with permutation matrices, which yields

$$\text{GMG}_{r_n}^{\langle \cdot, \cdot \rangle}(T) = \frac{1}{n^2} \sum_{i,j=1}^{n} \frac{1}{2} |\langle \mathbf{x}_i, \mathbf{x}_j \rangle - \langle T(\mathbf{x}_i), T(\mathbf{x}_j) \rangle|^2$$

$$- \frac{1}{n^2} \min_{\mathbf{P} \in P_n} \sum_{i,j,i',j'=1}^{n} \frac{1}{2} |\langle \mathbf{x}_i, \mathbf{x}_{i'} \rangle - \langle T(\mathbf{x}_j), T(\mathbf{x}_{j'}) \rangle|^2 \mathbf{P}_{ij} \mathbf{P}_{i'j'}$$

From this expression, $\text{GMG}_{r_n}^{\langle \cdot, \cdot \rangle}$ can be reformulated as a matrix input function. Indeed, it only depends on the map $T$ via its values on the support of $r_n$, namely $\mathbf{x}_1, ..., \mathbf{x}_n$. Therefore, we write $\mathbf{t}_i := T(\mathbf{x}_i)$, and define $\mathbf{X}, \mathbf{T} \in \mathbb{R}^{n \times d}$ which contain observations $\mathbf{x}_i$ and $\mathbf{t}_i$ respectively, stored as rows. Then, studying $\text{GMG}_{r_n}^{\langle \cdot, \cdot \rangle}$ remains to study

$$f(\mathbf{T}) := \frac{1}{n^2} \sum_{i,j=1}^{n} \frac{1}{2} |\langle \mathbf{x}_i, \mathbf{x}_j \rangle - \langle \mathbf{t}_i, \mathbf{t}_j \rangle|^2 - \frac{1}{n^2} \min_{\mathbf{P} \in P_n} \sum_{i,j,i',j'=1}^{n} \frac{1}{2} |\langle \mathbf{x}_i, \mathbf{x}_{i'} \rangle - \langle \mathbf{t}_j, \mathbf{t}_{j'} \rangle|^2 \mathbf{P}_{ij} \mathbf{P}_{i'j'}$$

By developing each term and exploiting that for any $\mathbf{P} \in P_n$, $\mathbf{P} 1_n = \mathbf{P}^\top 1_n = \frac{1}{n} 1_n$, we derive

$$f(\mathbf{T}) = \frac{1}{n^2} \sum_{i,j=1}^{n} -\langle \mathbf{x}_i, \mathbf{x}_j \rangle \cdot \langle \mathbf{t}_i, \mathbf{t}_j \rangle - \min_{\mathbf{P} \in P_n} \frac{1}{n^2} \sum_{i,j,i',j'=1}^{n} -\langle \mathbf{x}_i, \mathbf{x}_{i'} \rangle \cdot \langle \mathbf{t}_j, \mathbf{t}_{j'} \rangle \mathbf{P}_{ij} \mathbf{P}_{i'j'}$$

$$= \max_{\mathbf{P} \in P_n} \frac{1}{n^2} \sum_{i,j,i',j'=1}^{n} \langle \mathbf{x}_i, \mathbf{x}_{i'} \rangle \cdot \langle \mathbf{t}_j, \mathbf{t}_{j'} \rangle \mathbf{P}_{ij} \mathbf{P}_{i'j'} - \frac{1}{n^2} \sum_{i,j=1}^{n} \langle \mathbf{x}_i, \mathbf{x}_j \rangle \cdot \langle \mathbf{t}_i, \mathbf{t}_j \rangle$$

$$= \max_{\mathbf{P} \in P_n} \langle \frac{1}{n^2} \mathbf{P}^\top \mathbf{X} \mathbf{X}^\top \mathbf{P}, \mathbf{T} \mathbf{T}^\top \rangle - \langle \frac{1}{n^2} \mathbf{X} \mathbf{X}^\top, \mathbf{T} \mathbf{T}^\top \rangle$$

$$= \max_{\mathbf{P} \in P_n} \langle \frac{1}{n^2} (\mathbf{P}^\top \mathbf{X} \mathbf{X}^\top \mathbf{P} - \mathbf{X} \mathbf{X}^\top), \mathbf{T} \mathbf{T}^\top \rangle$$

$$= \max_{\mathbf{P} \in P_n} \langle \frac{1}{n^2} (\mathbf{P}^\top \mathbf{X} \mathbf{X}^\top \mathbf{P} - \mathbf{X} \mathbf{X}^\top) \mathbf{T}, \mathbf{T} \rangle$$

$$= \max_{\mathbf{P} \in P_n} \langle \mathbf{A}_{\mathbf{X}, \mathbf{P}} \mathbf{T}, \mathbf{T} \rangle$$

where we define $\mathbf{A}_{\mathbf{X}, \mathbf{P}} := \frac{1}{n^2} (\mathbf{P}^\top \mathbf{X} \mathbf{X}^\top \mathbf{P} - \mathbf{X} \mathbf{X}^\top) \in \mathbb{R}^{n \times n}$. To study the convexity of this matrix input function, we vectorize it. From (Petersen and Pedersen, 2008, Eq. (520)), we note that, for any $\mathbf{M} \in \mathbb{R}^{n \times n}$

$$\langle \mathbf{M} \mathbf{T}, \mathbf{T} \rangle = \mathbf{vec}(\mathbf{T})^\top \mathbf{vec}(\mathbf{M} \mathbf{T}) = \mathbf{vec}(\mathbf{T})^\top (\mathbf{M} \otimes I_n) \mathbf{vec}(\mathbf{T})$$

where $\mathbf{vec}$ is the vectorization operator, raveling a matrix along its rows, and $\otimes$ is the Kronecker product. Applying this identity, we reformulate:

$$f(\mathbf{T}) = \max_{\mathbf{P} \in U_n} \mathbf{vec}(\mathbf{T})^\top (\mathbf{A}_{\mathbf{X}, \mathbf{P}} \otimes I_n) \mathbf{vec}(\mathbf{T}) \tag{22}$$

To study the convexity of $r$, we study the convexity of each $r_{\mathbf{A}_{\mathbf{X}, \mathbf{P}}}(\mathbf{T}) := \mathbf{vec}(\mathbf{T})^\top (\mathbf{A}_{\mathbf{X}, \mathbf{P}} \otimes I_n) \mathbf{vec}(\mathbf{T})$, which are quadratic forms induced by the $\mathbf{A}_{\mathbf{X}, \mathbf{P}} \otimes I_n$. This remains to study the (semi-) positive definiteness of the matrices $\mathbf{A}_{\mathbf{X}, \mathbf{P}} \otimes I_n$. As each $\mathbf{A}_{\mathbf{X}, \mathbf{P}} \in \mathbb{R}^{n \times n}$ is symmetric and square, $\mathbf{A}_{\mathbf{X}, \mathbf{P}} \otimes I_n$ is also symmetric and from (Petersen and Pedersen, 2008, Eq. (519)) its eigenvalues are the outer products of the eigenvalues of $\mathbf{A}_{\mathbf{X}, \mathbf{P}}$ and $I_n$, namely

$$\mathbf{eig}(\mathbf{A}_{\mathbf{X}, \mathbf{P}} \otimes I_n) = \{\lambda_i(\mathbf{A}_{\mathbf{X}, \mathbf{P}}) \cdot \lambda_j(I_n)\}_{1 \leq i,j \leq n}$$

$$= \{\underbrace{\lambda_1(\mathbf{A}_{\mathbf{X}, \mathbf{P}}), \ldots, \lambda_1(\mathbf{A}_{\mathbf{X}, \mathbf{P}})}_{n \text{ times}}, \ldots, \underbrace{\lambda_n(\mathbf{A}_{\mathbf{X}, \mathbf{P}}), \ldots, \lambda_n(\mathbf{A}_{\mathbf{X}, \mathbf{P}})}_{n \text{ times}}\} \tag{23}$$

It follows that the minimal eigenvalue of $\mathbf{A}_{\mathbf{X}, \mathbf{P}} \otimes I_n$ is $\lambda_{\min}(\mathbf{A}_{\mathbf{X}, \mathbf{P}} \otimes I_n) = \lambda_{\min}(\mathbf{A}_{\mathbf{X}, \mathbf{P}})$. Utilizing the expression of $\mathbf{A}_{\mathbf{X}, \mathbf{P}}$

$$\lambda_{\min}(\mathbf{A}_{\mathbf{X}, \mathbf{P}}) = \frac{1}{n^2} \lambda_{\min}(\mathbf{P}^\top \mathbf{X} \mathbf{X}^\top \mathbf{P} - \mathbf{X} \mathbf{X}^\top)$$

$$\geq \frac{1}{n^2} (\lambda_{\min}(\mathbf{P}^\top \mathbf{X} \mathbf{X}^\top \mathbf{P}) + \lambda_{\min}(-\mathbf{X} \mathbf{X}^\top)) \tag{24}$$

$$= \frac{1}{n^2} (\lambda_{\min}(\mathbf{P}^\top \mathbf{X} \mathbf{X}^\top \mathbf{P}) - \lambda_{\max}(\mathbf{X} \mathbf{X}^\top))$$

Reminding that $\mathbf{P} \in U_n$, one has $\mathbf{P}^\top = \mathbf{P}^{-1}$, so $\mathbf{P}^\top \mathbf{X} \mathbf{X}^\top$ and $\mathbf{X} \mathbf{X}^\top$ are similar, and they have the same eigenvalues. In particular $\lambda_{\min}(\mathbf{P}^\top \mathbf{X} \mathbf{X}^\top \mathbf{P}) = \lambda_{\min}(\mathbf{X} \mathbf{X}^\top)$. Combining these results, it follows that

$$\lambda_{\min}(\mathbf{A}_{\mathbf{X},\mathbf{P}} \otimes I_n) = \lambda_{\min}(\mathbf{A}_{\mathbf{X},\mathbf{P}}) \geq \tfrac{1}{n^2}(\lambda_{\min}(\mathbf{X}\mathbf{X}^\top) - \lambda_{\max}(\mathbf{X}\mathbf{X}^\top)) \tag{25}$$

We then remind that each $r_{\mathbf{A}_{\mathbf{X},\mathbf{P}}}$ is the quadratic form defined by $\mathbf{A}_{\mathbf{X},\mathbf{P}} \otimes I_n$, so by applying Prop. B.5, it is $\mathbf{A}_{\mathbf{X},\mathbf{P}} \otimes I_n$-weakly convex, and hence $\tfrac{1}{n^2}(\lambda_{\max}(\mathbf{X}\mathbf{X}^\top) - \lambda_{\min}(\mathbf{X}\mathbf{X}^\top))$-weakly convex. Therefore, applying Prop. (B.6), $r$ is $\tfrac{1}{n^2}(\lambda_{\max}(\mathbf{X}\mathbf{X}^\top) - \lambda_{\min}(\mathbf{X}\mathbf{X}^\top))$-weakly convex, in $\mathbb{R}^d$. Reminding that $\gamma_{\text{inner}} = \tfrac{1}{n}(\lambda_{\max}(\mathbf{X}\mathbf{X}^\top) - \lambda_{\min}(\mathbf{X}\mathbf{X}^\top))$, $r$ is $\tfrac{1}{n}\gamma_{\text{inner}}$ weakly convex. This implies that $\mathbf{T} \mapsto f(\mathbf{T}) + \tfrac{1}{n}\gamma_{\text{inner}}\|\mathbf{T}\|_2^2$ is convex. By reminding that $\mathbf{T}$ stores the $T(\mathbf{x}_i)$ as rows, $\tfrac{1}{n}\|\mathbf{T}\|_2^2 = \|T\|_{L_2(r_n)}$. Consequently, $\text{GMG}_{r_n}^{\langle\cdot,\cdot\rangle}$ is $\gamma_{\text{inner}}$ in $L_2(r_n)$.

We then study the convexity of $\text{GMG}_{r_n}^2$. We follow exactly the same approach. One has:

$$\text{GMG}_{r_n}^2(T) = \tfrac{1}{n^2} \sum_{i,j=1}^{n} \tfrac{1}{2}|\|\mathbf{x}_i - \mathbf{x}_j\|_2^2 - \|T(\mathbf{x}_i) - T(\mathbf{x}_j)\|_2^2|^2$$

$$- \tfrac{1}{n^2} \min_{\mathbf{P} \in P_n} \sum_{i,j,i',j'=1}^{n} \tfrac{1}{2}|\|\mathbf{x}_i - \mathbf{x}_j\|_2^2 - \|T(\mathbf{x}_i) - T(\mathbf{x}_j)\|_2^2|^2|^2 \mathbf{P}_{ij}\mathbf{P}_{i'j'}$$

Similarly, studying the convexity of $\text{GMG}_{r_n}^2(T)$ remains to study the convexity of the matrix input function:

$$g(\mathbf{T}) := \tfrac{1}{n^2} \sum_{i,j=1}^{n} \tfrac{1}{2}|\|\mathbf{x}_i - \mathbf{x}_j\|_2^2 - \|\mathbf{t}_i - \mathbf{t}_j\|_2^2|^2$$

$$- \tfrac{1}{n^2} \min_{\mathbf{P} \in P_n} \sum_{i,j,i',j'=1}^{n} \tfrac{1}{2}|\|\mathbf{x}_i - \mathbf{x}_j\|_2^2 - \|\mathbf{t}_i - \mathbf{t}_j\|_2^2|^2 \mathbf{P}_{ij}\mathbf{P}_{i'j'}$$

As before, by developing each term, one has:

$$g(\mathbf{T}) = \max_{\mathbf{P} \in P_n} \tfrac{1}{n^2} \sum_{i,j,i',j'=1}^{n} \langle \mathbf{x}_i, \mathbf{x}_{i'} \rangle \cdot \langle \mathbf{t}_j, \mathbf{t}_{j'} \rangle \mathbf{P}_{ij}\mathbf{P}_{i'j'} + \tfrac{1}{2n} \sum_{i,j=1}^{n} \mathbf{P}_{ij}\|\mathbf{x}_i\|_2^2\|\mathbf{t}_i\|_2^2$$

$$- \left( \tfrac{1}{n^2} \sum_{i,j=1}^{n} \langle \mathbf{x}_i, \mathbf{x}_j \rangle \cdot \langle \mathbf{t}_i, \mathbf{t}_j \rangle + \tfrac{1}{2n} \sum_{i,j=1}^{n} \|\mathbf{x}_i\|_2^2\|\mathbf{t}_i\|_2^2 \right)$$

The quadratic terms in $\mathbf{P}$ can be factorized as before using $\mathbf{A}_{\mathbf{X},\mathbf{P}}$. For the new terms w.r.t. the inner product case, we introduce $\mathbf{D}_{\mathbf{X}} := \text{diag}(\|\mathbf{x}_1\|_2^2, \ldots, \|\mathbf{x}_n\|_2^2)$, and remark that we can rewrite:

$$\tfrac{1}{2n} \sum_{i,j=1}^{n} \mathbf{P}_{ij}\|\mathbf{x}_i\|_2^2\|\mathbf{t}_i\|_2^2 - \tfrac{1}{2n} \sum_{i,j=1}^{n} \|\mathbf{x}_i\|_2^2\|\mathbf{t}_i\|_2^2 = \mathbf{vec}(T)^\top \left( \tfrac{1}{2n}(\mathbf{P}^\top - I_n) \otimes \mathbf{D}_{\mathbf{X}} \right) \mathbf{vec}(T)$$

As we can always symetrize the matrix when considering its associated quadratic form, we have:

$$\tfrac{1}{2n} \sum_{i,j=1}^{n} \mathbf{P}_{ij}\|\mathbf{x}_i\|_2^2\|\mathbf{t}_i\|_2^2 - \tfrac{1}{2n} \sum_{i,j=1}^{n} \|\mathbf{x}_i\|_2^2\|\mathbf{t}_i\|_2^2 = \mathbf{vec}(T)^\top \left( \tfrac{1}{2}(\tfrac{1}{2n}(\mathbf{P}^\top + \mathbf{P}) - I_n) \otimes \mathbf{D}_{\mathbf{X}} \right) \mathbf{vec}(T)$$

As a result, we denote $\mathbf{B}_{\mathbf{X},\mathbf{P}} = \tfrac{1}{n}(\tfrac{1}{2}(\mathbf{P}^\top + \mathbf{P}) - I_n) \otimes \mathbf{D}_{\mathbf{X}}$ and finally get:

$$g(\mathbf{T}) = \max_{\mathbf{P} \in P_n} \mathbf{vec}(T)^\top (\mathbf{A}_{\mathbf{X},\mathbf{P}} \otimes I_n + \mathbf{B}_{\mathbf{X},\mathbf{P}}) \mathbf{vec}(T)$$

As we did for $f$, studying the weak convexity of $f$ remains to lower bound the minimal eigenvalue of $\mathbf{A}_{\mathbf{X},\mathbf{P}} \otimes I_n + \mathbf{B}_{\mathbf{X},\mathbf{P}}$. First, one remark that:

$$\lambda_{\min}(\mathbf{A}_{\mathbf{X},\mathbf{P}} \otimes I_n + \mathbf{B}_{\mathbf{X},\mathbf{P}}) \geq \lambda_{\min}(\mathbf{A}_{\mathbf{X},\mathbf{P}} \otimes I_n) + \lambda_{\min}(\mathbf{B}_{\mathbf{X},\mathbf{P}})$$

As we we have already lower bounded $\lambda_{\min}(\mathbf{A}_{\mathbf{X},\mathbf{P}} \otimes I_n) \geq \frac{1}{n^2}(\lambda_{\min}(\mathbf{X}\mathbf{X}^\top) - \lambda_{\max}(\mathbf{X}\mathbf{X}^\top))$, we focus on the RHS. Similarly, one has:

$$\begin{aligned}
\lambda_{\min}(\mathbf{B}_{\mathbf{X},\mathbf{P}}) &= \lambda_{\min}\left(\frac{1}{2n}(\frac{1}{2}(\mathbf{P}^\top + \mathbf{P}) - I_n) \otimes \mathbf{D}_{\mathbf{X}}\right) \\
&\geq \lambda_{\min}\left(\frac{1}{4n}(\mathbf{P}^\top + \mathbf{P}) \otimes \mathbf{D}_{\mathbf{X}}\right) + \lambda_{\min}\left(-\frac{1}{2n}I_n \otimes \mathbf{D}_{\mathbf{X}}\right) \\
&\geq \lambda_{\min}\left(\frac{1}{4n}(\mathbf{P}^\top + \mathbf{P}) \otimes \mathbf{D}_{\mathbf{X}}\right) - \lambda_{\max}\left(\frac{1}{2n}I_n \otimes \mathbf{D}_{\mathbf{X}}\right)
\end{aligned} \tag{26}$$

For both terms, we apply again (Petersen and Pedersen, 2008, Eq. (519)). For the LHS, one has:

$$\mathbf{eig}\left(\frac{1}{4n}(\mathbf{P}^\top + \mathbf{P}) \otimes \mathbf{D}_{\mathbf{X}}\right) = \{\lambda_i(\frac{1}{4n}(\mathbf{P}^\top + \mathbf{P}))\lambda_j(\mathbf{D}_{\mathbf{X}})\}_{1 \leq i,j \leq n} \tag{27}$$

We remark that $\frac{1}{2}(\mathbf{P}^\top + \mathbf{P})$ is a symetric bi-stochastic matrix, so $\lambda_{\min}(\frac{1}{2}(\mathbf{P}^\top + \mathbf{P})) \geq -1$. Therefore, $\lambda_{\min}(\frac{1}{4n}(\mathbf{P}^\top + \mathbf{P})) \geq -\frac{1}{2n}$. As a result, since the eigenvalues of $\mathbf{D}_{\mathbf{X}}$ are the $\|\mathbf{x}_i\|_2^2$, this yields:

$$\lambda_{\min}\left(\frac{1}{4n}(\mathbf{P}^\top + \mathbf{P}) \otimes \mathbf{D}_{\mathbf{X}}\right) \geq -\frac{1}{2n} \max_{i=1,\dots,n} \|\mathbf{x}_i\|_2^2$$

Similarly, we have:

$$-\lambda_{\max}\left(\frac{1}{2n}I_n \otimes \mathbf{D}_{\mathbf{X}}\right) \geq -\frac{1}{2n} \max_{i=1,\dots,n} \|\mathbf{x}_i\|_2^2$$

from which we deduce that:

$$\lambda_{\min}(\mathbf{B}_{\mathbf{X},\mathbf{P}}) \geq -\frac{1}{n} \max_{i=1,\dots,n} \|\mathbf{x}_i\|_2^2$$

We can then lower bound:

$$\begin{aligned}
\lambda_{\min}(\mathbf{A}_{\mathbf{X},\mathbf{P}} \otimes I_n + \mathbf{B}_{\mathbf{X},\mathbf{P}}) &\geq \frac{1}{n^2}(\lambda_{\min}(\mathbf{X}\mathbf{X}^\top) - \lambda_{\max}(\mathbf{X}\mathbf{X}^\top)) - \frac{1}{n} \max_{i=1,\dots,n} \|\mathbf{x}_i\|_2^2 \\
&= -\frac{1}{n}\gamma_{2,n}
\end{aligned} \tag{28}$$

which yields the $\frac{1}{n}\gamma_{2,n}$-weak convexity of $g$, and finally the $\gamma_{2,n}$-weak convexity of $\mathcal{GM}_{r_n}^2$.

**Asymptotic**. For any $T$, we note that, almost surely, $\|T\|_{L_2(r_n)}^2 \to \|T\|_{L_2(r)}^2$. As a result, since convexity is preserved under pointwise convergence and by virtue of Prop. (B.3), we study the (almost sure) convergence of $\gamma_{\text{inner},n}$ and $\gamma_{2,n}$.

We start by $\gamma_{\text{inner},n}$. We first remark that $\lambda_{\max}(\frac{1}{n}\mathbf{X}\mathbf{X}^\top) = \lambda_{\max}(\frac{1}{n}\mathbf{X}^\top\mathbf{X})$. Moreover, as $\mathbf{A} \in S_d^+(\mathbb{R}) \mapsto \lambda_{\max}(\mathbf{A})$ is continuous and $\frac{1}{n}\mathbf{X}^\top\mathbf{X} \to \mathbb{E}_{\mathbf{x}\sim r}[\mathbf{x}\mathbf{x}^\top]$ almost surely, one has $\lambda_{\max}(\frac{1}{n}\mathbf{X}\mathbf{X}^\top) \to \lambda_{\max}(\mathbb{E}_{\mathbf{x}\sim r}[\mathbf{x}\mathbf{x}^\top])$ almost surely. Moreover, for any $n > d$, $\lambda_{\min}(\frac{1}{n}\mathbf{X}\mathbf{X}^\top) = 0$. As a result, $\gamma_{\text{inner},n} \to \lambda_{\max}(\mathbb{E}_{\mathbf{x}\sim r}[\mathbf{x}\mathbf{x}^\top])$ almost surely, which provides the desired asymptotic result.

We continue with $\gamma_{2,n}$. We first remark that $\max_{i=1,\dots,n} \|\mathbf{x}_i\|_2^2 \leq \sup_{\mathbf{x}\in\text{Spt}(r)} \|\mathbf{x}\|_2^2$. As a result, by defining $\tilde{\gamma}_{2,n} = \gamma_{\text{inner},n} + \max_{\mathbf{x}\in\text{Spt}(r)} \|\mathbf{x}\|_2^2$, $\text{GMG}_{r_n}^2$ is also $\tilde{\gamma}_{2,n}$-weakly convex. Moreover, $\max_{\mathbf{x}\in\text{Spt}(r)} \|\mathbf{x}\|_2^2$ does not depends on $n$, $\tilde{\gamma}_{2,n} \to \lambda_{\max}(\mathbb{E}_{\mathbf{x}\sim r}[\mathbf{x}\mathbf{x}^\top]) + \max_{\mathbf{x}\in\text{Spt}(r)} \|\mathbf{x}\|_2^2$ almost surely, which also provides the desired asymptotic result.

$\square$

## C  EXPERIMENTAL DETAILS

All our experiments build on `python` and the `jax`-framework (Babuschkin et al., 2020), alongside `ott-jax` for optimal transport utilities.

Table 3: Hyperparameter grid searches for different baseline and proposed methods.

| Method | Parameter | Values |
|--------|-----------|--------|
| $\beta$-VAE | $\beta$ | [2, 4, 6, 8, 10, 16] |
| $\beta$-TCVAE | $\beta$ | [2, 4, 6, 8, 10, 16] |
| + HFS | $\gamma$ | [1, 10] |
| + DST | $\lambda$ | [0.1, 1, 5, 10, 20] |
| + GMG | $\lambda$ | [0.1, 1, 5, 10, 20] |
| + Jac | $\lambda$ | [0.1, 1, 5, 10, 20] |

## C.1 DETAILS ON DISENTANGLEMENT BENCHMARK

To effectively conduct comprehensive and representative research on disentangled representation learning, we convert the public PyTorch framework proposed in Roth et al. (2023) to an equivalent `jax` variant. We verify our implementation through replications of baseline and HFS results in Roth et al. (2023), mainting relative performance orderings and close absolute disentanglement scores (as measured using DCI-D, whose implementation directly follows from Locatello et al. (2019a) and leverages gradient boosted tree implementations from `scikit-learn`).

For exact and fair comparison, we utilize standard hyperparamater choices from Roth et al. (2023) (which leverages hyerparameters directly from Locatello et al. (2019a), Locatello et al. (2020) and `https://github.com/google-research/disentanglement_lib`). Consequently, the base VAE architecture utilized across all experiment is the same as the one utilized in Roth et al. (2023) and Locatello et al. (2020): With image input sizes of $64 \times 64 \times N_c$ (with $N_c$ the number of input image channels, usually 3). The latent dimensionality, if not otherwise specified, is set to 10. The exact VAE model architecture is as follows:

- **Encoder**: [conv$(32, 4 \times 4$, stride 2) + ReLU] $\times 2$, [conv$(64, 4 \times 4$, stride 2) + ReLU] $\times 2$, MLP(256), MLP($2 \times 10$)

- **Decoder**: MLP(256), [upconv$(64, 4 \times 4$, stride 2) + ReLU] $\times 2$, [upconv$(32, 4 \times 4$, stride 2) + ReLU], [upconv$(n_c, 4 \times 4$, stride 2) + ReLU]

Similar, we retain all training hyperparameters from (Roth et al., 2023) and (Locatello et al., 2020): Using an Adam optimizer ((Kingma and Ba, 2014), $\beta_1 = 0.9, \beta_2 = 0.999, \epsilon = 10^{-8}$) and a learning rate of $10^{-4}$. Following Locatello et al. (2020); Roth et al. (2023) we utilize a batch-size of 64, for which we also ablate all baseline methods. The total number of training steps is set to 300000.

As commonly done for this setting (Locatello et al., 2019a; 2020; Roth et al., 2023), we also perform a small grid search over all the hyperparameters. We report the full details in Tab. 3.

For $\lambda_e$ and $\lambda_d$, we set $\lambda_e = 0$ for the Decoder setting and $\lambda_d = 0$ for the Encoder setting while altering the weighting for the other $\lambda$. All experiments run on a single RTX 2080TI GPU.

## C.2 STABILITY ANALYSIS

For the gradient stability analysis experiment, we repeat the following experiment for each of the four image datasets $\mathcal{D}$ that we consider. We first considered a fixed neural map $T_\theta$, which we choose to be a randomly initialized neural network, consisting of encoder and decoder, before any training. For $k = 1, \ldots, 5$, we sample a batch $\mathbf{x}_1^k, \ldots, \mathbf{x}_n^k \sim \mathcal{D}$ and let $r_n^k = \frac{1}{n} \sum_{i=1}^n \delta_{\mathbf{x}_i^k}$. We report the pariwise cosine similarity between the gradients of the DST and the GMG. Formally, we compute cos-sim$(\nabla_\theta \text{DST}_{r_n^k}(T_\theta), \nabla_\theta \text{DST}_{r_n^l}(T_\theta))$, and cos-sim$(\nabla_\theta \text{GMG}_{r_n^k}(T_\theta), \nabla_\theta \text{GMG}_{r_n^l}(T_\theta))$, for $k, l = 1, \ldots, 5$.

# D ADDITIONAL EMPIRICAL RESULTS

In this section, we report additional empirical results revolving around the regularization of encoder. First in D.1, we conduct the stability analysis when regularizing the encoder. Then, we report further

results of encoder and decoder regularization on DSprites. Lastly, we take a first exploratory step towards decoder-free disentangled representation learning in D.6.

## D.1 ENCODER STABILITY ANALYSIS

We repeat the following experiment for each of the four image datasets $\mathcal{D}$ that we consider. We first considered a fixed neural map $T_\theta$. For $i = 1, \ldots, 5$, we sample a batch $\mathbf{x}_1^i, \ldots, \mathbf{x}_n^i \sim \mathcal{D}$ and let $r_n^i = \frac{1}{n} \sum_{i=1}^n \delta_{\mathbf{x}_n^i}$. We report the pariwise cosine similarity between the gradients of the DST. Namely, for $i, j = 1, \ldots, 5$, we compute $\text{cos-sim}(\nabla_\theta \text{DST}_{r_n^i}(T_\theta), \nabla_\theta \text{DST}_{r_n^j}(T_\theta))$, and the GMG, $\text{cos-sim}(\nabla_\theta \text{GMG}_{r_n^i}(T_\theta), \nabla_\theta \text{GMG}_{r_n^j}(T_\theta))$.

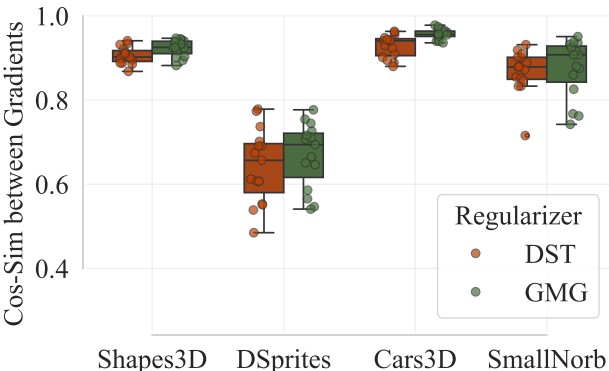

Figure 4: Gradient stability analysis on the DST and GMG as `Encoder` regularizations. The cosine similarity is computed between all pairs of gradients $\nabla_\theta$ obtained through 5 randomly sampled batches and a fixed network $T_\theta$ for each dataset.

## D.2 ENCODER ANALYSIS ON DSPRITES

Table 4: Disentanglement of regularizing the Encoder and the Encoder and Decoder as measured by **DCI-D** on DSprites. We highlight **best**, second best, and *third best* results for each method and dataset.

| DCI-D | $\beta$-VAE | $\beta$-TCVAE | $\beta$-VAE + HFS | $\beta$-TCVAE + HFS |
|---|---|---|---|---|
| **DSprites** (Higgins et al., 2017) | | | | |
| Base | 27.6 $\pm$13.4 | 36.0 $\pm$5.3 | *38.7* $\pm$15.7 | 48.1 $\pm$10.8 |
| + Enc-DST | *32.8* $\pm$15.0 | 36.5 $\pm$5.9 | 33.9 $\pm$15.9 | 48.9 $\pm$11.1 |
| + Enc-GMG | 27.5 $\pm$14.3 | *37.4* $\pm$5.8 | 31.0 $\pm$14.3 | 45.9 $\pm$10.9 |
| + Dec-DST | 28.6 $\pm$19.3 | 32.4 $\pm$8.5 | 39.3 $\pm$18.1 | 49.0 $\pm$11.2 |
| + Dec-GMG | **39.5** $\pm$15.2 | **42.2** $\pm$3.6 | **46.7** $\pm$2.0 | **50.1** $\pm$8.5 |

### D.3 Ablation on the Entropic Regularization Strength

**Effect of $\varepsilon$ on the entropic GW solver.** In this section, we remind some insights about the entropic GW solver (Peyré et al., 2016) introduced in § 2.2 and used to compute the GMG in Alg. 1. We provide its algorithmic details in Alg. 2. This solver naturally provides a trade-off between (i) the approximation of the true GW distance $\mathrm{GW}(p, q)$ and (i) the optimization speed (i.e., convergence rate). This trade-off is controlled by the entropic regularization strength $\varepsilon$.

(i) **Optimization.** The solver employs a mirror descent scheme that iteratively linearizes the entropic GW problem (EGWP) and applies the Sinkhorn algorithm. Each mirror descent step corresponds to a projection with respect to the KL divergence, which can be efficiently performed using the Sinkhorn algorithm. The solver is initialized with $\mathbf{P}_0 = \frac{1}{n^2}\mathbf{1}\mathbf{1}^\top$ and iterates as $\mathbf{P}_{t+1} \leftarrow \textsc{Sinkhorn}\left(-\mathbf{C}_{\mathcal{X}}\mathbf{P}_t\mathbf{C}_{\mathcal{Y}}, \varepsilon\right)$. Since a larger $\varepsilon$ in Sinkhorn leads to faster convergence (Cuturi, 2013; Altschuler et al., 2018), this subsequently accelerates each mirror descent step. In other words, it reduces the number of iterations within each Sinkhorn call, i.e., the number of *inner iterations* of the solver. Furthermore, Rioux et al. (2023) recently demonstrated that increasing $\varepsilon$ enhances the convexity of the entropic GW problem, thereby improving the convergence rate of the mirror descent scheme. This provides theoretical justification for using a larger $\varepsilon$ to reduce the number of mirror descent steps. In other words, it reduces the number of calls to Sinkhorn, i.e., the number of *outer iterations* of the solver.

We have empirically validated this behavior on DSprites with BetaVAE + GMG. We plot the mean amount of inner and outer iterations for the first 3 epochs over 5 seeds for six different values of $\varepsilon_0$, which we provide in Figure 5. We can observe the expected scaling of increased *inner* and *outer* iterations with decreased $\varepsilon_0$.

(ii) **Approximation.** Zhang et al. (2023) show that when using inner products and squared Euclidean distances as costs, and for $\varepsilon \in (0, 1]$, the approximation error scales as:

$$|\mathrm{GW}_\varepsilon(p, q) - \mathrm{GW}(p, q)| \lesssim \varepsilon \log(1/\varepsilon),$$

where the constants in $\lesssim$ depend on $d_{\mathcal{X}}$ and $d_{\mathcal{Y}}$, that is, the dimensions of the support of $p$ and $q$, as well as their fourth-order moments.

Given this trade-off, the practical goal is to select an $\varepsilon$ value that is sufficiently large to ensure fast convergence while avoiding any degradation in performance. Across all our experiments, we found that $\varepsilon = 0.1$ struck the right balance. We validate this observation in the next paragraph.

---

**Algorithm 2** Entropic Gromov-Wasserstein solver (Peyré et al., 2016), (Scetbon et al., 2022, Alg. 2).

---

1: **Require:** samples $\mathbf{x}_1, \ldots, \mathbf{x}_n \sim p$; $\mathbf{y}_1, \ldots, \mathbf{y}_n \sim q$; cost functions $c_{\mathcal{X}}, c_{\mathcal{Y}}$; entropic regularization scale $\varepsilon_0$ (default = 0.1), statistic operator on cost matrix `stat` (default = `mean`).
2: $\mathbf{C}_{\mathcal{X}} \leftarrow [c_{\mathcal{X}}(\mathbf{x}_i, \mathbf{x}_{i'})]_{1 \leq i,i' \leq n}$          $\triangleright$ usually $\mathcal{O}(n^2 d_{\mathcal{X}})$
3: $\mathbf{C}_{\mathcal{Y}} \leftarrow [c_{\mathcal{Y}}(\mathbf{y}_j, \mathbf{y}_{j'})]_{1 \leq j,j' \leq n}$          $\triangleright$ usually $\mathcal{O}(n^2 d_{\mathcal{Y}})$
4: $\varepsilon \leftarrow \varepsilon_0 \cdot \texttt{stat}(\mathbf{C}_{\mathcal{X}}) \cdot \texttt{stat}(\mathbf{C}_{\mathcal{Y}})$          $\triangleright$ usually $\mathcal{O}(n^2)$
5: $\mathbf{P}_t \leftarrow \frac{1}{n^2}\mathbf{1}_n\mathbf{1}_n$          $\triangleright n^2$
6: **while** converged **do**
7:      $\mathbf{C}_{t+1} \leftarrow -\mathbf{C}_{\mathcal{X}}\mathbf{P}_t\mathbf{C}_{\mathcal{Y}}$          $\triangleright n^3$ or $n^2(d_{\mathcal{X}} + d_{\mathcal{Y}})$
8:      $\mathbf{P}_{t+1} \leftarrow \textsc{Sinkhorn}(\mathbf{C}_{t+1}, \varepsilon)$          $\triangleright \mathcal{O}(n^2)$
9: **end while**
10: Compute $\mathrm{GW}_\varepsilon(p_n, q_n)$ from $\mathbf{P}_t$ using Eq. (EGWP)          $\triangleright n^3$ or $n^2(d_{\mathcal{X}} + d_{\mathcal{Y}})$
11: **return** $\mathrm{GW}_\varepsilon(p_n, q_n)$

---

**Effect of $\varepsilon$ in disentanglement.** We investigate the effect of the entropic regularization strength $\varepsilon$ used to compute GMG on the disentanglement performances. The results are presented in Fig. 5 and show that performance is robust to the choice of entropic regularization scale $\varepsilon_0$. We observed this both with respect to a setting, where we see major improvements from the GMG (DSprites with BetaVAE) as well as one where we only observe minor improvements (DSprites with BetaTCVAE). This validates our choice to use a single reasonable value for all our experiments, namely $\varepsilon_0 = 0.1$. This robustness with respect to the entropic strength was also observed in a recent work (Piran et al., 2024) proposing a similar gap regularization, based on the entropic multi-marginal OT problem. See (Piran et al., 2024, Fig. 3) for experiments highlighting this robustness. Our intuition is as

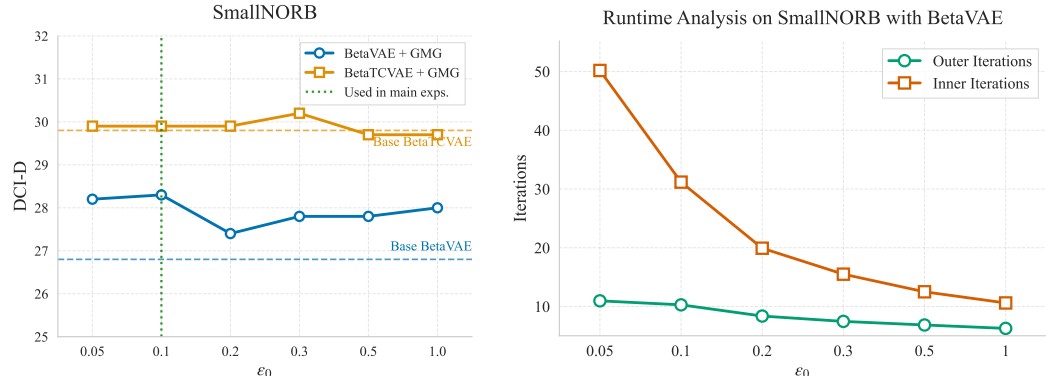

Figure 5: Analysis of the effect of the entropic regularization scale $\varepsilon_0$ on the disentanglement performances when learning with the GMG. We use the SmallNORB dataset and consider the $\beta$-VAE (in blue) and $\beta$-TCVAE (in orange) settings. The GMG is applied to the decoder $d_\theta$ with cost functions $c_\mathcal{X} = c_\mathcal{Y} = \text{cos-sim}(\cdot)$, corresponding to the setting that provides better disentanglement. We compute the GMG using Algorithm 1 with `stat=mean`. We investigate the effect of the entropic regularization scale $\varepsilon_0$ by testing five other values of $\varepsilon_0$ besides the one used in all other experiments in the paper (namely, $\varepsilon_0 = 0.1$). The values tested are $\varepsilon_0 \in \{0.05, 0.2, 0.3, 0.5, 1\}$. We also include the baseline result without using the GMG (dashed line) as a comparison. Additionally, we provide a runtime analysis with respect to both the inner and outer iteration of the GW-solver.

Table 5: Hyperparameter grid search for Shapes3D $128 \times 128$.

| Method | Parameter | Values |
|:---:|:---:|:---:|
| $\beta$-(TC)VAE + GMG | $\beta$ $\lambda$ | [10, 16] [0.1, 1, 10] |
| $\beta$-(TC)VAE + GMG + HFS | $\beta$ $\lambda$ $\gamma$ | [2, 4] [0.1, 1, 10] [1, 10] |

follows: when using Sinkhorn (in the case of M3G) or GW/quadratic OT (in our work) to compute a *training loss*, $\varepsilon$ acts as a *sharpness* parameter, emphasizing certain pairs of points more strongly. While the *loss* value changes with $\varepsilon$, the optimization of network variables on top of this loss appears to be largely unaffected by the sharpness introduced by $\varepsilon$. However, when the output of Sinkhorn or GW is used directly for predictions or learning flows, for example, in Monge maps (Pooladian and Niles-Weed, 2021; Kassraie et al., 2024) or Gromov-Monge maps (Klein et al., 2024), the entropic regularization strength $\varepsilon$ has a much stronger influence and requires careful tuning.

## D.4 SCALING TO HIGHER IMAGE RESOLUTIONS

As detailed in § 3.2, the computation of the GMG scales linearly with the data dimension, enabling our method to handle high-dimensional settings effectively. To demonstrate this scalability, we benchmark our approach on the Shapes3D dataset upscaled to $128 \times 128$ resolution. To accommodate the increased image resolution, we extend the Decoder by adding one additional layer. Using our best-performing configuration from the $64 \times 64$ experiments**Cos** costs and the GMG applied to the decoder $d_\theta$we conduct a focused hyperparameter search as described in Table 5. The results, summarized in Table 6, compare the four base models Beta(TC)VAE (+ HFS), both with and without the GMG. Our findings indicate that the proposed setup scales to higher resolutions while preserving the performance improvements achieved by the GMG. Finally, we visually validate these results by plotting the latent traversals for the best-performing configuration, as shown in Figure 6.

Table 6: Impact of the GMG, applied with **Cos** as the cost function on the decoder $d_\theta$, on disentanglement performance using upscaled $128 \times 128$ Shapes3D images. Performance is evaluated using **DCI-D**, with the **best** result highlighted for each method.

| With **Cos** costs | $\beta$-VAE | $\beta$-TCVAE | $\beta$-VAE + HFS | $\beta$-TCVAE + HFS |
|---|---|---|---|---|
| **Shapes3D** ($128 \times 128$) (Burgess and Kim, 2018) | | | | |
| Base | 54.3 $\pm$19.3 | 74.6 $\pm$17.3 | 87.2 $\pm$2.7 | 88.6 $\pm$11.4 |
| + Dec-GMG | **63.9** $\pm$8.2 | **82.0** $\pm$12.8 | **90.4** $\pm$3.7 | **92.2** $\pm$8.2 |

## D.5 LATENT TRAVERSAL VISUALIZATION

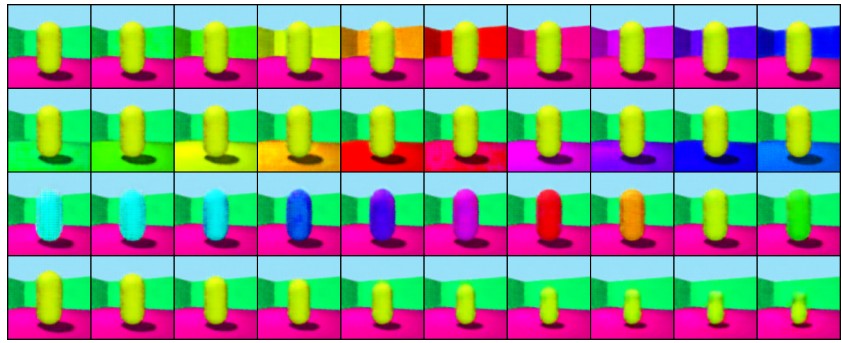

Figure 6: Latent traversal visualization for Shapes3D $128 \times 128$ with our best performing setup, BetaTCVAE + HFS + GMG. We select the best performing result out of 5 seeds achieving a DCI-D of 99.4. We plot four different latent dimensions while traversing them from $-1.0$ to $1.0$. As visualized the model has clearly learned to separate wall hue, object hue, scale, and floor hue into different latent dimensions.

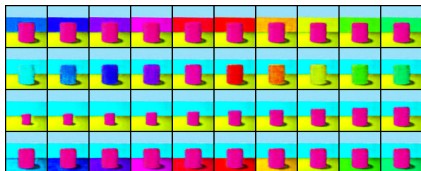

Figure 7: Latent traversal visualization for Shapes3D $64 \times 64$ with our best performing setup, BetaTCVAE + HFS + GMG. We select the best performing result out of 5 seeds achieving a DCI-D of 100.0. We plot four different latent dimensions while traversing them from $-1.0$ to $1.0$. As visualized the model has clearly learned to separate wall hue, object hue, scale, and floor hue into different latent dimensions.

### D.6 Towards Decoder-free Disentanglement

Recently, works such as (Burns et al., 2021; von Kgelgen et al., 2021; Eastwood et al., 2023; Matthes et al., 2023; Aitchison and Ganev, 2024) have shown the possibility of disentanglement through self-supervised, contrastive learning objectives in an effort to align with the scalability of encoder-only representation learning (Chen et al., 2020b; Zbontar et al., 2021; Bardes et al., 2022; Garrido et al., 2023). However, these encoder-only approaches still require weak supervision or access to multiple views of an image to learn meaningful representations of the data samples.

As the goal of geometry preservation connects the data manifold and the latent domain through a minimal distortion objective and is applicable to both the encoder and decoder of a VAE (§3, Table 4), we posit that its application may provide sufficient training signal to learn meaningful representations and encourage disentanglement, eliminating the need for a reconstruction loss and decoder. Table 7 shows preliminary results on unsupervised decoder-free disentangled representation learning on the Shapes3D benchmark, where the decoder and associated reconstruction objective have been removed.

Table 7: Disentanglement (DCI-D) without a decoder trained with various regularizations on Shapes3D (Burgess and Kim, 2018).

| Decoder-free | $\beta$-VAE | $\beta$-TCVAE |
|---|---|---|
| Base | $0.0 \pm 0.0$ | $0.0 \pm 0.0$ |
| **L2$^2$** : $\| \cdot - \cdot \|_2^2$ | | |
| + DST | $\underline{38.2} \pm 0.8$ | $42.7 \pm 1.6$ |
| + GMG | $13.9 \pm 0.4$ | $20.5 \pm 0.5$ |
| **ScL2$^2$** : $\alpha \| \cdot - \cdot \|_2^2, \alpha > 0$ learnable | | |
| + DST | $\mathbf{45.6} \pm 1.2$ | $\mathbf{53.5} \pm 1.0$ |
| + GMG | $15.2 \pm 0.3$ | $25.2 \pm 0.6$ |
| **Cos** : cos-sim$(\cdot, \cdot)$ | | |
| + DST | $37.0 \pm 0.4$ | $\underline{46.1} \pm 1.5$ |
| + GMG | $37.0 \pm 0.9$ | $38.8 \pm 1.1$ |

Standard approaches such as $\beta$-VAE or $\beta$-TCVAE collapse and do not achieve measurable disentanglement (DCI-D of $0.0$). However, the inclusion of either DST or GMG significantly raises achievable disentanglement and, combined with the $\beta$-TCVAE matching objective, can achieve DCI-D scores of up to $53.5$ without needing any decoder or reconstruction loss. While these are preliminary insights, we believe they offer promise for more scalable approaches to unsupervised disentangled representation learning and potential bridges to popular and scalable self-supervised representation learning approaches. Note, that here the distortion loss significantly outperforms the GMG. This is expected due to the nature of the GMG, as the distortion loss offers a more restrictive and, thus, stronger signal for learning representations, which is necessary in the absence of a reconstruction objective. This highlights that while in most scenarios (§ 2.1, Figure 2), the GMG is preferable over the distortion loss, there also exist settings where a more restrictive optimization signal is desirable.

## E   PYTHON CODE FOR THE COMPUTATION OF THE GROMOV-MONGE GAP

```python
import jax
import jax.numpy as jnp

from ott.geometry import costs, geometry
from ott.solvers.quadratic import gromov_wasserstein
from ott.problems.quadratic import quadratic_problem

def gromov_monge_gap_from_samples(
    source: jax.Array,
    target: jax.Array,
    cost_fn: costs.CostFn = costs.Cosine(),
    epsilon: float = 0.1,
    stat_fn: Callable, # usually computes the mean of the cost matrix
    **kwargs,
) -> float:
    """Gromov Monge gap regularizer on samples."""

    # define source and target geometries
    cost_matrix_x = cost_fn.all_pairs(x=source, y=source)
    scale_cost_x = stat_fn(scale_cost_x)
    cost_matrix_x = cost_matrix_x / jax.lax.stop_gradient(scale_cost_x)
    geom_xx = geometry.Geometry(cost_matrix=cost_matrix_x)

    cost_matrix_y = cost_fn.all_pairs(x=target, y=target)
    scale_cost_y = stat_fn(cost_matrix_y)
    cost_matrix_y = cost_matrix_y / jax.lax.stop_gradient(scale_cost_y)
    geom_yy = geometry.Geometry(cost_matrix=cost_matrix_y)

    # define and solve entropic GW problem
    prob = quadratic_problem.QuadraticProblem(geom_xx, geom_yy)

    solver = gromov_wasserstein.GromovWasserstein(
        epsilon=epsilon, **kwargs
    )
    out = solver(prob)

    # compute the distortion induced by the map
    distortion_cost = jnp.nanmean(
        (geom_xx.cost_matrix - geom_yy.cost_matrix)**2
    )

    # compute optimal (entropic) gromov-monge displacement
    reg_gw_cost = out.reg_gw_cost
    ent_reg_gw_cost = reg_gw_cost - 2 * epsilon * jnp.log(len(source))

    # compute gromov-monge gap
    loss = distortion_cost - ent_reg_gw_cost

    return loss * jax.lax.stop_gradient(scale_cost_x * scale_cost_y)
```

