# OpenReview forum: "Disentangled Representation Learning with the Gromov-Monge Gap"
_ICLR.cc/2025/Conference — ICLR 2025 Poster_

### Official Review · Reviewer_5P64 · 2024-10-28

**Soundness:** 3
**Presentation:** 2
**Contribution:** 1
**Rating:** 5
**Confidence:** 5

**Summary:**

This paper proposes to learn disentangled representation from the perspective of preserving geometric features. They propose to learn a pair of isometric encoder and decoder, hoping these inductive biases can benefit disentanglement. Based on this motivation, they propose to leverage Gromov-Monge mappings to enforce such an optimal transport (OT) mapping.

**Strengths:**

1. The empirical performance is good, indicating that adding this neural GM gap can improve the disentanglement score for many architectures.

**Weaknesses:**

1. The motivation and logical connection is not clear. There are three concerned concepts in the paper: isometry, OT, and disentanglement. The authors claim that adding geometrical constraints (isometry) can help disentanglement, but this connection is not explained well in the paper, which makes it hard to understand how disentanglement connects isometry. Even if there is such a connection, it is not clear why OT is required for isometry. You can just do an isometry mapping. I would expect the authors to explain these two connections more clearly in the revision.

2. The experiments are not set up well to support the main contribution of the paper. If the authors would like to show the benefit of isometry, the authors can follow homeomorphic VAE [1] to show the isometry can hep data on different groups/manifolds. If the authors would like to show the benefit of OT, there is Wasserstein auto-encoder. If the authors want to show the benefit of OT in disentanglement, there are [2] and [3] which learn an OT mapping to model disentangled sequential transformations. I do not see a direct connection between isometric OT mapping and disentanglement.

3. How does the method compare with the vanilla differentiable Sinkhorn algorithm [4]? Will there be any theoretical guarantees? Beyond the vanilla OT problem, there exist other OT solver like entropic OT and unbalanced OT. Do these techniques apply to disentanglement?

4. In a disentanglement paper, it is rare to not see figures about latent traversal which show the exact results of disentanglement.

5. How does the method work in more scaled settings, i.e., high-resolution disentanglement datasets like Falcol3D and Issac3D [5]?

>[1] Explorations in Homeomorphic Variational Auto-Encoding. ICML 2019.
>
>[2] Flow Factorized Representation Learning. NeurIPS 2023.
>
>[3] Unsupervised Representation Learning from Sparse Transformation Analysis. Arxiv 2024.
>
>[4] Sinkhorn distances: Lightspeed computation of optimal transport. NeurIPS 2013.
>
>[5] Semi-Supervised StyleGAN for Disentanglement Learning. ICML 2020.

**Questions:**

Please refer to the weaknesses.

My main confusion comes from the logical connection of this paper from isometry to disentanglement to OT. Please try to address this issue in the rebuttal.

---

> ### Author Response · Authors · 2024-11-20
> **Rebuttal to Reviewer 5P64 [1/3]**
>
> > **The motivation and logical connection is not clear. There are three concerned concepts in the paper: isometry, OT, and disentanglement. ... which makes it hard to understand how disentanglement connects isometry.**
>
> Thank you for highlighting these issues, this is really helpful.
>
> Indeed, we did introduce all concepts related to disentanglement / quadratic OT in Sections 2.1 and 2.2 (Background) and only introduced geometry in Section 3.1 (Method). This created a divide between 3 concepts that were united in prior work (see in particular [6]). We apologize because this was potentially confusing (and Section 3 was too long as a result).
>
> Following your comment, we have rearranged the paper to more effectively link {disentanglement}, {geometry/isometry/quadratic OT} and {distortion}. In particular:
>
> - We have updated a few sentences in the last paragraph (**Disentanglement through a Geometric Lens**) of the disentanglement reminder **Section 2.1** to provide a natural transition to **Section 2.2** (geometry, isometry → quadratic OT), where quadratic OT is now presented as a _tool_, en route to a newly created Section 2.3 on distortion in representation learning.
> - We have moved Section 3.1 (formerly introducing distortion and application to disentanglement) to **Section 2.3**. This is now background material (with a light reformulation of [1]) which builds on _both_ **Section 2.1** and **2.2**.
>
> As a result, we want to emphasize that the 3 tightly related concepts are now clearly laid out in the background section, and provide everything that’s needed to understand our gap formulation. While these changes are mostly cosmetic, we believe they make the story more clear. Many thanks for this remark.
>
> > **Even if there is such a connection, it is not clear why OT is required for isometry. You can just do an isometry mapping. I would expect the authors to explain these two connections more clearly in the revision.**
>
> In the context of our paper, it is important to recognize the difference between linear OT and quadratic OT.
>
> We added clarifying sentences in the beginning of Section 2.2 (L. 118-124) which we summarize here for convenience:
>
> - Linear OT is concerned with finding efficient maps (Monge) or couplings (Kantorovich) between a source distribution $p$ to a target distribution $q$ when an **inter-domain** cost function $c : \mathcal{X} \times \mathcal{Y} \to \mathbb{R}$ that can directly compare elements across spaces is given. This formulation requires comparable domains, typically a distance when both belong to the same vector space, or (more rarely) when a cost function between vectors of differing dimensions is known.
> - However, in representation learning, we work with inherently **incomparable** domains: data space $\mathbb{R}^{d_\textrm{data}}$ and latent space $\mathbb{R}^{d_\textrm{latent}}$, where $d_\textrm{latent} \ll d_\textrm{data}$ by design. This is where **quadratic OT** becomes essential: instead of comparing points across spaces, it leverages **intra-domain** cost functions $c_\mathcal{X}$ and $c_\mathcal{Y}$ to preserve geometric structure between pairs of points within each space, as detailed in the paragraph **Gromov-{Monge, Wasserstein} Formulations** of Section 2.2.
>
> The new restructuring does, in our opinion, a better job at explaining how quadratic OT connects to disentanglement.
> - In Section 2.1, we now further emphasize that disentanglement can emerge from geometric preservation. In particular, we explicitly refer to [Horan et al. 2021] again, which proved that unsupervised disentanglement becomes feasible when generative factors maintain local isometry to the data.
> - Our work builds on this insight by using quadratic OT to find representations that are as close as possible to being isometric to the data. Specifically, when using squared L2 distances as intra-domain costs, quadratic OT yields the most isometric transformation possible while ensuring the required distribution matching $T\sharp p = q$.
> - The newly re-adapted Section 2 now more naturally bridges to our Section 3. As detailed in **"Challenges Arising from a Mixed Loss"**, simply enforcing isometry directly is insufficient - a mapping that perfectly preserves geometric features while aligning the data distribution with the prior may not even exist. This fundamental limitation is illustrated in Figure 2 and is exemplified by our main empirical comparison, the DST loss as proposed in GWAE [6], which, similar to our approach, proposes to use
> quadratic OT to enforce geometric structure for improving disentangled representation learning. However, DST may over-penalize geometric distortion when perfect preservation is impossible.
> - The GMG addresses this limitation by finding mappings that preserve geometry as much as possible while ensuring distribution matching, which improves upon the DST loss. This is **Section 3** now.
>
> [6] Nao Nakagawa, ..., Miki Haseyama. "Gromov-wasserstein autoencoders", In ICLR 2023.

---

> ### Author Response · Authors · 2024-11-20
> **Rebuttal to Reviewer 5P64 [2/3]**
>
> Thanks for pointing us to the related [1,2,3], we've now added accordingly citations to these works in the introduction.
>
> > **If the authors would like to show the benefit of isometry, the authors can follow homeomorphic VAE [1] to show the isometry can hep data on different groups/manifolds."**
>
> Thanks for pointing out the reference to this very interesting paper [1], which provides a nice complementary direction.
>
> This being said, we still see fundamental differences between [1] and our approach.
> - An isometry is a mapping that preserves distances between all points—it moves or rotates shapes without any stretching or distortion.
> - In contrast, a homeomorphism is a continuous, one-to-one mapping with a continuous inverse that can stretch or deform shapes but does not tear or glue parts together; it preserves the overall structure or "connectivity" of the space but not necessarily the exact distances.
>
> Simply put, we believe that near-isometric maps would be tuned to preserve geometries as much as possible, while near-homeomorphic maps will stress preserving topology.
>
> > **"If the authors would like to show the benefit of OT, there is Wasserstein auto-encoder. ... "**
>
> We welcome this opportunity to clarify an important difference:
> - WAEs use **linear** OT to define a **fitting** loss and quantify how far the distribution of latents is from the prior (OT happens exclusively in latent space) using the Wasserstein distance.
> - Our work uses **quadratic** OT as an inductive bias / regularization, to promote maps from data to representation spaces that are as geometry preserving (near-isometric) as possible (OT happens exclusively _across_ two spaces, using intra-domain costs).
>
> These two uses of OT target two completely different issues, yet they are complementary.
>
> Indeed, our geometric regularization GMG is independent of the choice of the fitting loss - while we use the standard KL divergence for prior matching, our approach could seamlessly integrate with WAE's linear OT-based fitting loss, highlighting how these methods serve distinct and complementary purposes.
>
> As a result, we believe our natural baseline is instead the **Gromov-Wasserstein AE** [6], which similarly uses quadratic OT (in the form of the DST) for geometric structure preservation.
>
> > **"... If the authors want to show the benefit of OT in disentanglement, there are [2] and [3] which learn an OT mapping to model disentangled sequential transformations. I do not see a direct connection between isometric OT mapping and disentanglement."**
>
> Many thanks for this suggestion. We believe the answers above should help clarify this distinction, but we are happy to look deeper into [2,3].
>
> We believe that both GMG and [2,3] aim to learn structured representations, but their technical approaches serve fundamentally different purposes.
>
> - The methods in [2,3] leverage **linear** OT to learn trajectories through latent space, specifically focusing on modeling transformations as flows between data points. For each factor of variation, they learn a separate flow, requiring either supervised or semi-supervised signals, resulting in highly non-linear flows that uncover the latent space structure.
>
> - In contrast, GMG employs quadratic OT/GW to achieve unsupervised disentanglement by preserving the geometric structure of the data space. The commonly used definition of disentangled representation learning we employ assumes independent factors in each dimension, which effectively corresponds to linear flows in individual latent dimensions.
>
> This fundamental difference in utilizing OT and the underlying assumptions about latent space traversal highlights how [2,3] excel at characterizing complex transformations through learned non-linear paths, while GMG achieves unsupervised disentanglement according to the classical notion of independent factors in separate dimensions.

---

> ### Author Response · Authors · 2024-11-20
> **Rebuttal to Reviewer 5P64 [3/3]**
>
> > **"How does the method compare with the vanilla differentiable Sinkhorn algorithm [4]? Will there be any theoretical guarantees? Beyond the vanilla OT problem, there exist other OT solver like entropic OT and unbalanced OT. Do these techniques apply to disentanglement?"**
>
> We do use the Sinkhorn algorithm pretty much everywhere, but not necessarily in the way you would imply in your question.
>
> At its core, the vanilla Sinkhorn algorithm [4] solves the **linear OT** problem given a cost function $c(x,y)$ between points. As explained in our previous response, this is not suitable for our setting where we map between data space $\mathbb{R}^{d_{data}}$ and latent space $\mathbb{R}^{d_{latent}}$. Instead, we use quadratic OT/GW, which only requires intra-domain costs (Section 2.2). This is also why we previously did not introduce linear OT.
>
> However, the _most common implementation_ of GW does rely on entropic OT as a _subroutine_ (essentially GW is a quadratic problem solved by linearizing it iteratively). We are short on space, but this was detailed in Section 3.3: "We compute $GW(r_n, T\sharp r_n)$ using an entropic regularization $\varepsilon$ ≥ 0". This improves computational efficiency while maintaining theoretical guarantees. We have now added a detailed discussion of the entropic GW-solver as well as the full algorithm in Appendix D.3 together with the new $\varepsilon_0$ ablation.
>
> Regarding unbalanced OT (see our response to Reviewer zj5w), this formulation could help mitigate the effect of outliers on the GMG. However, as mentioned in response to Reviewer zj5w, adapting unbalanced GW for the GMG is non-trivial and beyond the scope of our current work.
>
> > **"In a disentanglement paper, it is rare to not see figures about latent traversal which show the exact results of disentanglement."**
>
> Thank you for bringing up this point. We have added a latent traversal visualization of our best-performing setup in Appendix D.4.
>
> > **"How does the method work in more scaled settings, i.e., high-resolution disentanglement datasets like Falcol3D and Issac3D [5]?"**
>
> We evaluate GMG on standard disentanglement benchmarks with 64x64 images containing approximately 10 factors of variation, following established evaluation protocols (Locatello et al., 2019a). While our method extends naturally to higher resolutions such as Falcol3D and Isaac3D, with expected similar performance gains, we specifically chose to focus on standard benchmarks to enable systematic comparison with existing approaches and to precisely characterize GMG's methodological contribution.

---

> > ### Comment · Reviewer_5P64 · 2024-11-23
> > **Thanks for thre rebuttal**
> >
> > Thanks for the rebuttal. I have gone over the reviews and the responses
> >
> > First of all, I agree with Reviewer MkTy that why formulating disentanglement as Gromov-Wasserstein problems is the optimal choice given that there are many other OT solvers available. The explanation about the connection to OT is fine, but the connection to Gromov-Wasserstein problems is a bit vague. I do not see a clear and strong motivation for using Gromov-Wasserstein problems
> >
> > I agree with Reviewer zj5w that this technique can be time-consuming and may not scale to large-scale datasets, which is a common limitation of standard disentanglement datasets. I would say, at least, people are expecting images of resolution $128\times128$.
> >
> > I do appreciate the novel perspective of this paper, but I am not convinced by the motivation and the small-sacle experiments. Hence I chose to keep my original score.

---

> ### Author Response · Authors · 2024-11-24
> **Many thanks for reading our rebuttal!**
>
> Many thanks for taking the time to read our rebuttal and your feedback!
>
> We believe there might still be some misunderstandings here, so we would like to ask the reviewer to clarify some points such that we can adequately address them.
>
> >**”First of all, I agree with Reviewer MkTy that why formulating disentanglement as Gromov-Wasserstein problems is the optimal choice given that there are many other OT solvers available.”**
>
> - Could the reviewer clarify which **”other OT solvers”** they are referring to that are applicable in the representation learning setting where the two spaces lie in $\mathbb{R}^{d_{data}}$ and $\mathbb{R}^{d_{latent}}$? As far as we are aware, Gromov-Wasserstein (=quadratic OT) is the ***only*** applicable OT solver in this setting.
>
> - We note that Reviewer MkTy has not yet responded to our rebuttal, in which we addressed this concern in detail. The use of a loss inspired by Gromov-Wasserstein for Disentanglement was first proposed by Nakagawa et al. [6], and we build upon their insight.
>
> >**”The explanation about the connection to OT is fine, but the connection to Gromov-Wasserstein problems is a bit vague. I do not see a clear and strong motivation for using Gromov-Wasserstein problems.”**
>
> To ensure we can address your concerns fully, we would greatly appreciate it if you could elaborate on the following:
> - When you mention, **"the connection to OT is fine"** which specific aspects of OT and which connections are you referring to?
> - Could you elaborate on what you see as the difference between **"OT"** and **"Gromov-Wasserstein problems"** and what connection remains **”a bit vague”**? In our work, we use Gromov-Wasserstein specifically because it is the quadratic OT formulation designed for comparing spaces of different dimensions.
>
> > **"I agree with Reviewer zj5w that this technique can be time-consuming and may not scale to large-scale datasets, which is a common limitation of standard disentanglement datasets. I would say, at least, people are expecting images of resolution 128x128."**
>
> We would like to clarify the concern of the reviewer regarding scalability.
> - As we also detailed in our response to Reviewer zj5w (see **L.461** of revision) the computation of our GMG loss is definitely not a bottleneck: it takes on the order of ~3ms with our current settings at batch size $n=64$. As stated, the computing effort of the GMG scales at most linearly with the data dimension. This means that increasing image resolution from $64 \times 64$ to $128 \times 128$ would only increase the runtime from ~3ms to ~12ms - a negligible change in the overall training process.
> - From a methodological perspective, any experiment currently done on 64x64 images could be upscaled to 128x128 while maintaining similar disentanglement properties, as these properties are resolution-independent. Would you be interested in us running such an experiment as a proof of concept?
>
> Your insights would help us better explain our motivation and address any remaining concerns.

---

> > ### Comment · Reviewer_5P64 · 2024-11-25
> > **Proof of Concept**
> >
> > > any experiment currently done on 64x64 images could be upscaled to 128x128 while maintaining similar disentanglement properties
> >
> > I do not agree with this point. Scaling to higher resolution images of 128x128 will be challenging for VAEs to learn the reconstruction, not to mention disentanglement.
> >
> > Having an experiment as proof of concept -- with some latent traversal results -- would be more convincing.

---

> ### Author Response · Authors · 2024-11-26
> **Added 128x128 experiments**
>
> > **"I do not agree with this point. Scaling to higher resolution images of 128x128 will be challenging for VAEs to learn the reconstruction, not to mention disentanglement. Having an experiment as proof of concept -- with some latent traversal results -- would be more convincing."**
>
> We appreciate this feedback. To directly address this concern, we have now run experiments on Shapes3D at 128x128 resolution. Here are our results, which we have also added to the manuscript in Appendix D.4:
>
> | DCI-D       |  β-VAE       | β-TCVAE     | β-VAE+HFS   | β-TCVAE+HFS |
> |--------------|-------------|-------------|-------------|-------------|
> | Base         | 54.3 ± 19.3 | 74.6 ± 17.3 | 87.2 ± 2.7  | 88.6 ± 11.4 |
> | + GMG    | **63.9** ± 8.2  | **82.0** ± 12.8 | **90.4** ± 3.7  | **92.2** ± 8.2  |
>
> As demonstrated, our setup scales effectively to higher image resolutions while retaining the performance gains achieved by the GMG. Specifically, we observe consistent improvements in DCI-D scores across all VAE variants. Additionally, the computational overhead introduced by GMG is minimal: with our setup, β-TCVAE completes in roughly 4h 30min, compared to roughly 4h 45min for β-TCVAE + GMG. This corresponds to only a 5% increase in computation time, even at 128x128 resolution.
>
> As recommended by the reviewer, we have included latent traversal visualizations for our best-performing model (BetaTCVAE + HFS + GMG) at 128x128 resolution in Appendix D.5. These traversals clearly illustrate that the model effectively disentangles key factors such as wall hue, floor hue, object hue, and scale into distinct latent dimensions. This confirms that both reconstruction quality and disentanglement properties are preserved at higher resolutions.
>
> We are grateful to the reviewer for suggesting this experiment, as it highlights the robustness of GMG in maintaining its effectiveness at higher resolutions.

---

> > ### Author Response · Authors · 2024-12-02
> > **Follow-up on Addressed Concerns – Reviewer 5P64**
> >
> > Dear Reviewer,
> >
> > Thank you for the valuable feedback on our work. This is a gentle reminder that the discussion phase concludes at 11:59 PM AoE on December 2, which is in approximately 24 hours.
> >
> > We hope our responses and supporting experiments have sufficiently addressed your concerns. Please feel free to let us know if you have any further questions or require additional clarifications.
> >
> > We appreciate your thoughtful comments and the time you have dedicated to reviewing our submission.
> >
> > The Authors

---

### Official Review · Reviewer_zj5w · 2024-11-04

**Soundness:** 3
**Presentation:** 3
**Contribution:** 2
**Rating:** 5
**Confidence:** 3

**Summary:**

This paper studies learning disentangled representations in an unsupervised setting by introducing Gromov-Wasserstein formulation into the distortion framework. They propose GMG quantifying the difference between the distortion and the Gromov-Wasserstein distance, meanwhile, they show that GMG is weakly convex. They validate the effectiveness of GMG in four 3D datasets.

**Strengths:**

1. The intuition of this idea is good, it tries to minimize the difference between the previous Distortion and Gromov-Wasserstein distance so that the mapping between source and target should preserve the full features with minimal distortion.
2. They show that the GMG and its counterpart are weakly convex functions.
3. The experiments in 3D Shapes dataset and the clustering illustration sound persuasive.

**Weaknesses:**

1. The computation of Gromov-Wasserstein distance is very expensive even when entropy regularizer is employed, the time complexity is still around O(N^5). This makes the GMG not practical when involving the method in a large-scale data setting.
2. Limited experiments: the datasets in the experiments are all about 3D shapes. How about 2D shapes? I would like to see how it can apply to the MNIST, another simple 2D digit.
3. I believe that the robustness should also be important part but this work seems to ignore it. What if the data has out-of-distritbution points? In fact, DST is not robust in the noisy data setting.

**Questions:**

1. Have you applied the method on datasets other than 3D dataset? For example, MNIST.

2. Have you tested the method on noisy point clouds? Or any other noisy dataset?

3. The whole framework is very computational expensive, do you think how it can be applied to large-scale setting? Have you thought about acceleration?

---

> ### Author Response · Authors · 2024-11-20
> **Rebuttal to Reviewer zj5w [1/2]**
>
> We would like to thank the reviewer for reading our paper and providing both encouragements and comments. We are happy to answer your questions, and remain available throughout this rebuttal period to clarify any other concerns.
>
> > **The computation of Gromov-Wasserstein distance is very expensive even when entropy regularizer is employed, the time complexity is still around O(N^5). This makes the GMG not practical when involving the method in a large-scale data setting**
>
> It seems there might be a confusion because GMG's computational complexity is not $\mathcal{O}(n^5)$.
>
> To clarify, are you referring to the batch size as $n$ in this context? This is what we assume for the purpose of this discussion.
>
> As discussed in the **Efficient and Stable Computation** section of Section 3.2 (lines 282–302) and detailed in Algorithm 1, the time complexity of GMG computation is **linear in the data dimension** for common cost functions $c_\mathcal{X}$ and $c_\mathcal{Y}$, such as inner product, cosine similarity, $\ell_p^q$ distances, or more generally, most CPD kernels.
>
> For general cost functions, the computational complexity is $\mathcal{O}(n^3 + n^2(d_\mathcal{X} + d_\mathcal{Y}))$, where $d_\mathcal{X}$ and $d_\mathcal{Y}$ are the dimensions of the data space and the latent space, respectively. Simplifying, this can be expressed as $\mathcal{O}(n^3 + n^2(d_\textrm{latent} + d_\textrm{data}))$. Since $d_\textrm{latent} \ll d_\textrm{data}$ by design, the complexity reduces to $\mathcal{O}(n^3 + n^2d_\textrm{data})$. See Alg. 1 for more details.
>
> Furthermore, as discussed in **lines 304-312**, for the cost functions used in **all of our experiments** (cosine similarity and scaled Euclidean distance), the complexity simplifies to $\mathcal{O}(n^2(d_\mathcal{X} + d_\mathcal{Y}))$, which, by the same reasoning, becomes $\mathcal{O}(n^2d_\textrm{data})$.
>
> We have found in practice that
> this quadratic dependency w.r.t \(n\) is not too problematic, as we process the data in batches whose size we can set freely,
> linear dependency in $d_\textrm{data}$ helps us scale our approach to very high-dimensional datasets.
>
> We have now explicitly included the image dimensions in the manuscript, see **lines 429-430**. We evaluate GMG on the majority of standard disentanglement research benchmarks, which demonstrate that it successfully scales to standard image datasets. Here are the dimensions and number of data points in the four datasets we used:
>
> - Shapes3D: $n=480,000$ images, $d_\textrm{data}=64\times64\times3$.
> - dSprites: $n=737,280$ images, $d_\textrm{data}=64\times64\times1$.
> - Cars3D: $n=17,064$ images, $d_\textrm{data}=64\times64\times3$.
> - SmallNORB: $n=48,600$ images,  $d_\textrm{data}=64\times64\times1$.
>
>
> > **Limited experiments: the datasets in the experiments are all about 3D shapes. How about 2D shapes? I would like to see how it can apply to the MNIST, another simple 2D digit.** / **"Have you applied the method on datasets other than 3D dataset? For example, MNIST."**
>
> We focused on standard benchmarks found in the disentanglement literature [Locatello et al., 2019a]. Please note that our experiments are not limited to 3D data. DSprites, one of our benchmark datasets contains images of 2D shapes that demonstrate our method's effectiveness also in this regime. All datasets in our systematic evaluation are 64x64 images containing approximately 10 independent factors of variation (e.g., shape, scale, rotation, color), which is the standard way of benchmarking disentangled representation learning methods.
>
> Under our operational definition of disentanglement - learning representations that separate underlying generative factors - MNIST is less suitable as it contains only one primary factor of variation (digit number). Even augmented versions of MNIST (e.g., colored and rotated) would only introduce 2-3 factors of variation. Consequently, MNIST is not commonly employed in disentanglement research, where the objective is to separate multiple independent generative factors.

---

> > ### Comment · Reviewer_zj5w · 2024-11-21
> > **About the computation of Gromov-Wasserstein distance**
> >
> > * You can check Remark 1 in [1] for reference. The computation of GW distance is expensive even with regularization.
> >
> > [1] Peyré, G., Cuturi, M., & Solomon, J. (2016, June). Gromov-wasserstein averaging of kernel and distance matrices. In International conference on machine learning (pp. 2664-2672). PMLR.
> >
> > * Can I understand correctly that your proposed approach is limited in the application of MNIST classification, thus making it weaker in some datasets?

---

> ### Author Response · Authors · 2024-11-20
> **Rebuttal to Reviewer zj5w [2/2]**
>
> > **"I believe that the robustness should also be important part but this work seems to ignore it. What if the data has out-of-distritbution points? In fact, DST is not robust in the noisy data setting."** / **"Have you tested the method on noisy point clouds? Or any other noisy dataset?"**
>
> We thank for you for this important question regarding robustness and out-of-distribution behavior.
>
> First, please note that robustness evaluations are not part of standard disentanglement benchmarks [1,2,3]. As our work aims to demonstrate primarily GMG's benefits within this established framework, we maintain consistency with these evaluation protocols.
>
> Second, while we do not explicitly measure robustness, prior work has shown that disentangled representations can enhance out-of-distribution generalization in downstream tasks [3,4,5], while providing improved interpretability and fairness.
>
> From a theoretical perspective, extending GMG to handle outliers and unbalanced data distributions could be achieved through unbalanced Gromov-Wasserstein [6], analogous to the unbalanced Monge Gap formulation in [7] but extended to quadratic OT.
>
> However, this extension would pose quite a few nontrivial practical and theoretical challenges. In particular, the GMG builds on comparing a ground truth coupling to an optimal one which, if computed using unbalanced relaxations, may no longer have the same mass, and using renormalization would create issues with differentiation.
>
> > **"Have you thought about acceleration?"**
>
> That's an excellent point, thanks for bringing it up. Indeed, we rely on acceleration techniques in our approach. As explained in the “Discrete Solvers” paragraph of Section 2.2, the entropic GW problem is solved using a mirror descent scheme that iteratively applies Sinkhorn's algorithm as a subroutine. To speed up each Sinkhorn step, we leverage an acceleration method. Specifically, for all our experiments, we use an adaptive momentum term, as introduced in [8], to accelerate the convergence of each Sinkhorn step.
>
> We use the OTT-JAX implementation of this scheme, detailed in its [documentation](https://ott-jax.readthedocs.io/en/latest/solvers/_autosummary/ott.solvers.linear.acceleration.Momentum.html) and demonstrated in this [tutorial](https://ott-jax.readthedocs.io/en/latest/tutorials/linear/000_One_Sinkhorn.html). The momentum parameters in our experiments are set following the guidance provided in the tutorial. In practice, we begin updating the momentum after 50 Sinkhorn iterations and set the momentum parameter to $\omega = 1.3$. We have clarified this point in the updated draft, see **Lines 301-302.**
>
>
> [1] Francesco Locatello, ..., Olivier Bachem. "Challenging common assumptions in the unsupervised learning of disentangled representations". In ICML 2019.
>
> [2] Francesco Locatello, ..., Michael Tschannen. "Weakly-supervised disentanglement without compromises". In ICML 2020.
>
> [3] Karsten Roth, ..., Diane Bouchacourt. "Disentanglement of correlated factors via hausdorff factorized support". In ICLR 2023.
>
> [4] Kyle Hsu, Will Dorrell, James C. R. Whittington, Jiajun Wu, Chelsea Finn. "Disentanglement via Latent Quantization". In NeurIPS 2023.
>
> [5] Vitória Barin-Pacela, Kartik Ahuja, Simon Lacoste-Julien, Pascal Vincent. "On the Identifiability of Quantized Factors". In CLeaR 2024.
>
> [6] Thibault Séjourné, François-Xavier Vialard, Gabriel Peyré. "The Unbalanced Gromov Wasserstein Distance: Conic Formulation and Relaxation". In NeurIPS 2021.
>
> [7] Luca Eyring, Dominik Klein, Théo Uscidda, Giovanni Palla, Niki Kilbertus, Zeynep Akata, Fabian Theis. "Unbalancedness in Neural Monge Maps Improves Unpaired Domain Translation". In ICLR 2024.
>
> [8] Tobias Lehmann, Max-K. von Renesse, Alexander Sambale, and Andr Uschmajew. "A note on overrelaxation in the sinkhorn algorithm". 2021.

---

> ### Author Response · Authors · 2024-11-21
> **Many thanks for reading our rebuttal!**
>
> Many thanks for reading our rebuttal, we are very grateful for your time and for engaging with us!
>
> Please do not hesitate to share any other concerns you may have, we will do our best to answer them during the rebuttal period.
>
> > **You can check Remark 1 in [1] for reference. The computation of GW distance is expensive even with regularization.
> [1] Peyré, G., Cuturi, M., & Solomon, J. (2016, June). Gromov-wasserstein averaging of kernel and distance matrices. In International conference on machine learning (pp. 2664-2672). PMLR.**
>
> Many thanks for this reference and for making your concern more clear to us with this remark, this is truly helpful.
>
> Our initial response was focused on clarifying the complexity $O(n^5)$ you mentioned vs. our claim in the paper of $O(n^3)$ (found in [1] as well), and even lower w.r.t. $n$ using considerations related to dimensionality. We are happy to clarify this point more carefully.
>
> First, we do use this reference in our Algorithm 2 (p.28), and it is indeed central to the OTT-JAX implementation that we use. We also prefer (and also refer to) the more detailed rates taken from the **[Scetbon, Peyre, Cuturi, Neurips 22]** paper, which supersede [1] and clarify the computational cost linked to dimension, when the cost matrices are computed from vectors, data or latents (as we do in this paper).
>
> This being said, and we should have been more upfront about this, but **the overhead due to GW to compute the GMG loss is negligible** in our work, which is of course a nice feature for a loss. We have clarified this in **L. 461 in a new revision**. We truly thank you for raising this issue repeatedly, this has helped improve our paper on that aspect. Sometimes as contributors we fail to report things that are obvious for us!
>
> The cost of running GW for a batch of $n=64$ points with the regularizations that we use (the setting adopted in the entire paper in **L.1383**) is of the order of **3 milliseconds on average**. Even with batch sizes ten times bigger, the cost would be still negligible, as shown in our computations below:
>
> - Batch size: 64, Average running time: 0.0032 ± 0.0003 s
> - Batch size: 128, Average running time: 0.0038 ± 0.0004 s
> - Batch size: 256, Average running time: 0.0054 ± 0.0004 s
> - Batch size: 512, Average running time: 0.0107 ± 0.0002 s
>
> (as a side note, notice that [1] was using point clouds of $n=500$ points, 8 years ago)
>
> In practice, the costs needed to compute and differentiating the GMG loss are dwarfed by the cost of backpropagating that loss towards updating the network parameters.
>
> Therefore, although we understand your spontaneous reservations, given the bad reputation of GW as a costly subroutine, rest assured this is really not an issue in this paper, nor will it be in more challenging applications of the GMG loss with higher dimensional datasets _as long as the mini-batch size remains in the hundreds_ (which is almost always the case in representation learning, due to network storage overheads).
>
> > **Can I understand correctly that your proposed approach is limited in the application of MNIST classification, thus making it weaker in some datasets?**
>
> We claim that our proposed approach (and the entire disentanglement literature) is not relevant to datasets that are **as simple as** MNIST.
>
> Disentanglement is an issue that arises when computing latent representations (using e.g. VAE) for complex data that have multiple latent factors (the goal is to have somewhat "nicely orthogonal / disentangled" factors). MNIST has only one latent factor (each data point can be dominantly explained using a number from 0 to 9).
>
> A useful analogy would be to state that it makes little sense to benchmark PCA methods on data that is dominantly 1D. As a result, the MNIST classification dataset is not used in the disentangled representations literature.
>
> Instead, we worked with the four image datasets listed above, with higher dimensions (64×64×3 and 64×64) and variability compared to MNIST's 28×28. These datasets are widely recognized as classical benchmarks for disentanglement, as referenced in [2] with close to 2k citations.
>
> [2] Challenging Common Assumptions in the Unsupervised Learning of Disentangled Representations, Francesco Locatello, Stefan Bauer, Mario Lucic, Gunnar Rätsch, Sylvain Gelly, Bernhard Schölkopf, Olivier Bachem, ICML 2019

---

> > ### Author Response · Authors · 2024-12-02
> > **Follow-up on Addressed Concerns – Reviewer zj5w**
> >
> > Dear Reviewer,
> >
> > Thank you for the valuable feedback on our work. This is a gentle reminder that the discussion phase concludes at 11:59 PM AoE on December 2, which is in approximately 24 hours.
> >
> > We hope our responses and supporting experiments have sufficiently addressed your concerns. Please feel free to let us know if you have any further questions or require additional clarifications.
> >
> > We appreciate your thoughtful comments and the time you have dedicated to reviewing our submission.
> >
> > The Authors

---

### Official Review · Reviewer_MkTy · 2024-11-05

**Soundness:** 3
**Presentation:** 3
**Contribution:** 3
**Rating:** 6
**Confidence:** 2

**Summary:**

This paper introduces a new method for disentangled representation learning based on quadratic optimal transport theory using Gromov-Monge maps. The authors leverage Gromov-Monge Gap (GMG) to measure how well a mapping preserves geometric features while transporting between distributions with minimal distortion. The approach aims to effectively combine geometric constraints with prior matching by finding mappings that optimally balance geometric preservation and distribution alignment. The method is validated on four standard disentanglement benchmarks (Shapes3D, DSprites, SmallNORB, Cars3D) showing consistent improvements over existing disentanglement approaches.

**Strengths:**

1. This paper provides a fresh perspective on combining geometric constraints with prior matching in disentanglement learning. Different from previous works that use direct penalties for geometric preservation, the authors propose using Gromov-Monge maps to find mappings that optimally preserve geometric features while aligning distributions. The theoretical analysis proves weak convexity of the GMG regularizer and provides precise characterization of convexity constants.

2. The authors have conducted comprehensive experiments across multiple benchmarks and architectural choices. Through thorough ablation studies, they demonstrate that angle preservation outperforms distance preservation, decoder regularization is more effective than encoder regularization, and the GMG shows better gradient stability compared to distortion-based approaches. Additionally, they show the method can be integrated with existing approaches like β-VAE and β-TCVAE to enhance their performance.

**Weaknesses:**

1. It remains unclear to me why the Gromov-Monge map framework is the optimal choice for combining geometric preservation with distribution alignment. While the authors show empirically that GMG outperforms direct distortion penalties, they don't fully explain why measuring suboptimality in the Gromov-Monge problem provides a better learning signal than other potential optimal transport formulations or geometric metrics.

2. My another concern is about the practicality of solving Gromov-Wasserstein problems during training. Although the authors provide computational complexity analysis and propose using entropic regularization with scaled costs, they don't thoroughly investigate how different solver parameters affect the trade-off between computational efficiency and accuracy of the approximation. The sensitivity analysis of the entropic regularization parameter $\epsilon_0$ and its impact on training dynamics needs more exploration.

**Questions:**

1. Why does angle preservation work better than distance preservation? Is this finding specific to the datasets tested?

2. How sensitive is the method to the entropic regularization parameter $\epsilon_0$?

---

> ### Author Response · Authors · 2024-11-20
> **Rebuttal to Reviewer MkTy [1/3]**
>
> Thank you for reviewing our paper and providing both positive feedback and constructive suggestions. We are happy to address your questions and remain available during the rebuttal period to clarify any further concerns.
>
> > **It remains unclear to me why the Gromov-Monge map framework is the optimal choice for combining geometric preservation with distribution alignment. While the authors show empirically that GMG outperforms direct distortion penalties, they don't fully explain why measuring suboptimality in the Gromov-Monge problem provides a better learning signal than other potential optimal transport formulations or geometric metrics.**
>
> We apologize if our explanation in the paper was unclear. Let us begin by stating our goal: **we aim to learn a mapping that preserves geometric features (like distances or angles) while transforming one distribution into another**. However, this creates an inherent tension - there is no guarantee that a perfectly geometry-preserving mapping exists between two arbitrary distributions. This means that a distribution-fitting term would necessarily compete with the distortion loss.
>
> This fundamental issue is precisely what we demonstrate in Figure 2. When using the distortion (DST), "we preserve geometric features but do not fit the marginal constraint, i.e., $T_\theta\sharp p\ne q$" [line 406-407] - the mapping preserves geometry but fails to match the target distribution. This makes DST suboptimal as it sacrifices distribution matching to achieve geometric preservation.
>
> The **GMG** resolves this issue by **measuring whether a mapping preserves geometry as much as theoretically possible while still achieving distribution alignment**. We precisely define this as: "The GMG quantifies the difference between the distortion incurred when transporting $r$ to $T\sharp r$ via $T$, and this minimal distortion" [line 252-253]. In other words, while DST simply measures geometric distortion, GMG measures how close we are to the optimal possible map.
>
> We highlight this crucial distinction: "The GMG offers the optimal compromise: it avoids the over-penalization induced by the distortion when fully preserving $c_\mathcal{X}, c_\mathcal{Y}$ is not feasible, yet it coincides with it when such full preservation is feasible."[line 268-269]
>
> This theoretical advantage is clearly demonstrated in Figure 2, where with GMG (fifth column), "we get the best compromise by approximating a Gromov-Monge map: we fit $T_\theta\sharp p= q$ while preserving the geometric features as much as possible."[line 408-409] This empirical demonstration validates that GMG enables learning mappings that both match distributions and preserve geometric structure as much as theoretically possible, making it the optimal choice for combining these objectives.

---

> ### Author Response · Authors · 2024-11-20
> **Rebuttal to Reviewer MkTy [2/3]**
>
> > **My another concern is about the practicality of solving Gromov-Wasserstein problems during training. Although the authors provide computational complexity analysis and propose using entropic regularization with scaled costs, they don't thoroughly investigate how different solver parameters affect the trade-off between computational efficiency and accuracy of the approximation. The sensitivity analysis of the entropic regularization parameter $\varepsilon_0$ and its impact on training dynamics needs more exploration.**
>
> > **How sensitive is the method to the entropic regularization parameter $\varepsilon_0$?**
>
>
> Thank you for bringing up this important point.
>
> **First, we were guilty of not clarifying an important piece of context in the initial version of our paper**: The cost of running GW (and therefore the GMG loss) for a batch of $n=64$ points with the regularization that we use (the setting adopted in the entire paper in **L.1383**) is of the order of **3 milliseconds on average**. Even with batch sizes ten times bigger, the cost would be still negligible (10ms). We have clarified this in **L.461**. Therefore the discussion on setting $\varepsilon$ is at this point mostly focused on smoothness and training dynamics as you mention, but never really impact (comparatively to the rest of the pipeline) computational time.
>
>
> To clarify, as noted below Table 1 (line 459), we did not **tune** $\varepsilon$ during our experiments. Instead, the regularization parameter $\varepsilon$ was automatically determined from basic statistics of the cost matrices and scaled by a fixed factor $\varepsilon_0$ (see line 6 in Algorithm 1), which we **consistently set to $\varepsilon_0 = 0.1$ for all experiments**.
> We opted not to tune $\varepsilon$ because our experimental results indicated that our pipeline was quite robust to this setting. This robustness is also observed in recent related work on the multimarginal matching gap (M3G), as shown in Figure 3 of this paper: https://arxiv.org/pdf/2405.19532.
>
> Our intuition for this is as follows: When using Sinkhorn (in the case of M3G) or GW/quadratic OT (in the case of our paper) to compute a training loss, $\varepsilon$ acts like a sharpness parameter (choosing to highlight more strongly pairs of points). While the loss value might change according to $\varepsilon$, that loss is optimized w.r.t. network variables that are on top of that loss. That bit seems to be not so impacted by the sharpness set by $\varepsilon$.
>
> When, on the other hand, Sinkhorn or GW’s output is used directly to make a prediction or learn a flow, e.g. for Monge maps [1,2] or Gromov-Monge maps [3], then, in our experience, $\varepsilon$ has a very strong impact and must be tuned more carefully. This is not the case in our paper. We also added this discussion to the newly added Appendix D.3.
>
> In contrast, when Sinkhorn or GW outputs are used directly for making predictions or learning flows in other works (such as when estimating Monge maps or Gromov-Monge maps), $\varepsilon$ has a significantly stronger impact and requires careful tuning.
>
> In response to your valuable feedback, we performed additional experiments to **validate** these findings. An **ablation** study on the entropic regularization scale $\varepsilon_0$ has been included in Appendix D.3 of the revision; please see the ``**Effect of $\varepsilon$ in disentanglement**'' section. The results indicate that performance remains **fairly stable** across a range of sensible $\varepsilon$ values.
>
> Furthermore, in the newly added Appendix D.3, we offer additional insights into the approximation and trade-off introduced by the entropic regularization strength $\varepsilon$ in the entropic GW solver. Specifically, refer to the `**Effect of $\varepsilon$ on the entropic GW solver**'' section. We also empirically analyze how $\varepsilon$ affects the solver's convergence speed.
>
> [1] Entropic estimation of optimal transport maps, Aram-Alexandre Pooladian, Jonathan Niles-Weed, https://arxiv.org/abs/2109.12004
>
> [2] Progressive Entropic Optimal Transport Solvers, Parnian Kassraie, Aram-Alexandre Pooladian, Michal Klein, James Thornton, Jonathan Niles-Weed, Marco Cuturi, https://arxiv.org/abs/2406.05061
>
> [3] GENOT: Entropic (Gromov) Wasserstein Flow Matching with Applications to Single-Cell Genomics, Dominik Klein, Théo Uscidda, Fabian Theis, Marco Cuturi, https://arxiv.org/abs/2310.09254

---

> ### Author Response · Authors · 2024-11-20
> **Rebuttal to Reviewer MkTy [3/3]**
>
> > **"Why does angle preservation work better than distance preservation? Is this finding specific to the datasets tested?"**
>
> That's an excellent question. Lines 468–470 provide an initial explanation: “This result is intuitive, because preserving the angles imposes a weaker constraint, allowing greater expressivity of the latent space. In practice, preserving scaled distances seems to excessively restrict the expressiveness of the latent space.” Indeed, since preserving distances inherently implies preserving angles, requiring only angles to be preserved imposes fewer constraints on the geometry of the latent space. For the four datasets we considered, this additional degree of freedom leads to better disentanglement. Given the diversity of the four datasets considered—particularly as the generative factors of interest differ for each—we believe this finding is not specific to these datasets.
>
> In addition, a theoretical argument based on Thm 3.5 supports this finding. As detailed in lines 339-341, directly below the statement of Thm 3.5, it has been shown that $\textrm{GMG}_r^{\langle\cdot,\cdot\rangle}$ (the GMG used to preserve angles) exhibits greater convexity compared to $\textrm{GMG}r^2$ (the GMG used to preserve scaled distances). Consequently, for a neural network $T({\theta})$, the loss $\theta \mapsto \textrm{GMG}r^{\langle\cdot,\cdot\rangle}(T({\theta}))$ is provably easier to optimize. This theoretical result aligns with our practical observation that preserving angles is more feasible than preserving scaled distances.

---

> > ### Author Response · Authors · 2024-12-02
> > **Follow-up on Addressed Concerns – Reviewer MkTy**
> >
> > Dear Reviewer,
> >
> > Thank you for the valuable feedback on our work. This is a gentle reminder that the discussion phase concludes at 11:59 PM AoE on December 2, which is in approximately 24 hours.
> >
> > We hope our responses and supporting experiments have sufficiently addressed your concerns. Please feel free to let us know if you have any further questions or require additional clarifications.
> >
> > We appreciate your thoughtful comments and the time you have dedicated to reviewing our submission.
> >
> > The Authors

---

### Official Review · Reviewer_Fss4 · 2024-11-12

**Soundness:** 4
**Presentation:** 3
**Contribution:** 3
**Rating:** 6
**Confidence:** 3

**Summary:**

This paper introduces a novel approach for unsupervised disentangled representation learning, aiming to balance prior alignment with the preservation of geometric features within latent spaces. Achieving disentangled representations without supervision is difficult; recent studies suggest that maintaining the geometric structure of the data distribution can promote disentanglement. However, aligning data with a prior distribution while fully preserving geometric properties often requires trade-offs. Therefore, the authors introduce the Gromov-Monge Gap (GMG), a regularizer based on optimal transport theory that learns a mapping with minimal distortion.

**Strengths:**

- The paper is well-organized and clearly written.
- Introducing the Gromov-Monge Gap as a regularizer provides a fresh approach to minimizing geometric feature distortion, presenting a promising advancement for unsupervised disentangled representation learning.
- The authors offer valuable insights, particularly in showing that preserving angles through cosine similarity can enhance disentanglement more effectively than preserving distances.
- The proof demonstrating the GMG as a weakly convex regularizer is a useful contribution for future work in this area.

**Weaknesses:**

- The paper provides limited discussion on the scalability of GMG for high-dimensional real world datasets.
- There is an absence of ablation studies on key hyperparameters, such as entropic regularization.

**Questions:**

- Considering the entropic regularization parameter plays a role in the optimization, how does adjusting the hyperparameter affect the computational efficiency and convergence of the GMG computation? An ablation study of varying hyperparameters would be nice.
- How scalable is GMG to high-dimensional real world datasets for the VAE configuration?

---

> ### Author Response · Authors · 2024-11-20
> **Rebuttal to Reviewer Fss4 [1/2]**
>
> Thank you for taking the time to read our paper and for providing both encouraging feedback and constructive comments. We are pleased to address your questions and are available throughout the rebuttal period to clarify any additional concerns.
>
> > **The paper provides limited discussion on the scalability of GMG for high-dimensional real world datasets.**
>
> As discussed in the **Efficient and Stable Computation section** of § 3.3 (lines 286–306) and detailed in **Algorithm 1**, the time complexity of GMG computation is **linear in the data dimension** for common cost functions $c_\mathcal{X}$ and $c_\mathcal{Y}$, such as inner product, cosine similarity, $\ell_p^q$ distances, or more generally, most CPD kernels.
>
> In particular, in lines 310-315, we explain that for the cost functions used in our experiments (cosine similarity and scaled Euclidean distance), the time complexity is $\mathcal{O}(n^2(d_\mathcal{X} + d_\mathcal{Y}))$, where $n$ is the batch size, and $d_\mathcal{X}$ and $d_\mathcal{Y}$ are the data and latent space dimensions. To simplify, we can rewrite it as $\mathcal{O}(n^2(d_\mathrm{latent} + d_\mathrm{data}))$. Since $d_\mathrm{latent} \ll d_\mathrm{data}$ by design, the dominant term is $d_\mathrm{data}$, resulting in an overall complexity of $\mathcal{O}(n^2 d_\mathrm{data})$, which is then linear in the data dimension. We now insist on this point in the manuscript, see lines 315-316.
>
> This linear dependency on the data dimension makes our method applicable to high-dimensional real-world datasets. We evaluate GMG on the majority of standard disentanglement research benchmarks, which demonstrate that it successfully scales to standard $64 \times 64$ and $64 \times 64 \times 3$ image datasets. We have now explicitly included the image dimensions in the manuscript, see lines 459-461. We include them here for completeness, along with the number of images per dataset:
>
> - **Shapes3D**: $n=480,000$ images, $d_\textrm{data}=64 \times 64 \times 3$.
> - **dSprites**: $n=737,280$ images, $d_\textrm{data}=64 \times 64 \times 1$.
> - **Cars3D**: $n=17,064$ images, $d_\textrm{data}=64 \times 64 \times 3$.
> - **SmallNORB**: $n=48,600$ images, $d_\textrm{data}=64 \times 64 \times 1$.
>
> While we do agree that extending our experiments to much larger, more compute-intensive real-world datasets would be a welcome addition to our work, larger-scale real-world evaluation of disentanglement is particularly challenging, as it requires precise labeling of ground truth factors and a clearly controlled generative process. The generation of such datasets is still very much an ongoing research problem.

---

> ### Author Response · Authors · 2024-11-20
> **Rebuttal to Reviewer Fss4 [2/2]**
>
> > **There is an absence of ablation studies on key hyperparameters, such as entropic regularization.** [...]
>
> > **Considering the entropic regularization parameter plays a role in the optimization, how does adjusting the hyperparameter affect the computational efficiency and convergence of the GMG computation? An ablation study of varying hyperparameters would be nice.**
>
> Thanks for pointing out this important point.
>
> To clarify: when we ran experiments, as we mentioned below Table 1 (line 459), **we did not tune $\varepsilon$**. The regularization level $\varepsilon$ was selected automatically from elementary statistics on the cost matrices and multiplied by one “knob” was $\varepsilon_0$ (see line 6 in Algorithm 1), and **we always set it to $\varepsilon_0 = 0.1$**.
>
> The reason we did not tune it at the time is that we saw experimentally that our pipeline was fairly robust to that choice. This robustness is echoed in the recent related work of the multimarginal matching gap (M3G), in Figure 3 from https://arxiv.org/pdf/2405.19532.
>
> Thanks to your insightful feedback, we conducted additional experiments to **validate** these results. We included an **ablation study** on the entropic regularization scale $\varepsilon_0$ in Appendix D.3 of the revision. See, in particular, the "**Effect of $\varepsilon$ in disentanglement**" paragraph (L.1504). The results demonstrate fairly **stable performance** across multiple sensible $\varepsilon$ values.
>
> Our intuition (summarized from new addition in L. 1504 in appendix) for this is as follows: When using Sinkhorn (in the case of M3G) or GW/quadratic OT (in the case of our paper GMG) to compute a training loss, $\varepsilon$ acts like a sharpness parameter (choosing to highlight more strongly pairs of points). While the loss value might change according to $\varepsilon$, that loss is optimized w.r.t. network variables that are on top of that loss. That bit seems to be not so impacted by the sharpness set by $\varepsilon$.
>
> When, on the other hand, Sinkhorn or GW’s output is used directly to make a prediction or learn a flow, e.g. for Monge maps [1,2] or Gromov-Monge maps [3], then, in our experience, $\varepsilon$ has a very strong impact and must be tuned more carefully. This is not the case in our paper. We also added this discussion to the newly added Appendix D.3.
>
> Furthermore, we provide insights into the approximation and trade-off introduced by the entropic regularization strength $\varepsilon$ in the entropic GW solver. See, in particular, the "**Effect of $\varepsilon$ on the entropic GW solver**" paragraph. Additionally, we empirically analyze how $\varepsilon$ influences the solver's convergence speed.
>
> [1] Entropic estimation of optimal transport maps, Aram-Alexandre Pooladian, Jonathan Niles-Weed, https://arxiv.org/abs/2109.12004
>
> [2] Progressive Entropic Optimal Transport Solvers, Parnian Kassraie, Aram-Alexandre Pooladian, Michal Klein, James Thornton, Jonathan Niles-Weed, Marco Cuturi, https://arxiv.org/abs/2406.05061
>
> [3] GENOT: Entropic (Gromov) Wasserstein Flow Matching with Applications to Single-Cell Genomics, Dominik Klein, Théo Uscidda, Fabian Theis, Marco Cuturi, https://arxiv.org/abs/2310.09254
>
> > **How scalable is GMG to high-dimensional real world datasets for the VAE configuration?**
>
> As outlined in our initial response, the time complexity of GMG is linear with respect to the data dimension, making it scalable to high-dimensional datasets. We successfully applied it to image datasets with dimensions $64 \times 64$ and $64 \times 64 \times 3$.

---

> > ### Author Response · Authors · 2024-12-02
> > **Follow-up on Addressed Concerns – Reviewer Fss4**
> >
> > Dear Reviewer,
> >
> > Thank you for the valuable feedback on our work. This is a gentle reminder that the discussion phase concludes at 11:59 PM AoE on December 2, which is in approximately 24 hours.
> >
> > We hope our responses and supporting experiments have sufficiently addressed your concerns. Please feel free to let us know if you have any further questions or require additional clarifications.
> >
> > We appreciate your thoughtful comments and the time you have dedicated to reviewing our submission.
> >
> > The Authors

---

### Author Response · Authors · 2024-11-28
**Author Rebuttal to Reviewers**

Dear Reviewers,

We would like to thank you for your time, and for your interest in our submission. We are grateful for your many encouraging comments, covering both the methodological and theoretical aspects of our contribution:

- “*Introducing the Gromov-Monge Gap as a regularizer provides a fresh approach [...], presenting a promising advancement for unsupervised disentangled representation learning. [...] The proof demonstrating the GMG as a weakly convex regularizer is a useful contribution for future work in this area*” (**Reviewer Fss4**)

- "*This paper provides a fresh perspective on combining geometric constraints with prior matching in disentanglement learning. [...] The theoretical analysis proves weak convexity of the GMG regularizer and provides precise characterization of convexity constants.*" (**Reviewer MkTy**)

as well as its potential practical impact:

- “*The authors offer valuable insights, particularly in showing that preserving angles through cosine similarity can enhance disentanglement more effectively than preserving distances.*” (**Reviewer Fss4**)

- "*The authors have conducted comprehensive experiments across multiple benchmarks and architectural choices. Through thorough ablation studies, they demonstrate that angle preservation outperforms distance preservation, decoder regularization is more effective than encoder regularization [...].*" (**Reviewer MkTy**)

- "*The empirical performance is good, indicating that adding this neural GM gap can improve the disentanglement score for many architectures.*" (**Reviewer 5P64**)

Most importantly, we would like to thank you for your **actionable and constructive criticism**. We have taken into consideration your comments, leading to a few cosmetic changes and addition of experiments (highlighted in blue in the manuscript). Here is a summary of the changes/additions we have made:

- We have conducted new experiments with results presented in **Appendix D.3** of the revised manuscript showing a consistent performance across various reasonable values of $\varepsilon_0$ for GMG (this was requested by Reviewers `Fss4` and `MkTy`). This robustness aligns with findings from related work, as illustrated, for instance, in [1, Figure 3].

- We have clarified in **L.461** of the revised manuscript that the computational overhead of using GW to compute the GMG loss is **negligible**. This was an important concern raised by Reviewers `Fss4`, `zj5w` and `5P64` that we have clarified. An important aspect of our work is that we can **set** the mini-batch size (here $n = 64$ points, **L.1283**) making GW computations very cheap (they average approximately 3 milliseconds at each batch in our main experiments). Even with larger mini-batches (e.g. 512) the cost of evaluating GMG does not exceed 15ms.

- We have added, following Reviewer `5P64` request, **additional experiments on an upscaled $128 \times 128$** version of the Shapes3D dataset (Appendix D.4). These results demonstrate that the GMG regularizer can be scaled up to larger **data dimension** and still yield performance gains, with consistent improvements in DCI-D scores across all VAE variants. Even with this augmented setup at higher-resolution, GMG only increases computation time by 5% ($\beta$-TCVAE: 4h 30min vs $\beta$-TCVAE + GMG: 4h 45min), with significant gains.

- As requested by Reviewer `5P64`, we provide traversal visualizations at both $64\times 64$ and $128\times 128$ resolutions (Appendix D.5). These visualizations demonstrate effective disentanglement of factors like wall hue, floor hue, object hue, and scale at both resolutions.

We believe these small changes reinforce the message of our paper that GMG is a cheap, geometrically intuitive and suitable regularizer to learn disentangled representations. We are very grateful for your time and efforts during this rebuttal, your actionable criticism, and for your overall interest in the paper.

[1] Zoe Piran, Michal Klein, James Thornton, and Marco Cuturi. Contrasting multiple representations with the multi-marginal matching gap, 2024, https://arxiv.org/pdf/2405.19532.

---

### Public Comment · ~Hongteng_Xu1 · 2025-12-24

Glad to see the idea of Gromovized Monge gap (GMG) is accepted by the ICLR community. Notably, this regularizer can also be derived from the information bottleneck perspective https://arxiv.org/pdf/2405.15505, which might further support its rationality in the context of representation learning.

---

### Meta-Review · Area_Chair_9jXd · 2024-12-23

**Metareview:**

This paper proposes a quadratic optimal transport-based disentanglement approach to match data distributions with the priors while preserving geometry features. The problem is formulated  using Gromov-Monge maps that transport one distribution onto another with minimal distortion of predefined geometric features. The effectiveness of the method is demonstrated on standard benchmarks, outperforming similar approaches that leveraging geometric constraints. The main concerns from the reviewers are 1) the unclear motivation of using Gromov-Monge, 2) the computational complexity is relatively high, 3) the datasets used in the experiments are small scale. The authors did a great job in the rebuttal and clarified the concerns. The remaining concern on MNIST, and Facol3D, Issac3D datasets does not seem to be a strong reason to reject this paper. I recommend the paper to be accepted, though the overall rating is below 6.

**Additional Comments On Reviewer Discussion:**

The main concerns from the reviewers are 1) the unclear motivation of using Gromov-Monge, 2) the computational complexity is relatively high, 3) the datasets used in the experiments are small scale. The authors have clarified in the rebuttal that Gromov-Monge should be used due to the quadratic optimal transport. Also, the computational complexity is given, which is linear in the data dimensions square of batch size, which is not expensive.. The two reviewers with rating 6 did not participate in the discussion. The two reviewers with rating 5 had a thorough discussion with the authors, and I think the authors have clarified all the issues. The remaining concern on MNIST, and Facol3D, Issac3D datasets does not make much sense to me, as the datasets in the paper are commonly used ones for disentanglement. I think the reviewers have addressed all the concerns.

---

### Decision · Program_Chairs · 2025-01-22

Accept (Poster)